# Contribution of local and remote anthropogenic aerosols to a record-breaking torrential rainfall event in Guangdong province, China

Z. Liu[1,2,3], Y. Ming[5], C. Zhao[6], N.C. Lau[1,2,4], J.P. Guo[7], M. Bollasina[3], Steve H.L. Yim[1,4,2]

[1]Institute of Space and Earth Information Science, The Chinese University of Hong Kong, Hong Kong, China
5 [2]Institute of Environment, Energy and Sustainability, The Chinese University of Hong Kong, Sha Tin, N.T., Hong Kong
[3]School of Geosciences, University of Edinburgh, UK
[4]Department of Geography and Resource Management, The Chinese University of Hong Kong, Sha Tin, N.T., Hong Kong
[5]Geophysical Fluid Dynamics Laboratory/NOAA, Princeton, New Jersey, USA
[6]School of Earth and Space Sciences, University of Science and Technology of China, Hefei, Anhui, China
10 [7]State Key Laboratory of Severe Weather, Chinese Academy of Meteorological Sciences, Beijing 100081, China

*Correspondence to*: Steve H.L. Yim (steveyim@cuhk.edu.hk)

**Abstract.** A torrential rainfall case, which happened in Guangdong Province during December 14–16, 2013, broke the historical rainfall record in the province in terms of duration, affected area, and accumulative precipitation. The influence of anthropogenic aerosols on this extreme rainfall event was examined using a coupled meteorology-chemistry-aerosol model. 15 Enhancement of precipitation in the estuary and near the coast up to 33.7 mm was mainly attributed to aerosol-cloud interactions, whereas aerosol-radiation interactions partially compensated 14% of the precipitation increase. Further analysis of different hydrometeors and latent heat sources suggests that the ACI effects on intensifying the precipitation can be divided into two stage: cold rain enhancement in the former stage while warm rain in the latter. Responses of precipitation to changes in anthropogenic aerosols concentrations from local (i.e., Guangdong province) and remote (i.e., outside Guangdong province) 20 sources were also investigated through simulations with reduced aerosol emissions from either local or remote sources. Accumulated aerosol concentration from local sources aggregated mainly near the surface and diluted quickly after the precipitation initiated. By contrast, aerosol concentration from remote emissions extended up to 8 km and lasted much longer before decreasing until peak rainfall began, because aerosols were continuously transported by the strong northerly. The patterns of precipitation response to remote and local aerosols concentrations resembled each other. However, compared with 25 local aerosols through warm rain enhancement, remote aerosols contributed more than twice the precipitation increase via intensifying both cold and warm rain, occupying a predominant role. Ten times of the emission sensitivity test resulted in about ten times of $PM_{2.5}$ concentration compared with the control run. Warm rain is drastically suppressed in 10× run. Compared with CLEAN experiment, the precipitation and cloud property changes in 10× run also resembled that in the control run, but with much greater magnitude. With aerosols, the precipitation average over Guangdong province decreased by 1.0 mm but 30 increased by 1.4 mm in the control run by comparing with CLEAN run. We noted that the precipitation increase was concentrated within a more narrowed downstream region of aerosol source, whereas the precipitation decrease was more dispersed across the upstream region. This indicates that the excessive aerosols not only suppress rainfall but also change the spatial distribution of precipitation, increasing the rainfall range, thereby potentially exacerbating flood and drought elsewhere.

This study highlights the importance of considering aerosols in meteorology to improve extreme weather forecasting. Furthermore, aerosols from remote emissions may outweigh those from local emissions in the convective invigoration effect.

## 1 Introduction

Synoptic weather is a key factor driving air pollution events through photochemical, turbulence, wet deposition, and transport processes (Ding et al., 2009; Guo et al., 2017; Liu et al., 2001; Madronich, 1987). Numerous studies have predicted air quality either numerically or statistically based on weather conditions (Dutot et al., 2007; Otte et al., 2005). In recent years, more and more efforts have been made to identify the influence of aerosols on synoptic weather (Ding et al., 2013; Grell et al., 2011), particularly on different types of extreme weather, such as tropical cyclone (Wang et al., 2014; Zhao et al., 2018), hail storm (Ilotoviz et al., 2016), and extreme rainfall (Fan et al., 2015; Zhong et al., 2015). However, the climate effects of aerosols have long been analyzed (Hansen et al., 1997; Myhre et al., 2013; Twomey, 1977).

For decades, China has been affected by severe pollution induced by rapid urbanization and economic development (He et al., 2002). The Pearl River Delta (PRD) region, situated on the south coast of China, is one of the most developed and also the most polluted regions. The aerosol optical depth retrieved from the Moderate Resolution Imaging Spectroradiometer is typically higher than 0.6 in Guangzhou, a megacity in the PRD region (Wu et al., 2005).

In addition to reducing visibility and inducing respiratory diseases (Cohen et al., 2015; Gu and Yim, 2016; Chen et al., 2017), high aerosol concentrations can also affect weather and climate through interactions with radiation and clouds (Bollasina et al., 2011; Lau and Kim, 2006; Wang et al., 2011). Aerosols absorb and scatter solar radiation and serve as cloud condensation nuclei and ice nuclei, which are referred to as aerosol-radiation interactions (ARI) and aerosol-cloud interactions (ACI), respectively (IPCC, 2013). Both ARI and ACI influence deep convection and hence precipitation (Fan et al., 2008, 2013; Koren et al., 2004; Liu et al., 2018; Rosenfeld et al., 2008; Fan et al., 2018). Liu et al. (2018) found that ARI suppressed deep convection by reducing the relative humidity in the middle-upper troposphere and weakening the upward motion. Fan et al. (2015) revealed that ARI weakened convergence, enhanced atmospheric stability, and suppressed convection in the basin during the daytime. Excess moist static energy was transported to mountains, thus generating heavy rainfall at night. This suppression effect is dramatically modulated by the intensity of synoptic forcing (Zhong et al., 2017). Compared with the effects of ARI, those of ACI on deep convection and precipitation have received more attention and are more controversial in both observational and modeling studies. Increased aerosols can suppress or enhance precipitation depending on environmental conditions such as humidity, cloud type, cloud phase, and vertical wind shear (Khain, 2009; Lee et al., 2008; Tao et al., 2012). Khain (2009) and Fan et al. (2007) have reported that increases in humidity generate more condensate than lost, resulting in more precipitation from deep convective clouds, especially in a polluted environment. Studies have reported that aerosols inhibit precipitation from shallow clouds (Andreae et al., 2004; Chen et al., 2016; Rosenfeld, 2000), whereas they invigorate deep convection with warm (>15°C) cloud bases (Bell et al., 2008; Koren et al., 2010, 2014). By contrast, smaller cloud droplets induced by aerosols could remain liquid near or above 0°C when lacking ice nuclei, inhibiting precipitation (Cui et

al., 2006; Rosenfeld and Woodley, 2000). Fan et al. (2009, 2012) have suggested that increased aerosols enhanced convection under weak wind shear and suppressed convection under strong wind shear by increasing evaporative cooling for an isolated storm. However, the evaporative cooling induced by aerosols has also been found to enhance precipitation under strong wind shear for cloud systems (Lee et al., 2008; Tao et al., 2007). Recently, Fan et al. (2018) found that the latent heat release could be mainly attributed to condensational heating rather than ice-related processes at upper levels, differing from cold convective invigoration (Rosenfeld et al., 2008).

The competition between ARI and ACI has been discussed on both cloud-resolving scale (Lin et al., 2017; Wang et al., 2018) as well as regional scale (Wang et al., 2016). Fan et al. (2008) suggested that the suppressive effects of ARI can outweigh the invigorative effects of ACI on deep convection and precipitation as the absorption of aerosols enhances. Koren et al. (2008) showed that the net effect of two opposite influences, those of ARI and ACI, on clouds over the Amazon depends on the initial cloud fraction. Large cloud cover fractions were mostly invigorated by ACI, whereas small cloud cover fractions were suppressed by ARI. Different aerosol types can also be a critical factor to their radiative or microphysical properties, thus determining the invigoration or suppression effect of aerosols on deep convection (Jiang et al., 2018). The precipitation enhancement in the downwind area of a polluted environment could be induced mainly by either ARI or ACI (Fan et al., 2015; Zhong et al., 2015). Both studies have focused on summer extreme rainfall cases because most extreme rainfall events occur in summer over China (Fu et al., 2013).

We selected a torrential rainfall case in winter, which broke the record of Guangdong Province since 1951 in terms of duration, affected area, and cumulative rainfall over the PRD region, to further understand the combined effects and relative importance of ARI and ACI on precipitation. Before this heavy rainfall, the PRD region was affected by strong haze with $PM_{2.5}$ concentrations approaching 174 µg m$^{-3}$. The significant transboundary nature of air pollution in China has been well recognized (e.g., Gu and Yim, 2016). Effects of local and remote aerosol emissions on monsoons and associated precipitation, particularly the Indian summer monsoon, have been examined in recent years (Bollasina et al., 2014; Cowan and Cai, 2011; Guo et al., 2016b; Jin et al., 2016), which was comprehensively reviewed by Li et al. (2016). The effects of local and remote aerosol emissions on extreme rainfall events remain mostly unexplored. In addition, given the strong monsoonal flow and severe air pollution over the northeast of China (Figure S1b), the aerosol concentrations could either from local emissions or transport by prevailing northeasterly. A critical question, therefore, is whether aerosol concentrations that affected this extreme rainfall case originated from local or remote aerosol emission sources. The remainder of this study proceeds as follows: Section 2 describes the regional model associated with the experimental design as well as the observation datasets of this study. Main findings on the effects of aerosols on the extreme rainfall event are discussed in section 3. The main conclusions are summarized and discussed in section 4.

## 2 Model configurations and observational datasets

The principal tool for this work was the Weather Research and Forecasting (WRF) model coupled with Chemistry (WRF-Chem) v3.5.1 (Grell et al., 2005), with some recent improvement by the University of Science and Technology of China (Zhao et al., 2013a, 2014, 2016; Hu et al., 2016). The details of the WRF-Chem configuration are documented in section 2.1, followed by model experiment design in section 2.2. The observational datasets used for validating the simulated precipitation performance, along with hourly in situ $PM_{2.5}$ observations are described in section 2.3.

### 2.1 WRF-Chem

WRF-Chem is a fully online model coupled with gas-phase chemistry mechanisms and aerosol physiochemical modules. In this model, chemical and meteorological components use the same grid coordinates, time steps, transport schemes, and subgrid physics. The meteorological component (WRF) of this coupled model uses an Eulerian dynamical core with a nonhydrostatic solver (Skamarock et al., 2008). Gas-phase chemical reactions are estimated using the carbon bond chemical mechanism (Zaveri and Peters, 1999). Aerosol physics and chemistry are treated using the Model for Simulating Aerosol Interactions and Chemistry (MOSAIC) scheme (Zaveri et al., 2008) with aqueous chemistry. The aerosol size distribution is represented by four discrete size bins within the MOSAIC scheme: 0.039–0.156 μm, 0.156–0.625 μm, 0.625–2.5 μm, and 2.5–10 μm (Fast et al., 2006). The approach to aerosol dry deposition is based on Binkowski and Shankar (1995). In-cloud (rainout) and below-cloud (washout) removal of aerosols by resolved clouds and precipitation are simulated following Easter et al. (2004) and Chapman et al. (2009), respectively. The transport and wet removal of aerosols by convective clouds are also considered using the Kain–Fritsch (KF) scheme (Kain and Fritsch, 1990) following Zhao et al. (2009, 2013b). The major physical schemes of meteorological components comprise the KF cumulus scheme; the Yonsei University (YSU) planetary boundary layer (PBL) scheme (Hong et al., 2006); the National Center for Environmental Prediction, Oregon State University, Air Force, and Hydrologic Research Lab's (NOAH) land surface model (Chen and Dudhia, 2001); the Morrison two-moment scheme for cloud microphysics (Morrison et al., 2009); and the rapid radiative transfer for global (RRTMG) for both longwave and shortwave radiation schemes (Iacono et al., 2008). Aerosol interactions with shortwave and longwave radiation are incorporated into the model by linking aerosol optical properties, including optical depth, single-scattering albedo, and asymmetry factor, to RRTMG shortwave and longwave schemes, respectively (Zhao et al., 2010, 2011). The effects of ACI are estimated by considering the activation of aerosols to form cloud droplets based on the maximum supersaturation in the Morrison microphysical scheme (Chapman et al., 2009; Yang et al., 2011).

### 2.2 Experiment design

WRF-Chem simulations are conducted to investigate the effect of aerosols on the extreme rainfall event of December 14–16, 2013. Unless specified, all time points in this study refer to local standard time (LST), which is equal to UTC+8. Two nested grids (run simultaneously with one-way nesting) cover most of China (87.47°–131.67° E, 11.42°–41.22° N) and Guangdong

province (109.59°–117.32° E, 20.07°–25.62° N) with a horizontal resolution of 20 km and 4 km, respectively (Figure S1). The cumulus scheme is turned off in the inner domain. Both nested grids use 41 vertical levels extending from the surface to 100 hPa. The meteorological initial and boundary conditions (ICs and BCs) are derived from 6-hourly National Center for Environmental Prediction global final analysis data with a horizontal resolution of 1° × 1°. The 6-hourly chemical ICs and BCs are generated from the Model for Ozone and Related Chemical Tracer version 4 (MOZART-4), which is an offline global chemical transport model suited for tropospheric studies, at a horizontal resolution of 1.9° × 2.5° with 56 vertical levels (Emmons et al., 2010). Anthropogenic emissions are obtained from the Emissions Database for Global Atmospheric Research Hemispheric Transport of Air Pollution v2 inventory (Janssens-Maenhout et al., 2015) for the year 2010 with a resolution of 0.1° × 0.1° (http://edgar.jrc.ec.europa.eu/htap_v2/). Biomass burning emission data are extracted from FINN 1.5 (Wiedinmyer et al., 2010). Dust and sea salt emission schemes are updated following Zhao et al. (2010) and Zhao et al. (2013a), respectively. The results show marginal differences between simulations with and without dust and sea salt emissions (figure not shown) in our study case; possible reasons for this are discussed in section 4.

Six sets of experiments are performed in total (Table 1). To isolate robust signals from the model's natural variations, five ensemble members with perturbed ICs at 3-h intervals are conducted for each experiment. The simulations started from 08Z to 20Z on December 13 with 3-h intervals, and all end at 02Z on December 17. The simulation before December 14 is for model spin up, and the following analysis focuses on the results from December 14–16. In the first experiment (CTL), current emissions are used in the simulation with both ARI and ACI effects included (Table 1). Following Fan et al. (2015), we scale the anthropogenic and fire emissions by a factor of 0.1 and perform the CLEAN simulation. We adjust the factor to 0.1 from 0.3 in Fan et al. (2015) to represent the background aerosol concentration as the emissions in 2010 is much higher than that in 2006 (Chang et al., 2018). It is used to mimic the situation in which the background of aerosol concentrations serves as cloud condensation nuclei before the economic development in China. The differences between CTL and CLEAN denote the total effects of aerosols including both ARI and ACI effects on this extreme rainfall case. To examine the role and relative importance of ARI and ACI, the ARIoff run is conducted based on CTL run by excluding the ARI effect. Thus, the differences between CTL and ARIoff represent ARI effects (Zhong et al., 2015). The ACI effects are approximated by looking at differences between CTL – CLEAN and CTL – ARIoff. To distinguish and isolate the effects induced by local (i.e., domain 2, Guangdong province) emissions and remote (i.e., domain 1, outside Guangdong province) emissions, two other experiments are designed. In D1 (Table 1) experiment, the ICs, BCs, and emissions are kept as same with control run for domain 1. Meanwhile, the ICs and emissions are scaled by a factor of 0.1 for domain 2. Similarly, in D2 (Table 1) experiment, the ICs, BCs, and emissions are scaled by a factor of 0.1 for domain 1. The ICs and emissions are kept as same with control run for domain 2. Note that the offline chemical BCs extracted from MOZART are only applicable to domain 1. Along with CTL run, these experiments allow us to interpret and ascertain aerosol-related changes that would have occurred with either local or remote aerosol emissions by observing differences between CTL minus CLEAN and either D2 minus CLEAN or D1 minus CLEAN. To test the sensitivity of precipitation to aerosol concentrations, one more experiment for extreme polluted case is

conducted. In parallel to that in CLEAN run, we scale the emissions and chemical ICs and BCs in control run by a factor of 10 (10×).

### 2.3 Observational datasets

The model-simulated precipitation performance is evaluated with satellite-based precipitation products and in situ rainfall observations.

Climate Prediction Center morphing technique (CMORPH) data produced by the National Oceanic and Atmospheric Administration covering the period from December 2002 to present are used. In this technique, infrared geostationary satellites observe the motion vectors of precipitation patterns to generate half-hourly precipitation estimates by using passive microwave (PMW) sensors. Time-weighted linear interpolation is exploited to morph the shape and intensity of precipitation features

when and where PMW data are unavailable. This provides data for global (60° S–60° N) precipitation analysis with a horizontal resolution of 0.07277° (approximately 8 km at the equator) and temporal resolution of 30 minutes. More details of CMORPH products are documented by Joyce et al. (2004).

The in situ hourly precipitation dataset is developed at the National Meteorological Information Center of the China Meteorological Administration (source: http://data.cma.cn). A total of 115 stations are within domain 2. Their locations are

represented as colored circles in Figure 2a.

The ERA-Interim version 2 is used to evaluate the model performance in simulating large-scale circulation. This data is a global atmospheric reanalysis containing data since 1979, provided by the European Centre for Medium-Range Weather Forecasts (ECMWF) (Dee et al., 2011). The data is available at a horizontal resolution of approximately 0.25˚ which is comparable to the resolution of domain 1.

The hourly $PM_{2.5}$ concentration in situ dataset is obtained from the website of the Ministry of Environmental Protection (source: http://106.37.208.233:20035) (Zhang and Cao, 2015). In total, 58 stations are within domain 2. Their locations are denoted as colored circles in Figure 1c.

## 3 Results

During December 14–16, 2013, there is a rare continuous rainstorm over most of Guangdong Province. The 3-day accumulated

rainfall at most stations exceeded 100 mm (Figure 2a), which may benefit winter and spring water usage, promote air cleaning, and reduce forest fire risk. This is the most extreme precipitation event in the province in terms of duration, affected area, and cumulative rainfall in December since the meteorological record of Guangdong province set in 1951 (Deng et al., 2015). The mid-tropospheric flow pattern, with a ridge to the northeast of the Tibet Plateau and a trough over the west of the Indo-China Peninsula, is favorable for cold air moving southward, whereas moist and warm air from the Bay of Bengal and the South

China Sea move northward (see Figure S2). The meeting of these two flows results in intense convergence at lower levels over domain 2 (Figure 1b), resulting in strong deep convection indicated by bright white color in the natural-color satellite image

captured by NASA's Terra (Figure 1a). The cloud top temperature average over the land in domain 2 is lower than –15 °C almost everywhere with minimum reaching about –35 °C (Figure S1b). Before the study case occurs, Guangdong province is affected by severe pollution on December 13. The hourly-averaged PM$_{2.5}$ concentrations exceeded 100 µg m$^{-3}$ in the delta region, peaking at 173.58 µg m$^{-3}$ (Figure 1c). The area to the north of Guangdong province, including Zhejiang, Jiangsu, and Anhui provinces, is blanketed in grey haze in the satellite image (Figure 1a). Note the grey haze area is smog, whereas whiter areas with more defined features are clouds. The column-integrated PM$_{2.5}$ concentrations in these areas reach up to 2000 µg m$^{-2}$ during December 14–16, 2013, in the simulated control run (Figure 1b). Strong prevailing northeasterly winds south of 30° N along the east coast of China indicates a strong monsoonal flow (Chang et al., 2006). The patterns of circulation and pollutant are favorable for aerosol transport to the south of China. Built on the observational and modeling works discussed above, we examine in section 3.2 the total effects and relative importance of ARI and ACI on this extreme rainfall event. We also distinguish and isolate the response to local and remote aerosol emissions in section 3.3. In section 3.4, the sensitivity of precipitation to aerosol emissions is explored.

## 3.1 Model evaluation compared with observational datasets

The 500-hPa geopotential height and wind pattern simulated in the control run are evaluated with ERA-interim data (Figure S2). The model well replicates the trough over the west of the Indo-China Peninsula and sub-tropical high over the South China Sea and northwest Pacific. The pattern correlations of 500 h-Pa geopotential height reaches 0.99 at the 99% significance level. Model simulated PM$_{2.5}$ concentration is evaluated by comparing with the 58 in-situ station data over Guangdong Province. The spatial distribution of PM$_{2.5}$ concentration is generally reproduced with high concentration over mega cities and low over surrounding areas (Figure S6). The failure to capture the hot spot near the estuary may be related to the coarse resolution or uncertainty of emissions. In the time series, both the simulation and observation show a dramatically decreasing trend of PM$_{2.5}$ concentrations once the rainfall initiated (Figure S7). The model could generally replicate the spatial distribution and time evolution of PM$_{2.5}$ concentrations with some underestimation during the first two days. This bias may underestimate the aerosol impact on rainfall.

The model-simulated precipitation performance is evaluated through comparison with in situ observation and satellite data, as shown in Figure 2. The model output and satellite retrievals are interpolated to in situ observation through bilinear interpolation (Figure 2a–2c). Approximately 100 mm of precipitation accumulates during December 14–16, 2013, covering the entirety of Guangdong Province. However, CMORPH satellite data, which is often used to evaluate model rainfall performance, underestimates the precipitation, particularly near the coast. Previous studies have reported that CMORPH products substantially underestimated heavy rainfall (Jiang et al., 2018; Qin et al., 2014) and cold season rainfall (Xie et al., 2017). The time series of the average rain rate over Guangdong Province reveals a remarkable extreme rainfall event with a lasting rain rate of 2.5 mm h$^{-1}$ on the second and third day; CMORPH data distinctly underestimates rainfall for these days (Figure 2d). The model reproduces a similar magnitude to the observations with an earlier peak in the early morning near 08Z on December 15. The initial time and physics schemes including microphysics, land surface, and PBL are tuned to check whether the peak

time will be different. However, the rainfall changes are mostly happened in amplitude rather than peak time, thus we conclude that the bias may be induced by the meteorology boundary conditions from global model. The Taylor diagram for 3-day accumulated rainfall in Figure 2f suggests that the model simulation yields a higher pattern correlation of 0.50–0.55 and a lower bias of 5%–20% than the CMORPH retrieval does (0.4 and >20% for pattern correlation and bias, respectively). Signs of bias are represented by inverted (negative) or upright (positive) triangles, indicating that the model overestimated the rainfall amount while the satellite products underestimates it. The TRMM data is also used to evaluate this extreme case in Figure S5d. Precipitation is also underestimated along the coast as well as in CMORPH data. Overall, the model replicates the spatial distribution, time evolution, and the intensity of this extreme rainfall event. Note that all the analyses in the following sections are based on simulation results from domain 2.

## 3.2 Effects of ARI versus those of ACI

Aerosols can change cloud properties and precipitation through two processes, radiative and microphysical (Graf, 2004; Kaufman and Koren, 2006), which contribute to the largest uncertainty in human-induced climate changes. We attempt to isolate the effects of ARI and ACI and thus investigate their roles and relative importance in this extreme rainfall event. Figure 3 shows the spatial distribution of the daily accumulated precipitation differences for December 14 and 15 between the different scenarios. Because the results on the third day, December 16, illustrate a similar mechanism to those on December 15, our analysis focus on December 15. The differences between scenarios on December 16 are in the supplementary materials for reference (Figure S3). Distinct effects of aerosols appear during the second day when the rainfall peaked (Figure 3d), although aerosols lead to more cloud droplet number concentration associated with smaller radius on the first day (Figure 5a); this suggests that the effects of aerosols on precipitation are modulated by other factors (e.g. meteorological conditions). On December 15, the domain-averaged precipitation increases by 1.4 mm. A reduction of up to 19.4 mm appears in northern Guangdong province, whereas an increase of up to 33.7 mm occurs in southern Guangdong province, particularly in the region near the Pearl River estuary and land along the coast. The region 22°–24° N and 112°–115° E, denoted by red boxes in Figure 3, is our focus for the following analysis, because it exhibits prominent rainfall increases by 16.7% (+7.8 mm) on average and covers some of the most advanced city clusters in China including Hong Kong, Shenzhen, and Guangzhou. The corresponding precipitation differences induced by ARI and ACI are −1.3 mm (−2.8%) and +9.3 mm (+19.9%), respectively. Positive indicates an increase, and negative indicates a decrease. It is evident that from the pattern of precipitation changes that the net aerosol effects are dominated by ACI during this event. The time series of average precipitation over the red box shows that the model simulations reproduce a rainfall amount comparable to the observation (Figure S4). Compared with the CTL and ARIoff runs, the CLEAN run yields an analogous time evolution, with less rainfall during the peak time from 06Z on December 15 to 10Z on December 16. The next question that arose is how ACI can increase the rainfall amount over the region.

Figure 4a shows the time-height cross section of cloud fraction (shading) and $PM_{2.5}$ concentration (contour) in the CTL run. The cloud fraction is parameterized following Hong et al. (1998), which is calculated as sum of cloud water, cloud ice, and snow. Most cloud fraction concentrates below 8 km in the first day, associating with small amount of rainfall. Deep convection,

with a cloud base at approximately 500 m and cloud top extending to approximately 14 km, appears during December 15–16 when peak rainfall occurs. The PM$_{2.5}$ concentrations in Figure 4a portray a sharp contrast before and after the rainfall peak. After the rainfall peaked at near 07Z in Figure S3, aerosols are washed out dramatically by precipitation. However, before the peak, PM$_{2.5}$ concentrations decrease gradually from 40 µg m$^{-3}$ near the surface to 5 µg m$^{-3}$ near 7 km above ground. With aerosols acting as cloud condensation nuclei, more cloud droplets are formed with smaller radius, particularly before the rainfall peak when aerosol concentration is high (Figure 5a). Smaller cloud droplets evaporate associated with a reduction of cloud water (Figure 6a), resulting in cooling effect and weaker updraft (Figure 5g and 5i). Thus, the cloud fraction decreases before the peak, especially below 2 km. By contrast, there was a prominent cloud fraction band increase near 4 km throughout the peak period with aerosols. The increase of cloud fraction extends to the upper troposphere, near 14 km, corresponding to the increase of ice cloud shown in Figure 5d and 5f. As a result, the deep convection is enhanced associated with more rainfall during peak time. The similarity of cloud fraction changes between Figure 4b and Figure 4c suggests that ACI dominated the total aerosol effect in this event, which is consistent with the previous discussion.

Figure 5a–5c present the aerosol effects on cloud droplet number concentration (CDNC; shading) and cloud effective radius (contour). With aerosols, CDNC increases by 5.5 times accompanied by reduced cloud effective radius near 2 km from 00Z on December 14 to 00Z on December 15, which reduces the efficiency of collision-coalescence between cloud droplets into raindrops (Rosenfeld, 2000; Twomey, 1977). This is characterized by less rain water formed in Figure 6c, indicating suppress of the warm rain. Figure 6a shows more cloud water formed at 2–6 km due to higher supersaturation. The consumption of moisture and energy limits the formation of low cloud below. When droplet nucleate due to activating enormous aerosols, there are abundant latent heat release by enhanced condensation below the 0°C isotherm line. This is also reported in Fan et al. (2018) in which the mechanism responsible for convection intensification is latent heat release from cloud water formation with ultrafine aerosols. This is called "warm-phase invigoration" in their study which is different from "cold-phase invigoration" via suppressing the warm rain. Interestingly, unlike their work, the warm rain is still suppressed before 15Z on December 15 (Figure 6c) even though with strong latent heat release through cloud water formation. The rain water is not increased by accretion of added cloud droplets, suggesting that the precipitation increase is because of enhancement of cold rain. Both cloud ice number concentration and its effective radius are significantly increased between 6Z and 15Z on 15 December. Moreover, the mass and number of ice crystals including cloud ice, snow, and graupel increase drastically during this period. Note the magnitude of snow and graupel mass is ten times of that of rain water. A distinct latent heat release center appears above 0°C isotherm line, which is even stronger than the condensational heat below. These two peaks in aerosol induced diabatic heating are also discussed in Wang et al. (2014) for oceanic deep convection. However, the peaks at 3 km and 7 km are much higher. This may be because the convection occurs over the land. The latent heat from these two peaks thus will intensify convective strength. These findings suggest the cold-cloud process play a dominant role in the precipitation increase before 15Z on 15 December. To further analyze the source of this latent heat release, following Fan et al. (2018), the latent heat released from condensation, sublimation/deposition, and melting/freezing during cold and warm cloud processes are diagnosed (Figure 7).

More information about the latent heat due to different microphysical processes for warm and cold clouds are documented in Appendix A. The rimming processes are included into the freezing. Cold-phase invigoration by aerosols has been shown in both observational (Andreas et al., 2004) and modeling (Khain et al., 2005; Fan et al., 2007) studies. Particularly, much attention is paid to mixed and cold process in which supercooled droplets are likely to freeze and release latent heat, further

enhancing convection (Koren et al., 2008; Rosenfeld et al., 2008; Tao et al., 2007). Interestingly, the latent heat release due to freezing with aerosols is negligible compared with that due to condensation and deposition. The distinct latent heat changes mentioned above in Figure 5g is induced by deposition in cold cloud (Figure 7e). Figure S8 shows the time-height distribution of mass and number concentrations for different hydrometers in control run. Note the magnitude of snow and graupel mass is ten times of that of rain water. There are affluent snow and graupel before 15Z on 15 December located where the distinct

changes in depositional heat appears. With aerosols, the snow and graupel grows at the expenses of ice crystals and rain water via aggregation and rimming, respectively (Figure 6c–e). The former refers to the collision and coalescence of ice crystals to form snow while the latter represents the accretion of cloud drops and rain drops by snow and graupel to form larger graupels. These are the main processes of converting liquid mass to solid phase, contributing to additional precipitating particles. However, the latent heat due to rimming is relatively small (Figure 7f) because the latent heat release per unit for freezing (334

15  kJ kg$^{-1}$) is only 1/8 of that for deposition (2256 kJ kg$^{-1}$). The latent heat release due to deposition in cold cloud is stronger than that due to condensation in warm cloud even though the latter is also important (Figure 6a and f). In deep convection, the strong updraft usually makes the atmospheric condition saturated for water which is supersaturated with respect to ice. With the presence of ice crystals (Figure S8), the formation of ice crystals is enhanced accompanied by additional latent heat release due to deposition (Figure 6 and Figure 7). After 15Z on December 15, most of the snow and graupel sedimentate. Compared

with depositional heating, the condensational heating plays a dominant role in intensifying convective strength. The rain water increases through accretion of added cloud droplets, leading to precipitation increases. These findings highlight two different processes and mechanisms in the precipitation increase before and after 15Z on December 15. The dominant source for latent heat release is depositional heating in the former case (cold rain enhancement) while condensational heating in the latter (warm rain enhancement). Due to latent heat release with aerosols, the vertical motion is boosted (Figure 5g) which further enhance

the supersaturation and associated with latent heat release. Via microphysics–dynamics feedback, the convection is intensified, and precipitation increased. This feedback has been widely discussed in ACI effects on deep convection (Fan et al., 2018; Koren et al., 2015; Tao et al., 2012).

To further delineate the mechanism of this microphysics–dynamics feedback, the moisture budget tool is implemented based on the hourly model output. The atmospheric moisture balance is expressed as follows:

$$\frac{\partial Q}{\partial t} = E - P + MFC \quad (1)$$

where $Q$ is the column-integrated water vapor in the atmosphere, $t$ is time, $E$ is evaporation, $P$ is precipitation, and $MFC$ is the vertically integrated moisture flux convergence.

Evaporation is small in areas of intense precipitation and saturation (Banacos and Schultz, 2005). The column-integrated water vapor changes are small (figure not shown), thus precipitation is balanced by the moisture flux convergence as follows:

$$P \approx MFC \text{ (2)}$$

MFC can be further divided into two terms as

$$-\frac{1}{g} \int_0^{P_s} \nabla \cdot \left( q \overrightarrow{V_h} \right) dp = -\frac{1}{g} \int_0^{P_s} q \nabla \cdot \overrightarrow{V_h} \, dp - \frac{1}{g} \int_0^{P_s} \overrightarrow{V_h} \cdot \nabla q \, dp \quad \text{(3)}$$

where the first term on the right side is the horizontal moisture convergence (hereafter CON); the second term is the horizontal advection of water vapor (hereafter ADV). Thus, the precipitation is balanced by the sum of CON and ADV as

$$P \approx MFC = CON + ADV \quad \text{(4)}$$

The spatial distributions of column-integrated MFC (shading) and moisture flux (vector) between CTL and CLEAN on December 15 are displayed in Figure 8a. The MFC pattern is in good agreement with precipitation differences in Figure 3d, suggesting the validity of the derivation of Equation (2). The average MFC change over the analysis region is +8.1 mm, which is comparable to +7.8 mm in precipitation difference. The vertically integrated moisture flux changes in Figure 8a followed the wind pattern shown in Figure 20. Sd. The moisture flux is enhanced over the analysis region driven by strong convergence, which is consistent with microphysics-dynamics feedback discussed above. These flows converged in the estuary and near the coast with a magnitude of approximately 25 kg m$^{-1}$ s$^{-1}$. The overall pattern of CON is broadly consistent with that of MFC, which indicates that the MFC changes are mainly driven by CON changes (Figure 9a). The ADV changes contribute about 35% of MFC changes over the analysis region but are much more scattered than CON changes (Figure 9c). The pattern of differences between CTL and CLEAN resembles that between ARIoff and CLEAN (Figure 9), which suggest the dominant effect of ACI. The magnitude of changes over the analysis region is smaller in the former case, indicating the compensation effect between ARI and ACI in this case, as noted in section 3.1.

These findings reveal the prominent effects of aerosols on rainfall amount over the estuary and near the coast in this extreme rainfall event. The pattern of precipitation and associated cloud-related variables in CTL minus CLEAN (total effects) bears a resemblance to that in ARIoff minus CLEAN (ACI effects), which allows us to ascertain that ACI dominated the total effects. By applying the moisture budget tool, we confirm the microphysical–dynamic feedback of ACI effects on invigorating convection.

**3.3 Local versus remote aerosol emission effects**

A crucial question is the extent to which increased anthropogenic aerosols from either local (i.e., domain 2, which denotes Guangdong province) or remote (i.e., domain 1, which denotes outside Guangdong province) sources result in precipitation changes. Previous studies have reported different roles of local and remote aerosol sources in affecting tropical precipitation (Chou et al., 2005) and monsoons associated with precipitation (Bollasina et al., 2014; Cowan and Cai, 2011) from a climate perspective. However, the differing effects of local and remote aerosols on weather, such as extreme rainfall, are rarely

explored. In this section, we examine the roles and relative importance of local and remote aerosols in the precipitation increase in the estuary during this extreme rainfall event.

The differences in time-height cross section of cloud fraction (shading) and $PM_{2.5}$ concentration (contour) induced by the effects of local and remote emissions are shown in Figure 10a and 10b, respectively. With local emissions, the aerosol concentrations mainly increase within the PBL below 2 km before 12Z on December 15 (Figure 10b). The accumulated aerosols are washed out quickly after the rainfall initiated. By contrast, with remote emissions, a higher aerosol concentration extends to approximately 8 km after 3Z on December 14 (Figure 10a). Two peaks near 0.5 km and 5 km above ground are centered near 10Z and 18Z on December 14, respectively, indicating a strong transportation of aerosols. The earlier peak, near 5 km, is caused by strong wind speed in the free atmosphere compared with that within the PBL. Moreover, the aerosol concentrations last longer before decreasing dramatically until the peak rainfall starts at 07Z on December 15, because aerosols are transported continuously from the remote area. The cloud fraction reduction is coherent with aerosol concentration peaks, indicating that increased aerosols lead small cloud droplets to evaporate. Moreover, more deep cloud formation consuming moisture and energy. Comparing patterns of cloud fraction changes between Figure 10a and 10b and Figure 4b indicate the dominant effects of aerosols from remote areas. The CDNC (shading) increases in both D1 and D2 runs compared with the CLEAN run before the rainfall peak (Figure 11a and 11b). However, the discernible cloud effective radius (contours) decrease appears only in the D1 run and is attributed to a stronger CDNC increase. Correspondingly, the CINC and ice cloud effective radius show more remarkable increases in the D1 run during the rainfall peak time (Figure 11c and 11d). The associated latent heat and vertical velocity are much stronger in the D1 run compared with the D2 run (Figure 11e and 11f). Interestingly, most of latent heat release with local emissions are happened below 0°C isotherm line. Figure 12 shows the changes in mass and number of different hydrometeors with remote aerosols emissions. There are plenty of snow and graupel formation at the expense of rain water when precipitation increase before 15Z on 15 December, indicating intensified cold rain process. The corresponding latent heat release is dominated by deposition in cold cloud. By contrast, after 15Z on 15 December 15, rain water increases significantly during precipitation enhancement, representing stronger warm rain process. The associated latent heat release is due to condensational heating in warm cloud concentrated below 0°C isotherm line. The patterns of changes in hydrometeors and latent heat in D1 assembles that in CTL run, which further confirm the dominant role of remote aerosols emissions. The distribution of time-height changes in hydrometeors and latent heat between D2 and CLEAN run are shown in Figure S9 and Figure S10, respectively. As aerosols from local emissions are concentrated near the surface and are washed out dramatically once the rain initiated, much less cloud water formed with larger size than that in D1 run. More rain water is formed by accretion of cloud droplets which indicate that intensified warm rain is the only reason for precipitation increase with local aerosol emissions. As a result, the average precipitation increase over the analysis region on December 15 is 7.3 mm with remote aerosol emissions, much greater than that with local aerosol emissions (3.1 mm, Figure 14c and 14d). These findings suggest that both the effects of local and, to a much greater extent, remote aerosol emissions contribute to precipitation increases over the analysis region.

### 3.4 Tenfold anthropogenic emissions and chemical ICs and BCs

An optimal aerosol loading should exist in which the convection is the most vigorous (Rosenfeld et al., 2008). For aerosol concentrations below the optimum, the convection is invigorated by smaller droplets; thus, stronger updraft releases larger latent release (Dagan et al., 2015b). By contrast, suppression effects dominate above the optimum (Small et al., 2009). The optimum value is determined by environmental conditions (e.g., relative humidity, see Dagan et al., 2015a). In this section, a tenfold aerosol emission simulation (10×) is conducted to examine the sensitivity of precipitation and associated cloud properties to aerosol concentrations.

The $PM_{2.5}$ concentrations (contours) in 10× increased significantly to approximately ten times that in CTL, indicating a linear relationship from emissions to aerosol concentration (Figure 15). The associated boundary layer cloud formation (shading) is further suppressed below 2 km, which is consistent with the result in Figure 4b. The change patterns in cloud fraction and aerosol concentration in Figure 15. D are similar to that in Figure 4b, but Figure 15 shows a much greater magnitude. The CDNC (shading) increase and cloud effective radius (contours) reduction in Figure 16a are also more pronounced than those in Figure 5a. CDNC noticeably decreases below 1.5 km but increases substantially from 1.5 km to 4 km before 04Z on December 14, associating with smaller radius. Smaller cloud droplet tends to evaporate. In addition, more cloud droplets are produced due to higher supersaturation upward. The consumption of water and energy leads to further reduction of low cloud (Figure 17a). The involved latent heat and vertical velocity during the rainfall peak time from 8Z on December 15 to 10Z on December 16 in Figure 16c exhibit a stronger increase associated with a higher altitude above the freezing level than that in Figure 5c. Besides, a distinct weaker latent heat release associated with negative vertical velocity anomaly appear below freezing level between 10Z and 22Z on 15 December. This indicate a more important role of cold related processes in latent heat release. The ice crystals also increase drastically with bigger radius. Figure 17 shows the changes in mass and number concentrations of different hydrometeors in 10× simulation. Compared with CTL run, the snow and graupel are also increased with a stronger magnitude, particularly before 15Z on 15 December, indicating enhanced cold rain. However, rain water show decrease during all the time instead of increase after 15Z when precipitation increase. This means the warm rain is suppressed much stronger in 10× simulation. As with ten times of aerosols emissions, the aerosols lower the supersaturation much stronger by activation to form much smaller cloud droplets. The rain water evaporates rather than increase by accretion of additional cloud droplet, associating with strong condensational cooling in warm cloud (Figure 18a). Precipitation on December 15 is suppressed over the upstream region up to 39.6 mm in the northwest of Guangdong province but substantially enhanced up to 59.7 mm over the downstream region near the coastal region (Figure 19b). A similar finding is reported by Zhong et al. (2015). The delay of early rain in the upstream area results in more rainfall and stronger rain intensity within the downstream area and a more narrowed region compared with the red box in Figure 3b. The average precipitation in Guangdong province on December 15 decrease by 1.0 mm in 10×, whereas it increases by 1.4 mm in CTL. Tenfold aerosol emissions produce a more polluted environment, with $PM_{2.5}$ concentrations of approximately 300 µg m$^{-3}$. Although abundant moisture is transported from the Bay of Bengal and the South China Sea (Figure 1b), the aerosol loading may still have surpassed the optimal value

for cloud invigoration and thus suppressed precipitation over Guangdong province. Moreover, aside from suppressing the rainfall amount, excessive aerosols also have the potential to redistribute precipitation and increase its range in spatial distribution.

## 4 Summary and discussion

In this study, we find that aerosols significantly affect local extreme weather (i.e., torrential rainfall), invigorating deep convection, via ACI effects. Deep convection invigoration by aerosols has been discussed in both observation (Andreae et al., 2004; Koren et al., 2004) and model simulations (Khain et al., 2005; Storer et al., 2013). Most of these studies are focused on mixed and cold processes. Increasing aerosols can suppress warm rain because of smaller cloud droplets. These smaller cloud droplets are likely lifted upward to freeze. The latent heat due to freezing will further enhance convection (Rosenfeld et al., 2008). This is referred to as cold-phase invigoration. A recent interesting study conducted by Fan et al. (2018) found that additional nucleation of cloud droplet can release abundant condensational heat below freezing level. More cloud water will form via condensation on the additional cloud droplets. This process will increase both warm rain and supercooled cloud water. Furthermore, the ice-related processes are enhanced with latent heat release, further intensifying the convection. In their study, the source of latent heat is dominated by condensational heating, accompanied by enhanced warm rain. In contrast to cold-phase invigoration, the concept of warm-phased invigoration is proposed in their work.

With aerosols, the precipitation is increased between 03Z on December 15 to 10Z on December 16. CDNC increases remarkably, reducing the size of cloud droplets. Additional cloud water formed with intensified condensational heating, leading to enhanced convection and increased precipitation. However, rain water decreased substantially before 15Z on 15 December, indicating suppressed warm rain, which is different to Fan et al. (2018). The source of enhanced latent heat release is dominated by deposition in cold cloud associate with increase of snow and graupel, representing cold rain enhancement. There are abundant ice crystals including snow and graupel at 4–6 km from 00Z to 15Z on 15 December. The strong upward motion makes the atmospheric condition saturated for water which is supersaturated with respect to ice. As a result, the mass and number of snow and graupel increase drastically via aggregation and rimming, respectively, at the expense of rain water, suggesting a dominant role of cold rain before 15Z on 15 December. Most of snow and graupel fall as precipitation when the peak rainfall occurs after 15Z. By contrast, the warm rain is enhanced characterized by increase of rain water associating with condensational heating in warm cloud via accretion of cloud droplet, which is consistent with Fan et al. (2018). The enhanced latent heat boosts the vertical motion, leading to higher supersaturation accompanied by stronger latent heat release. This feedback between microphysical and dynamic processes results in more rainfall (Tao et al., 2007) up to 33.7 mm in our simulation. On average, ACI enhance precipitation over the analysis region. Conversely, ARI partially compensate for the precipitation increase by 14%. The analysis of the moisture budget suggests that the precipitation increase is manifested by strengthening the column-integrated MFC. Further decomposition of MFC suggest the importance of horizontal moisture

convergence. Our finding confirms that microphysical–dynamic feedback is at the core of the effects of ACI on convective invigoration.

An interesting question is why the precipitation increases induced by ACI appear on land near the Pearl River estuary and the coast. Khain et al. (2008) found that aerosols generally suppress cloud formation in relatively dry conditions, whereas they invigorate convection in moist environments. Fan et al. (2009) suggested that wind shear may take a dominant role in regulating the effects of aerosol on deep convection. Increased aerosols suppress (invigorate) convection under strong (weak) wind shear. These findings highlight the crucial roles of humidity and wind shear in modulating the convective invigoration effects induced by aerosols. Strong wind shear enhances the entrainment of dry air into clouds and transports cloud liquid to unsaturated regions; this leads to greater evaporation and sublimation of cloud particles. These processes are associated with cooling, downdrafts, and convergence, especially at high aerosol concentrations (Khain, 2009; Lee et al., 2008). The convergence thus fosters secondary cloud formation and contributes to an increase in precipitation. However, Fan et al. (2009) stressed that the net latent heat release, as an energy source for convection, is greater under weak wind shear than under strong wind shear. Aerosols enhance convection under weak wind shear until an optimal aerosol concentration is reached at which the net latent heat release equilibrates. This mechanism may only be applicable to isolated storms rather than to cloud systems. Note that the previous studies have used different wind components (zonal component, meridional component, or total wind) at different heights with different thresholds (e.g., upper limits vary from 10 m s$^{-1}$ to 20 m s$^{-1}$). These different standards may only suitable for specific environmental conditions, because previous studies have been based on limited cases. In our work, the wind shear is estimated as the difference between the maximum and minimum total wind speeds at 0–10 km. The spatial distribution of wind shear (first row) and column-integrated water vapor (second row) are presented in Figure 20. The wind shear increases with the southeast–northwest tilt ranging from 35 m s$^{-1}$ to 80 m s$^{-1}$ (Figure 20a and 20b). Our definition of wind shear is different from other studies (e.g., Lee et al. 2008; Fan et al. 2009; Li et al. 2011; Guo et al., 2016a), with a higher altitude. We chose 10 km because the latent heat release, which is a key factor determining convection intensity and partly depends on wind shear, extends up to approximately 10 km (Figure 5g). Although the wind shear in our work is stronger than that in other studies with magnitudes lower than 10 m s$^{-1}$, the aerosol-induced convective invigoration effect appears over the region with relatively weak wind shear and high humidity on the land along the coast, as presented in Figure 20. This invigoration effect under weak wind shear for cloud systems is described in a recent work (Li et al., 2011), whereas it is to some extent contradicted by the results of Lee et al. (2008). Conversely, precipitation was suppressed to the northwest of Guangdong, with relatively strong wind shear and low humidity, as shown in Figure 20b and 20d. The gradients of wind shear and humidity increase between the southeast and northwest of domain 2 on December 15 when peak rainfall occurred. The results confirm that the effect of aerosols on precipitation is related to relative humidity and wind shear. However, this relationship remains dependent on the situation and may be affected by other meteorological variables, such as convective available potential energy (Khain et al., 2005), cloud phase (Lin et al., 2006), and cloud type (Koren et al., 2008). The relative importance of different meteorological variables in regulating the aerosol-induced precipitation effect requires both long-term observation and model sensitivity tests to provide a more comprehensive picture.

Aerosol emissions were separated into those from Guangdong Province and those from elsewhere, named experiments D2 and D1, respectively, to represent the effects of aerosol concentrations from local and remote emissions on this extreme rainfall event. The surface aerosol concentrations accumulate slowly from local emissions if the rainfall system comes with strong northerlies. Instead, aerosols, transport from remote areas persistently, extend to higher altitudes, up to 8 km. The aerosol concentrations thus are maintained at a relatively high level in the D1 and invigorated convection. The resemblance of changes in different hydrometeors and latent heat between D1 and CTL further suggest the dominant role of remote aerosols in the convective invigoration. Interestingly, with local emissions, the precipitation enhancement is mainly through intensified warm rain only. This is because much less aerosols stay in the atmosphere with only local aerosols emissions once the rainfall initiated. The effect of nucleated cloud droplets on reducing supersaturation and size of droplets is much weaker than that with remote aerosol emissions. Thus, the rain water is increased by accretion of cloud droplets, enhancing the warm rain. The precipitation averaged over the analysis region on December 15 increased by 7.3 mm from the effects of remote aerosol emissions but only 3.1 mm from local aerosol emissions. These results suggest that the effects of remote aerosol emissions play a dominant role in the intensification of precipitation in the estuary, which implies the potential influence of remote aerosol emissions on extreme synoptic weather events. However, this crucial issue remains insufficiently explored.

Previous studies have suggested an optimal aerosol loading in which condensational heating and evaporative cooling are balanced, leading to the most vigorous convection (Fan et al., 2007, 2009; Rosenfeld et al., 2008; Wang, 2005). A tenfold emission experiment showed a similar pattern with CTL but with a much stronger signal. Further analysis of hydrometeors and latent heat reveal that the main reason for precipitation increase is via intensified cold rain. The warm rain is suppressed almost all the time because the saturation for water is supersaturation with respect to ice. With the presence of abundant snow and graupel, these precipitating particles grow by aggregation and rimming at the expense of rain water. Instead, the increase of rain water by accretion of droplets is suppressed. Excessive aerosols lead to more precipitation increases, up to 59.7 mm, which is much larger than the 33.7 mm from CTL. However, the precipitation increase is limited to a more narrowed region along the coast in the downwind area; this may be related to the adequate supply of water vapor from onshore wind, as shown in Figure 20d. The average precipitation over Guangdong province decreased by 1.0 mm in 10× but increased by 1.4 mm in CTL. These results indicate that aerosol concentrations in 10× exceed the optimal aerosol loading for convective invigoration and instead suppress the rainfall amount. The retribution for spatial distribution of precipitation with a sharper contrast implies that air pollution may increase the possibility of both flood and drought.

The mechanism of precipitation decreases over another region, in 24°–25°N, 110°–112°E, is also investigated. Figure S11 shows the distribution of time-height mass and number concentrations of different hydrometeors averaged over this region from CTL run. There are lots of ice crystals with cloud ice extending up to 16 km, indicating strong deep convection, which is consistent with low cloud top temperature in Figure S1b. However, the cloud base is higher than that over the region denoted by the red box, characterized by smaller low-level cloud water on December 15 when strong aerosol impact occurs. This can also be indicated from the surface temperature field with lower values in the top left corner of domain 2 (Figure S16). With aerosols, more cloud droplets nucleated on which water can condensate. Additional cloud water is subsequently formed near

to 4 km (Figure S12a), accompanied by reduced supersaturation. The reduction of rain water and ice crystals (particularly in graupel) suggest that both the warm rain and cold rain are suppressed. Less latent heat is released dominated by condensation in warm cloud and deposition in cold cloud. There could be three reasons for this. The first one is that the mass of water vapor is small over this region in the northwest corner of the domain (Figure 20b), so that not enough water supply for convective invigoration effect with aerosols. The second one is related to the very strong wind shear over this region with maximum value up to 80 m s$^{-1}$. This condition is unfavored for latent heat to accumulate, which is key factor to convection strength (Fan et al., 2009). In addition, the cold cloud bases may suppress convection and precipitation due to strong evaporative cooling and less efficient ice crystals formation (Fan et al., 2016). Thus, the precipitation is suppressed over this region with aerosols. With ten times of aerosol emissions, the mass and number of rain water and ice crystals are further reduced, accompanied by weaker latent heat release (Figure S14 and S15). As a result, the precipitation is further suppressed (Figure 19b).

One may wonder whether the precipitation differences over domain 2 in D1 experiment is driven by meteorological fields changes or by transport of aerosols because the scaling of emissions in domain 1 also modify the local atmospheric conditions. The changes in meteorology in turn may affect the precipitation in domain 2. Figure S17 shows the aerosol effects on 2-m temperature and column water vapor in domain 1. With aerosols, the moisture change is small over the whole China. The surface temperature decreases up to about 1 K is seen over northeastern China, Sichuan, and northeastern Indo-China Peninsula through absorbing and scattering solar radiation as well as serving cloud condensation nuclei. The temperature over Guangdong province show marginal changes as the aerosol concentration is concentrated to the north of Guangdong and incident solar radiation is weak in rainy days. The relatively small changes in meteorological fields over domain 2 may indicate a dominant role of transboundary aerosols. Figure S18 shows the precipitation differences over domain 2 on 15 December based on domain 1 output. The pattern of precipitation changes is very different from that calculated based on domain 2 output, suggesting that the atmospheric condition changes in domain 1 cannot account for the precipitation differences in Figure 3d. Moreover, the importance of aerosol-cloud interactions discussed above works for both D1 and D2 experiment which may further confirm the precipitation changes in Guangdong is driven by transboundary aerosols rather than changes in meteorology in domain 1. Note the cumulus scheme is used in domain 1 but not in domain 2 which may result in different response of precipitation to atmospheric changes in domain 1. To completely disentangle the meteorology impact from that of transboundary aerosols, the possible solution could be application of nudging to constrain the meteorology as same as CTL and scale the emissions in domain 1. This could be in future sensitivity studies.

We note that uncertainties exist in aerosol emission and the representation of ACI. Although ice nucleation may have little effect on the spatial distribution and temporal evolution of surface precipitation (Deng et al., 2018), this factor is not yet considered in the WRF-Chem model. This may explain negligible differences in results between simulations with and without dust and sea salt emissions. Additionally, dust sources are far from our analysis region and the prevailing wind is northerly; these produce low dust and sea salt concentrations, respectively. It is noteworthy that we assume the ARI and ACI effects are linear additive as previous studies (Fan et al., 2015; Zhong et al., 2015), so that the ACI effect is derived by subtracting ARI from total aerosol effects. We cannot check the nonlinearity between ARI and ACI effects because it is not easy to turn off

ACI effect. The problem is how to set the background concentration of cloud droplet number while keep the ARI as same in control run. This means that we could only prescribe the cloud droplet number concentration rather than adjust the emission or aerosol concentration. However, the ACI effect is very sensitive to the number we set (Gustafson et al., 2007).

Although our findings are limited to a case study, this case is, nevertheless, representative of the remarkable aerosol effect on an extreme rainfall event through ACI. This finding provides more evidence of the importance of considering aerosols in extreme weather (i.e., torrential rainfall) forecasting. More interestingly, aerosols from remote emission sources exhibited the potential to modify extreme weather through transboundary air pollution. It hints that we need be careful about the spatial scale when looking at the effect of aerosols on extreme weather event. This case clearly demonstrates the complicated feedback between the dynamic and microphysical processes induced by aerosols. Aerosols substantially redistributed the rainfall amount, a finding with crucial implications for the availability and usability of water resources in different regions of the world (Li et al., 2011). High aerosol concentrations may intensify both flood and drought by invigorating convection.

## Appendix A: Technical details of calculating latent heat

The latent heat release due to phase change for warm cloud and cold cloud are calculated separately in Morrison microphysical scheme. Warm (cold) cloud refers to the cloud at vertical layer with temperature above (below) 0°C.

For warm cloud:

$$\text{T3D\_Wcon(K)} = (\text{PRE(K)} + \text{PCC(K)}) * \text{XXLV(K)}/\text{CPM(K)} \quad (1)$$

$$\text{T3D\_Wdep(K)} = (\text{EVPMS(K)} + \text{EVPMG(K)} * \text{XXLS(K)})/\text{CPM(K)} \quad (2)$$

$$\text{T3D\_Wfrz(K)} = (\text{PSMLT(K)} + \text{PGMLT(K)} - \text{PRACS(K)} - \text{PRACG(K)}) * \text{XLF(K)}/\text{CPM(K)} \quad (3)$$

Where the left terms refer to latent heat release due to condensation, sublimation, and melting in Equation (1), (2), and (3), respectively. K is the layer in vertical for loop. The first term in the bracket on the right side represent different microphysical processes contributing the latent heat release. Based on Mao et al., (2018), more information on the warm-cloud transfer processes between different hydrometers for each process is described in Table 2. The terms of XXLV, XXLS, and XLF denote the latent heat release per unit of condensation, deposition, and freezing, respectively. CPM is specific heat at constant pressure for moist air.

Similarly, for cold clouds,

$$\text{T3D\_Ccon(K)} = (\text{PRE(K)} + \text{PCC(K)}) * \text{XXLV(K)}/\text{CPM(K)} \quad (4)$$

$$\text{T3D\_Cdep(K)} = (\text{PRD(K)} + \text{PRDS(K)} + \text{MNUCCD(K)} + \text{EPRD(K)} + \text{EPRDS(K)} + \text{PRDG(K)} + \text{EPRDG(K)})$$
$$* \text{XXLS(K)}/\text{CPM(K)} \quad (5)$$

$$\text{T3D\_Cfrz(K)} = (\text{PSACWS(K)} + \text{PSACWI(K)} + \text{MNUCCC(K)} + \text{MNUCCR(K)} + \text{QMULTS(K)} + \text{QMULTG(K)}$$
$$+ \text{QMULTR(K)} + \text{QMULTRG(K)} + \text{PRACS(K)} + \text{PSACWG(K)} + \text{PRACG(K)} + \text{PGSACW(K)}$$
$$+ \text{PGRACS(K)} + \text{PIACR(K)} + \text{PIACRS(K)}) * \text{XLF(K)}/\text{CPM(K)} \quad (6)$$

The information on the cold-cloud transfer processes between different hydrometers for each process is described in Table 3.

*Acknowledgment.* We thank the two anonymous reviewers whose insightful comments lead to a significant improvement of the manuscript. This work was jointly supported by the Early Career Scheme of Research Grants Council of Hong Kong (grant CUHK24301415), the National Key Basic Research Program of China (grant 2015CB954103), the Ministry of Science and Technology of China (grant 2017YFC1501401), the Improvement on Competitiveness in Hiring New Faculties Fund (2013/2014) of The Chinese University of Hong Kong (CUHK, grant 4930059), and the Vice-Chancellor's Discretionary Fund of the Chinese University of Hong Kong (grant 4930744). The appointment of NCL at the CUHK is supported by the AXA Research Fund. We would also like to acknowledge Met Office Climate Science for Service Partnership China as part of the Newton Fund for supporting Zhen Liu and Massimo Bollasina on the paper revision effort. Model simulations were conducted on computer clusters at Geophysical Fluid Dynamics Laboratory (GFDL). Larry Horowitz and Maofeng Liu provided helpful comments. We acknowledge the Global Scholarship Program for Research Excellence 2017–18 at the CUHK for supporting Zhen Liu's exchange visit to Princeton University. Chun Zhao is supported by the Thousand Talents Plan for Young Professionals, the Fundamental Research Funds for the Central Universities, and the National Natural Science Foundation of China (grant 41775146).

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

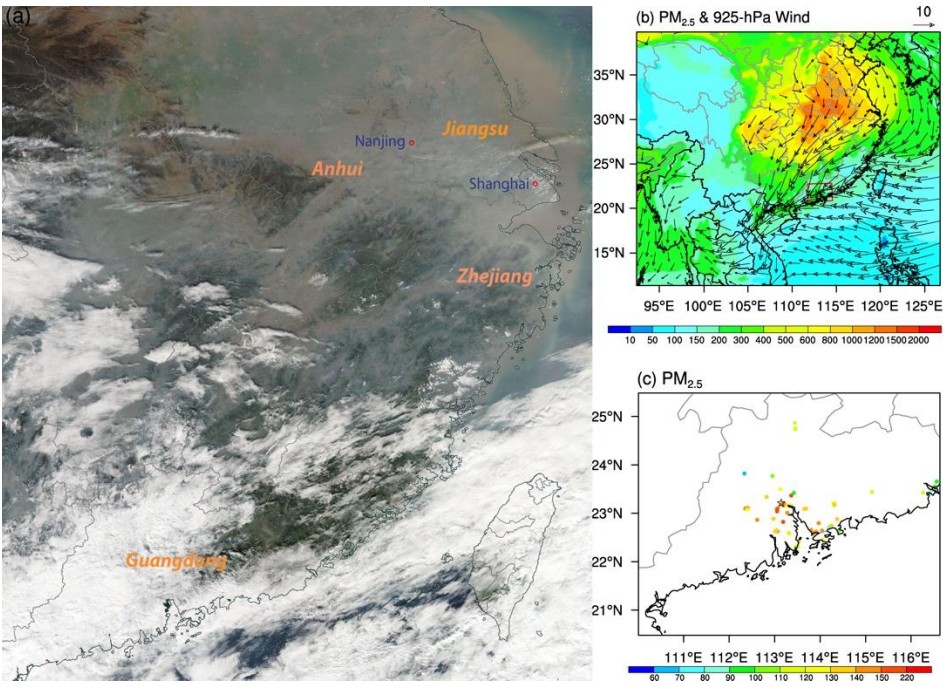

**Figure 1. (a) Terra satellite true-color image of east China on December 13, 2013 (UTC), provided by NASA's Worldview (source:** **https://worldview.earthdata.nasa.gov/****). Red circles denote city locations, blue fonts denote cities, and orange fonts in bold italic denote provinces. (b) Spatial distribution of 3-day averaged column-integrated PM$_{2.5}$ concentrations (shading; unit: µg m$^{-2}$) and 925-hPa wind (vector; unit: m s$^{-1}$) during December 14–16, 2013, in control run. The red box denotes the analysis region. (c) Hourly-averaged PM$_{2.5}$ (unit: µg m$^{-3}$) concentration on December 13, 2013, observed in Guangdong province. Colored circles denote in situ station locations, and black star denotes Guangzhou.**

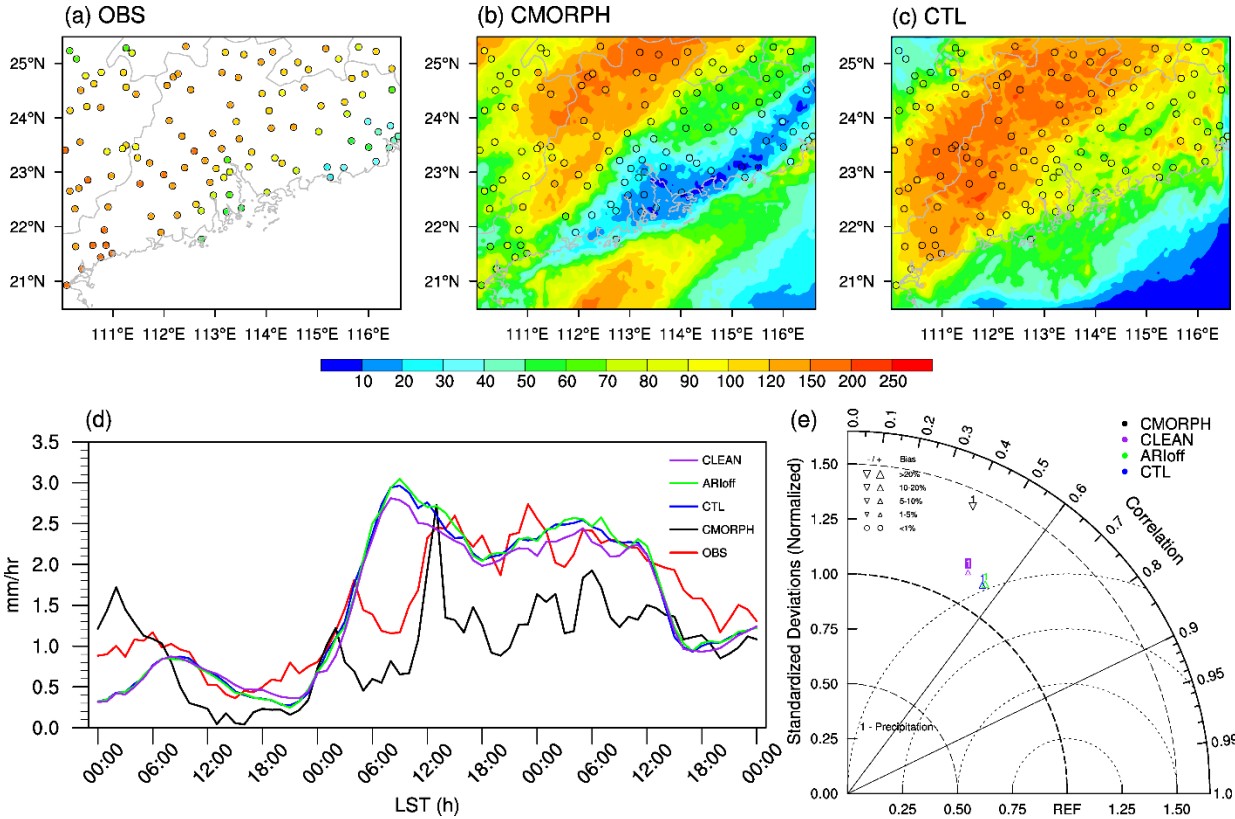

**Figure 2. Spatial distribution of accumulated precipitation (unit: mm) from 00Z on December 14, 2013, to 00Z on December 17, 2013 (local standard time [LST]) from (a) station observations (OBS), (b) CMORPH satellite, (c) control simulation (CTL). Circles denote locations of in situ observations. (d) Time series of station average of rain rate (unit: mm h⁻¹) over the entire domain 2 for OBS (red), CMORPH (black), and CTL (blue). (e) Taylor diagrams for 3-day accumulated precipitation in CTL (blue) and CMORPH (black) compared with OBS. Triangles and circles at top-left corner in (e) denote bias. Sizes of triangles indicate magnitude of bias. Inverted (upright) triangles represent a negative (positive) bias.**

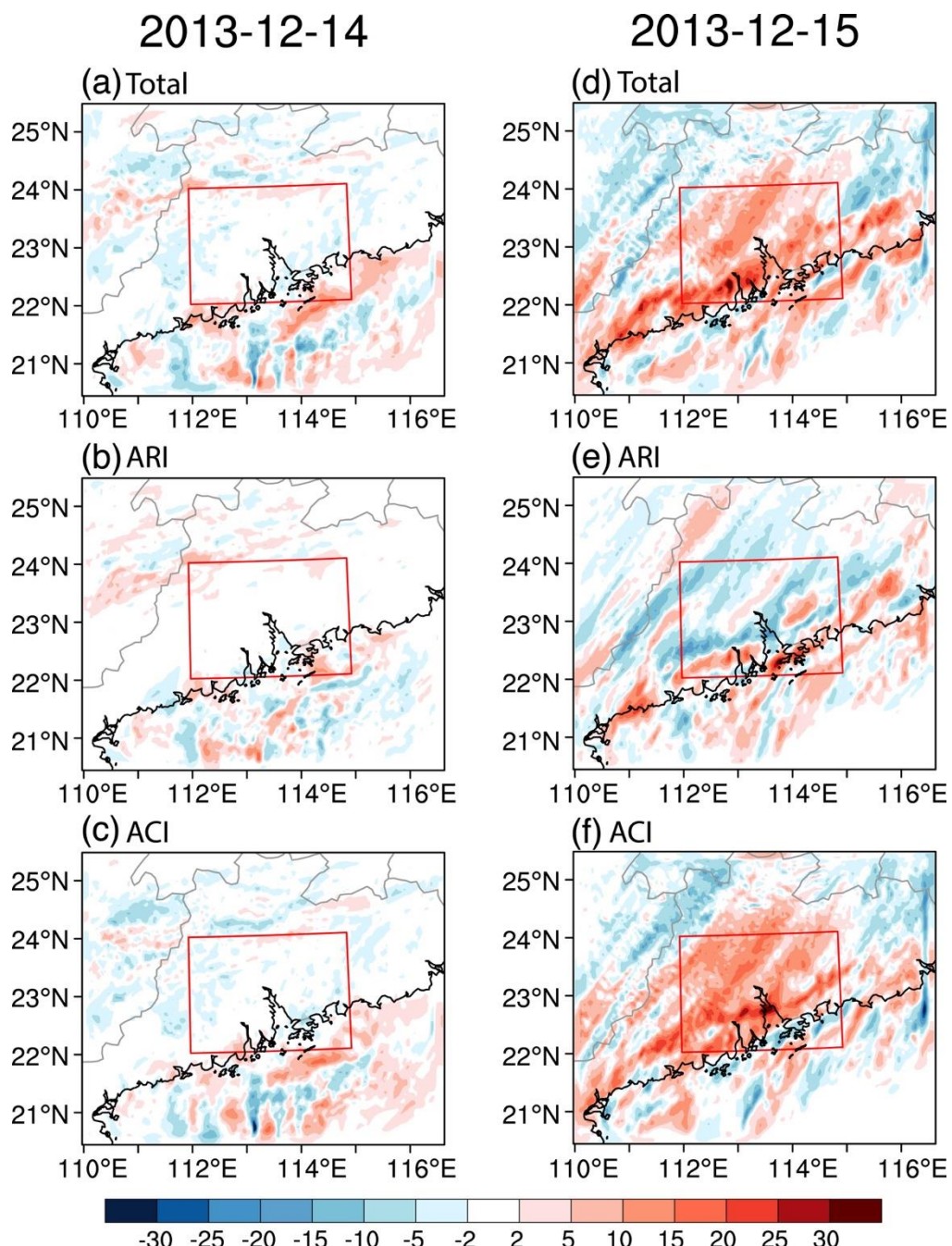

**Figure 3.** Differences in precipitation (unit: mm) (a) between CTL and CLEAN (i.e., CTL minus CLEAN), (b) CTL and ARIoff (i.e., CTL minus ARIoff), and (c) ARIoff and CLEAN (i.e., ARIoff minus CLEAN; third row) on December 14. (c–f) Same as (a–c) but for December 15. Red boxes (22°–24° N, 112°–115° E) denote the analysis region. ARIoff run refers to simulation with aerosol-radiation interactions off.

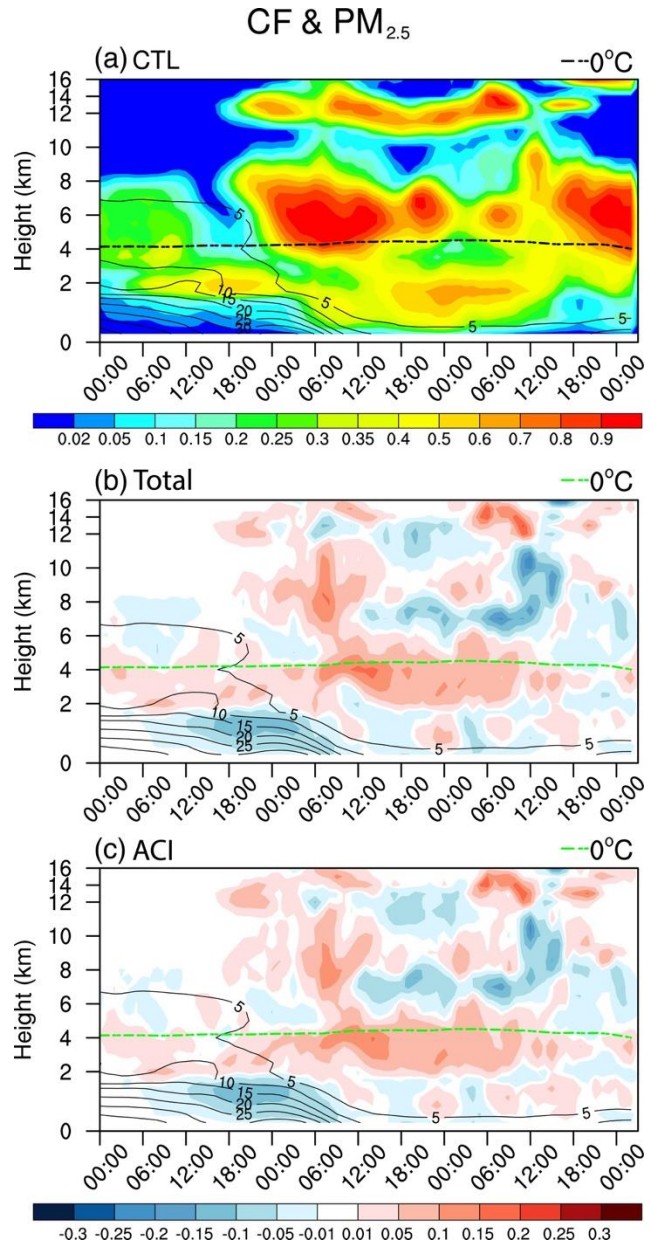

**Figure 4. (a) Time-height cross section of cloud fraction (CF; shading; unit: unitless) and PM$_{2.5}$ concentrations (contour; unit: μg m$^{-3}$) averaged over the red box shown in Figure 3 in CTL run. Differences in the time-height cross section of CF (shading; unit: unitless) and PM$_{2.5}$ concentration (contour; unit: μg m$^{-3}$) averaged over the red box shown in Figure 3 between (b) CTL and CLEAN (i.e., CTL minus CLEAN) and (c) ARIoff and CLEAN (i.e., ARIoff minus CLEAN). The cloud fraction is calculated as sum of cloud water, cloud ice and snow. Dashed lines denote 0°C isotherm calculated as the averaged zero-layer height over the red box in Figure 3.**

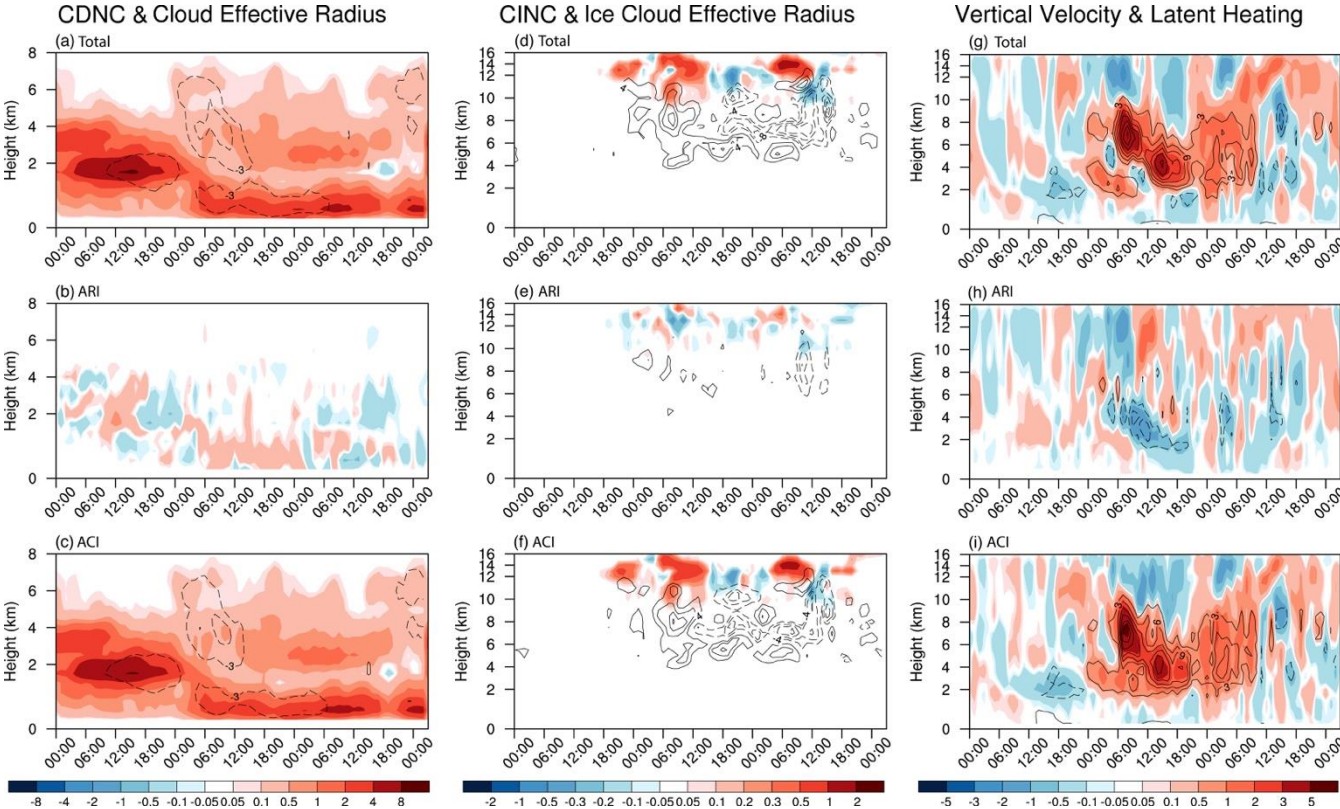

**Figure 5.** Differences with time (abscissa; from 00Z on December 14 to 02Z on December 17) and height (ordinate) in (a) cloud droplet number concentration (CDNC, shading; unit: $10^7$ kg$^{-1}$) and cloud effective radius (contour; unit: μm), (d) cloud ice number concentration (CINC, shading; unit: $10^5$ kg$^{-1}$) and ice cloud effective radius (contour; unit: μm), and (g) vertical velocity (shading; unit: cm s$^{-1}$) and latent heating (contour; unit: K d$^{-1}$) averaged over the red box shown in Figure 3 between CTL and CLEAN (i.e., CTL minus CLEAN; first row). (b, e, h) Same as (a, d, g) but for differences between CTL and ARIoff (i.e., CTL minus ARIoff; second row). (c, f, i) Same as (a, d, g) but for differences between ARIoff and CLEAN (i.e., ARIoff minus CLEAN; third row). For CINC and ice cloud effective radius, only cloud ice is considered. Zero-value contour lines are omitted, and negative values are dashed.

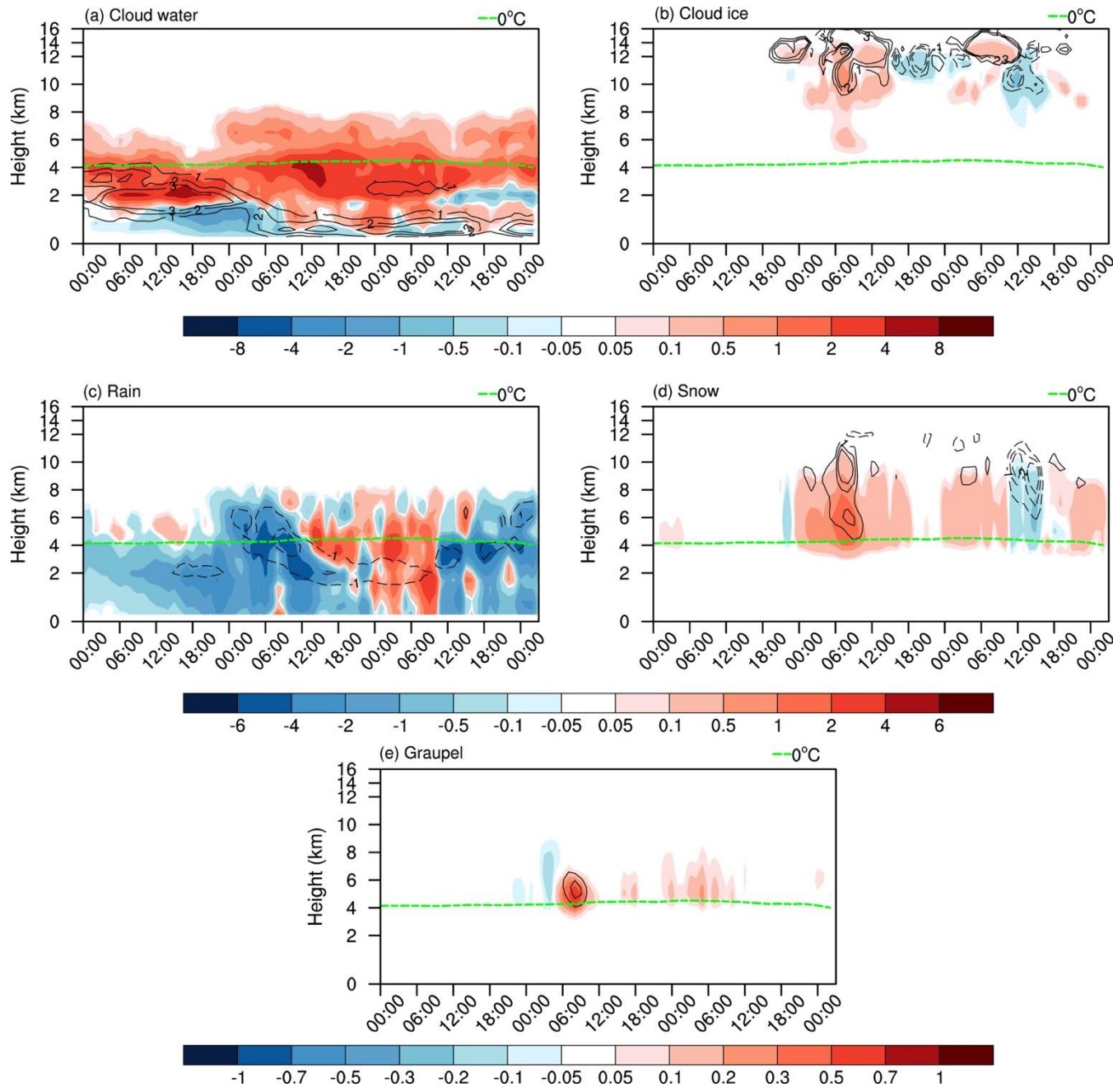

**Figure 6. Differences with time (abscissa) and height (ordinate) in (a) cloud water (shading; unit: $10^{-5}$ kg kg$^{-1}$) and CDNC (contour; unit: $10^7$ kg$^{-1}$), (b) cloud ice (shading; unit: $10^{-5}$ kg kg$^{-1}$) and CINC (contour; unit: $10^4$ kg$^{-1}$), (c) rain (shading; unit: $10^{-5}$ kg kg$^{-1}$) and rain number concentration (contour; unit: $10^5$ kg$^{-1}$), (d) snow (shading; unit: $10^{-4}$ kg kg$^{-1}$) and snow number concentrations (contour; unit: $10^3$ kg$^{-1}$), and (e) graupel (shading; unit: $10^{-4}$ kg kg$^{-1}$) and graupel number concentration (contour; unit: $10^3$ kg$^{-1}$) between CTL and CLEAN (i.e. CTL minus CLEAN) averaged over the red box.**

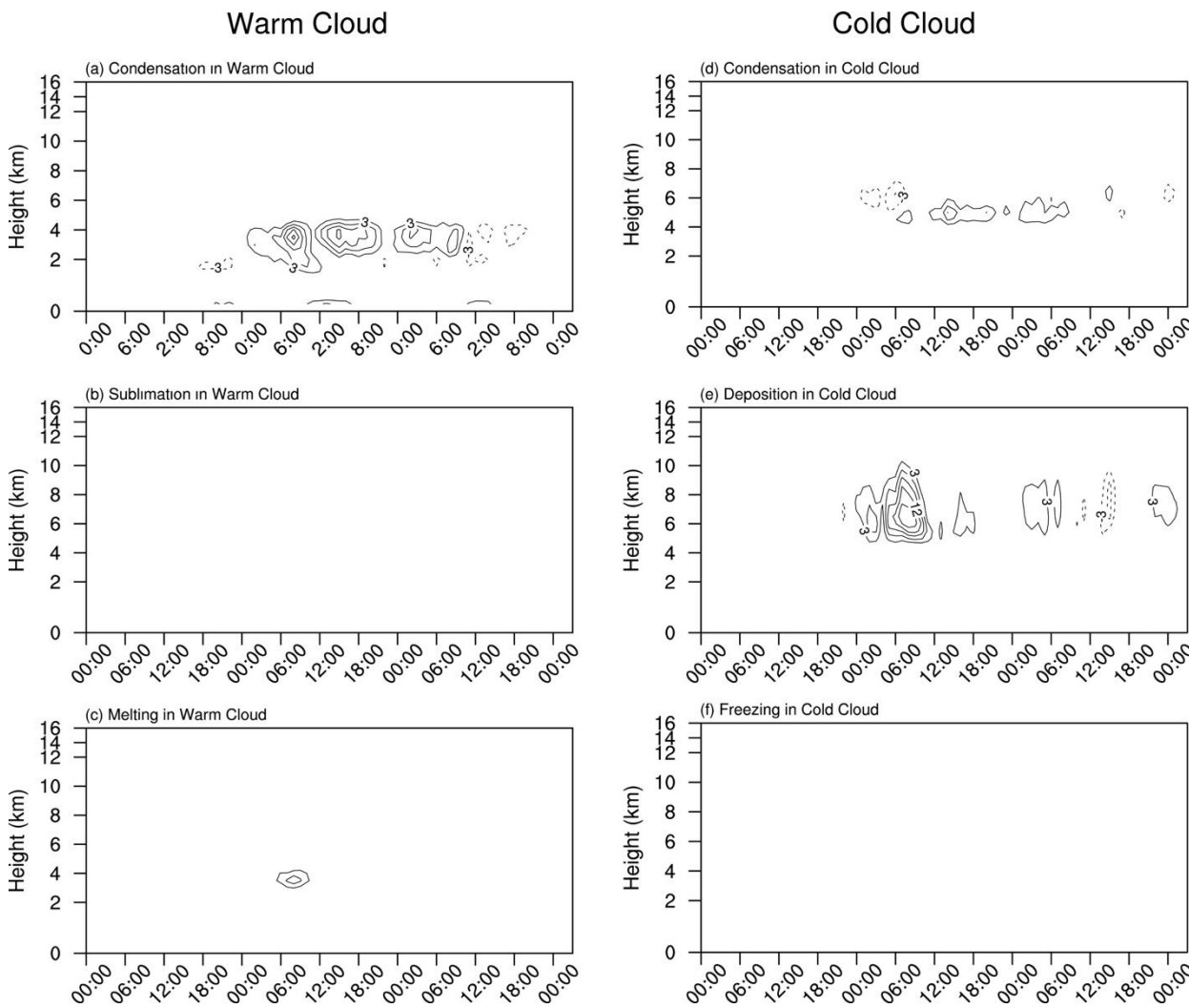

**Figure 7. Differences with time (abscissa) and height (ordinate) in latent heat release (unit: K d⁻¹) from (a) condensation, (b) deposition, and (c) freezing processes between CTL and CLEAN (i.e. CTL minus CLEAN) averaged over the red box for the warm cloud. (d–f) Same as (a–c) but from cold cloud. Zero-value contour lines are omitted, and negative values are dashed. The contour interval is 3 K d⁻¹.**

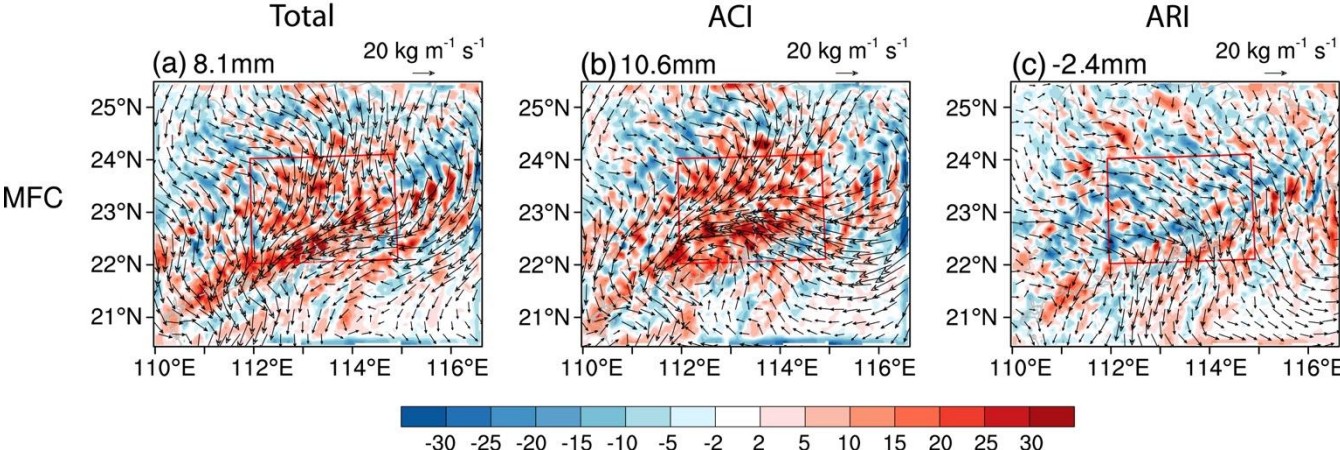

**Figure 8.** Differences in column-integrated flux convergence (MFC; shading; unit: mm) and moisture flux (vector; unit: kg m$^{-1}$ s$^{-1}$) between (a) CTL and CLEAN (i.e., CTL minus CLEAN), (b) ARIoff and CLEAN (i.e., ARIoff minus CLEAN), and (c) CTL and ARIoff (i.e. CTL minus ARIoff) on December 15. Numbers at top-left corner of each panel represent values averaged over red boxes. Red boxes (22°–24° N, 112°–115° E) denote the analysis region.

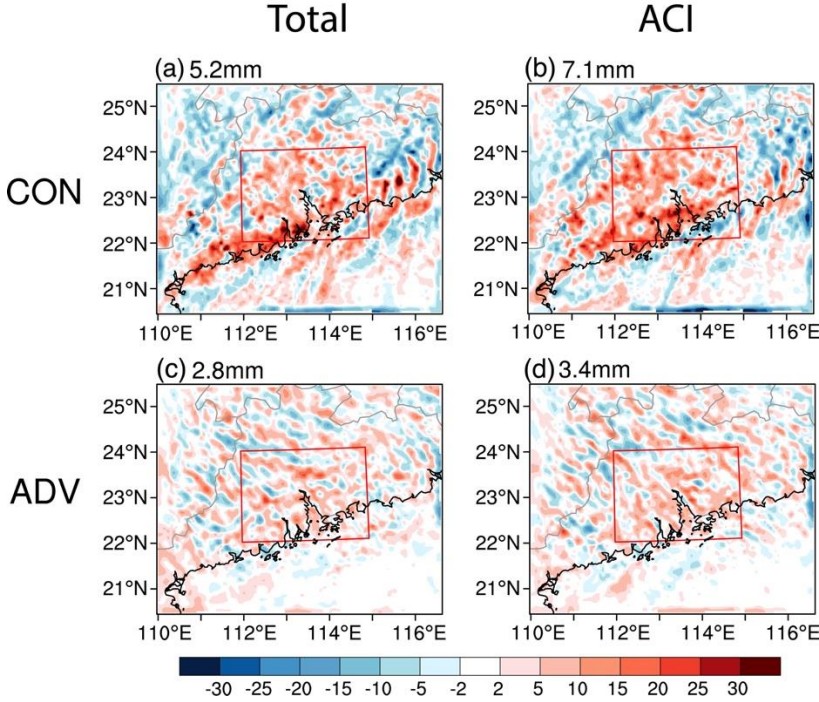

**Figure 9.** Differences in column-integrated moisture convergence (CON; unit: mm) between (a) CTL and CLEAN (i.e. CTL minus CLEAN) and (b) ARIoff and CLEAN (i.e., ARIoff minus CLEAN) on December 15. (c, d) Same as (a, b) but for column-integrated advection of water vapor (ADV; unit: mm). The numbers at the top-left corner of each panel represent the values averaged over the red boxes. The red boxes denote the analysis region.

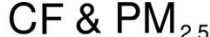

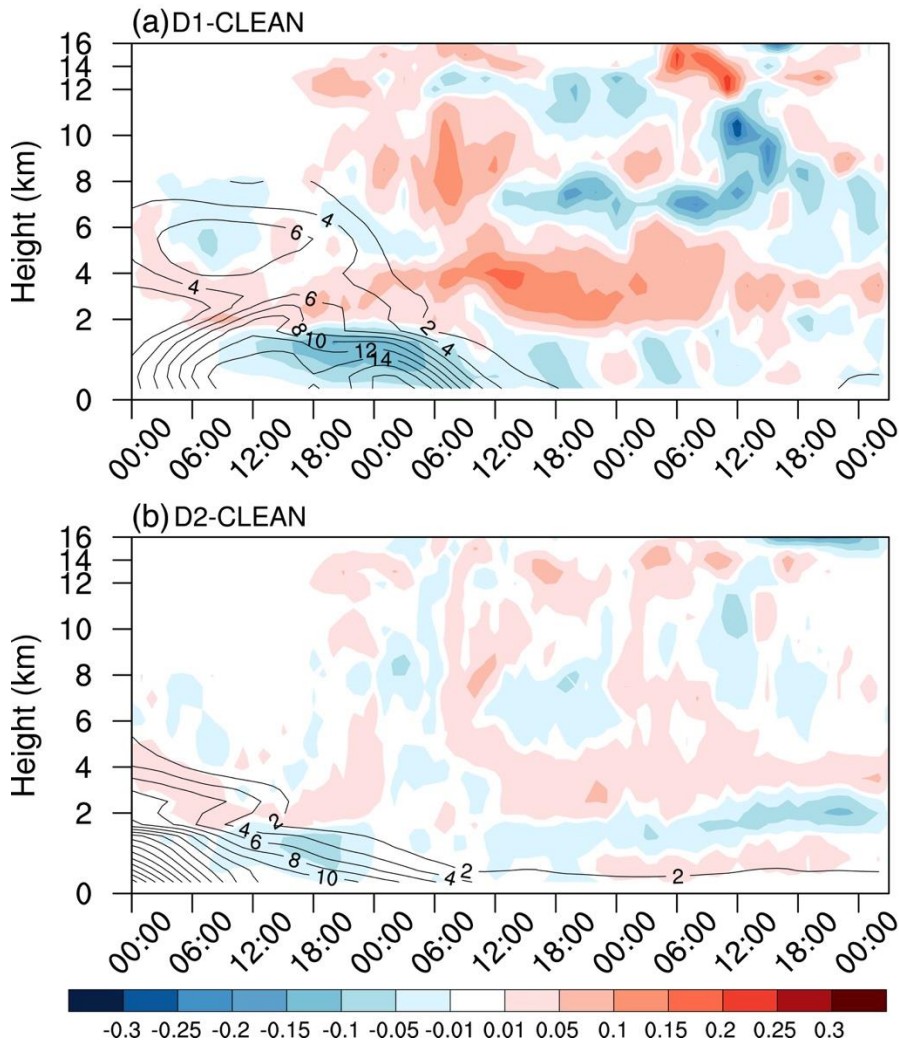

**Figure 10. Di fferences in time-height cross section of CF (shading; unit: unitless) and PM$_{2.5}$ concentration (contour; unit: μg m$^{-3}$) averaged over the red box shown in Figure 3 between (a) D1 and CLEAN (i.e., D1 minus CLEAN) and (b) D2 and CLEAN (i.e., D2 minus CLEAN).**

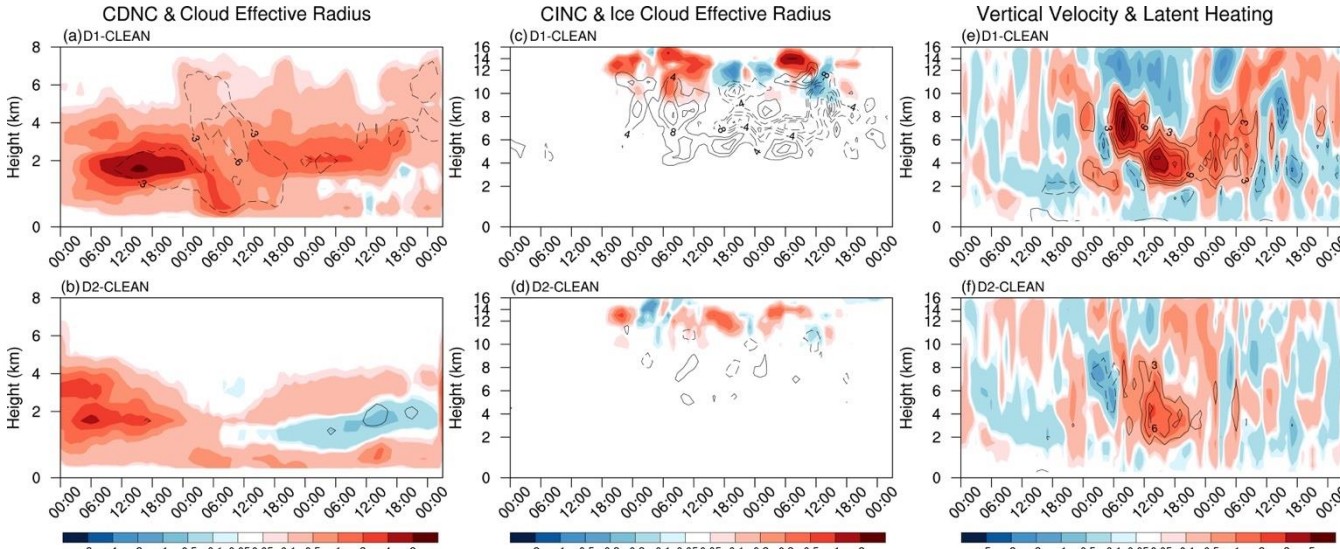

**Figure 11. Differences with time (abscissa; from 00Z on December 14 to 02Z on December 17) and height (ordinate) in (a) CDNC (shading; unit: $10^7$ kg$^{-1}$) and cloud effective radius (contour; unit: μm), (c) CINC (shading; unit: $10^5$ kg$^{-1}$) and ice cloud effective radius (contour; unit: μm), and (e) vertical velocity (shading; unit: cm s$^{-1}$) and latent heating (contour; unit: K d$^{-1}$) averaged over the red box shown in Figure 3 between D1 and CLEAN (i.e., D1 minus CLEAN; first row). (b, d, f) same as (a, c, e) but for differences between D2 and CLEAN (i.e., D2 minus CLEAN; second row). Zero-value contour lines are omitted, and negative values are dashed.**

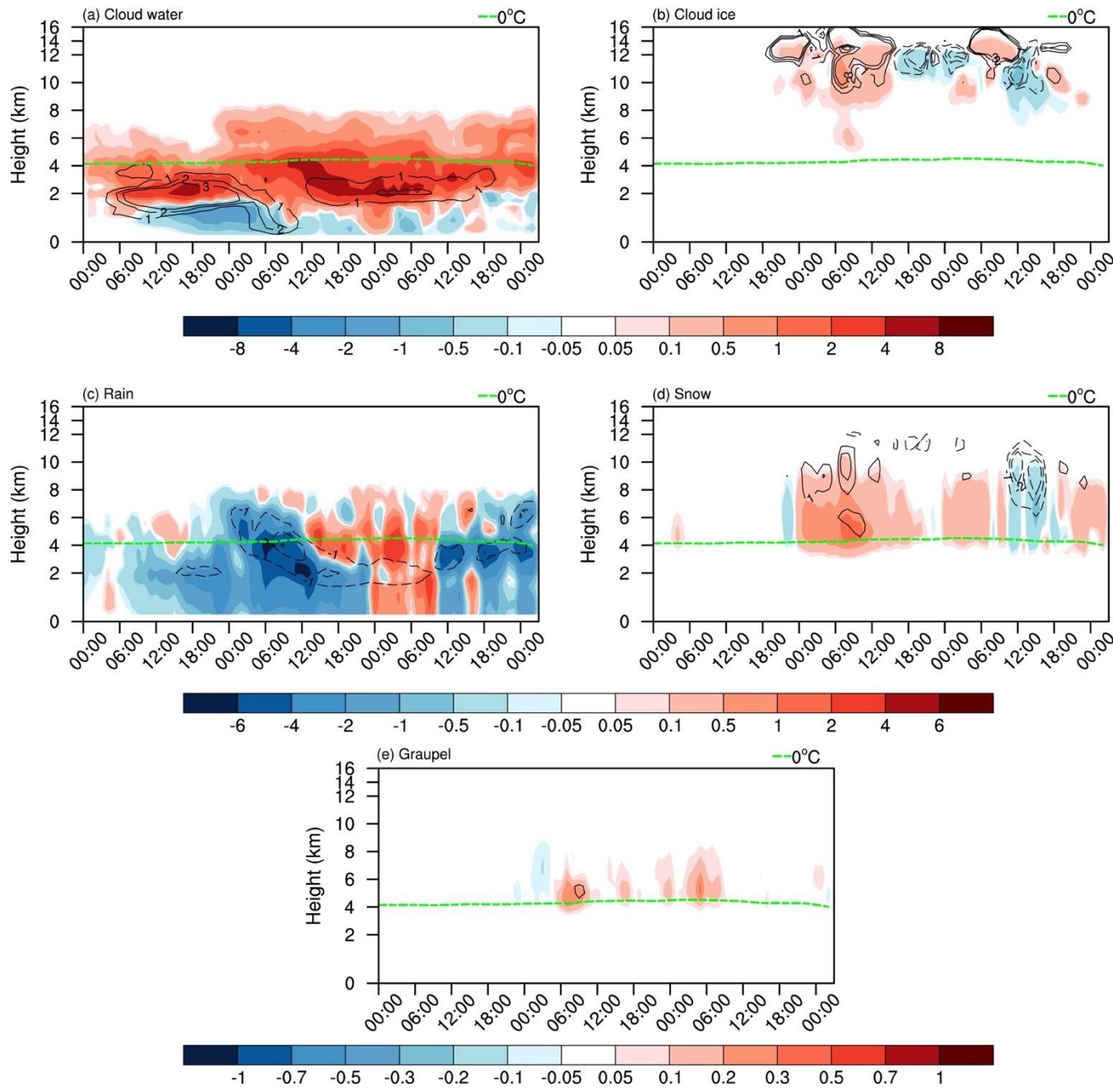

**Figure 12. Differences with time (abscissa) and height (ordinate) in (a) cloud water (shading; unit: $10^{-5}$ kg kg$^{-1}$) and CDNC (contour; unit: $10^7$ kg$^{-1}$), (b) cloud ice (shading; unit: $10^{-5}$ kg kg$^{-1}$) and CINC (contour; unit: $10^4$ kg$^{-1}$), (c) rain (shading; unit: $10^{-5}$ kg kg$^{-1}$) and rain number concentration (contour; unit: $10^5$ kg$^{-1}$), (d) snow (shading; unit: $10^{-4}$ kg kg$^{-1}$) and snow number concentrations (contour; unit: $10^3$ kg$^{-1}$), and (e) graupel (shading; unit: $10^{-4}$ kg kg$^{-1}$) and graupel number concentration (contour; unit: $10^3$ kg$^{-1}$) between D1 and CLEAN (i.e. D1 minus CLEAN) averaged over the red box.**

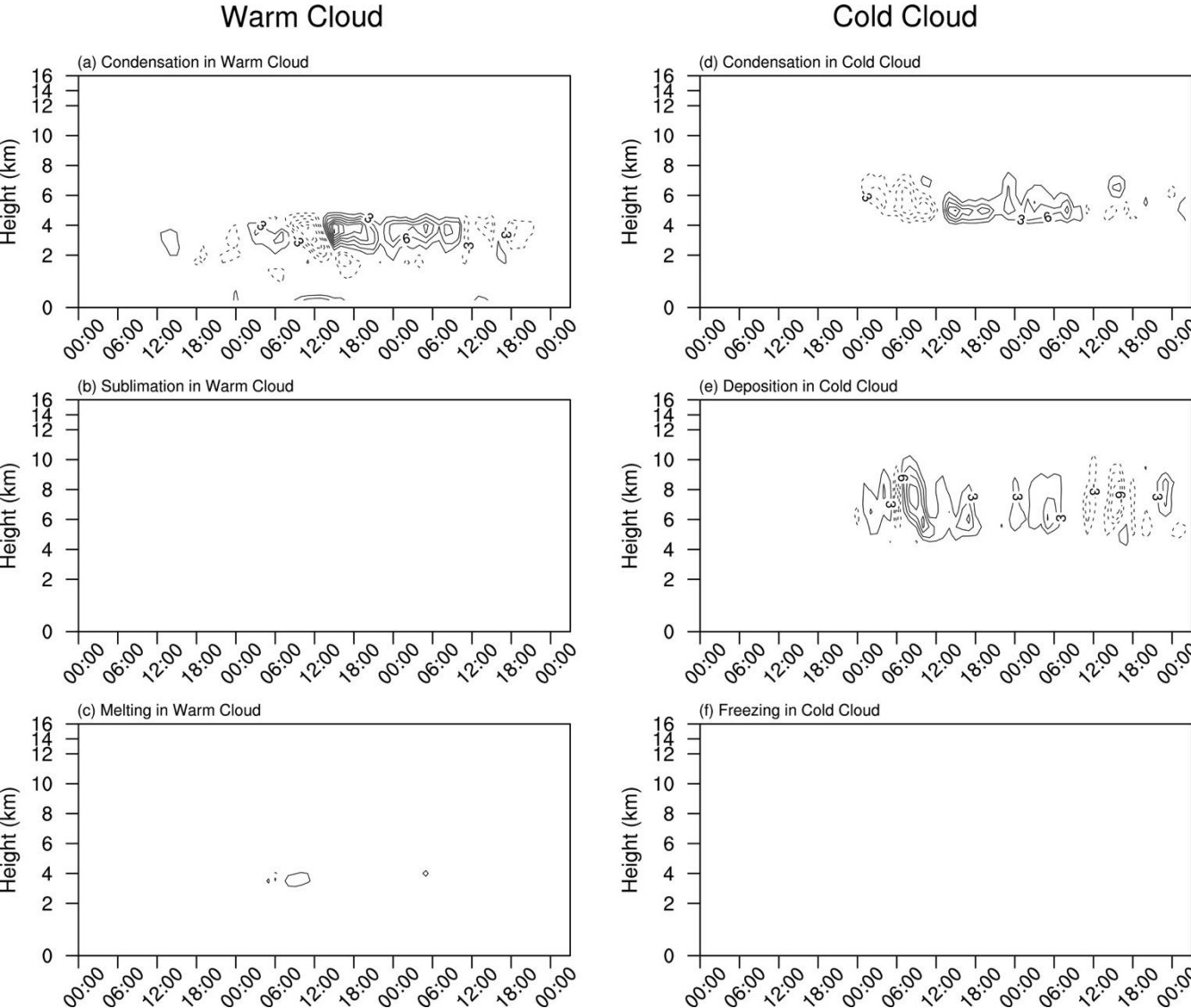

**Figure 13. Differences with time (abscissa) and height (ordinate) in latent heat release (unit: K d⁻¹) from (a) condensation, (b) deposition, and (c) freezing processes between D1 and CLEAN (i.e. D1 minus CLEAN) averaged over the red box for the warm cloud. (d–f) Same as (a–c) but from cold cloud. Zero-value contour lines are omitted, and negative values are dashed. The contour interval is 3 K d⁻¹.**

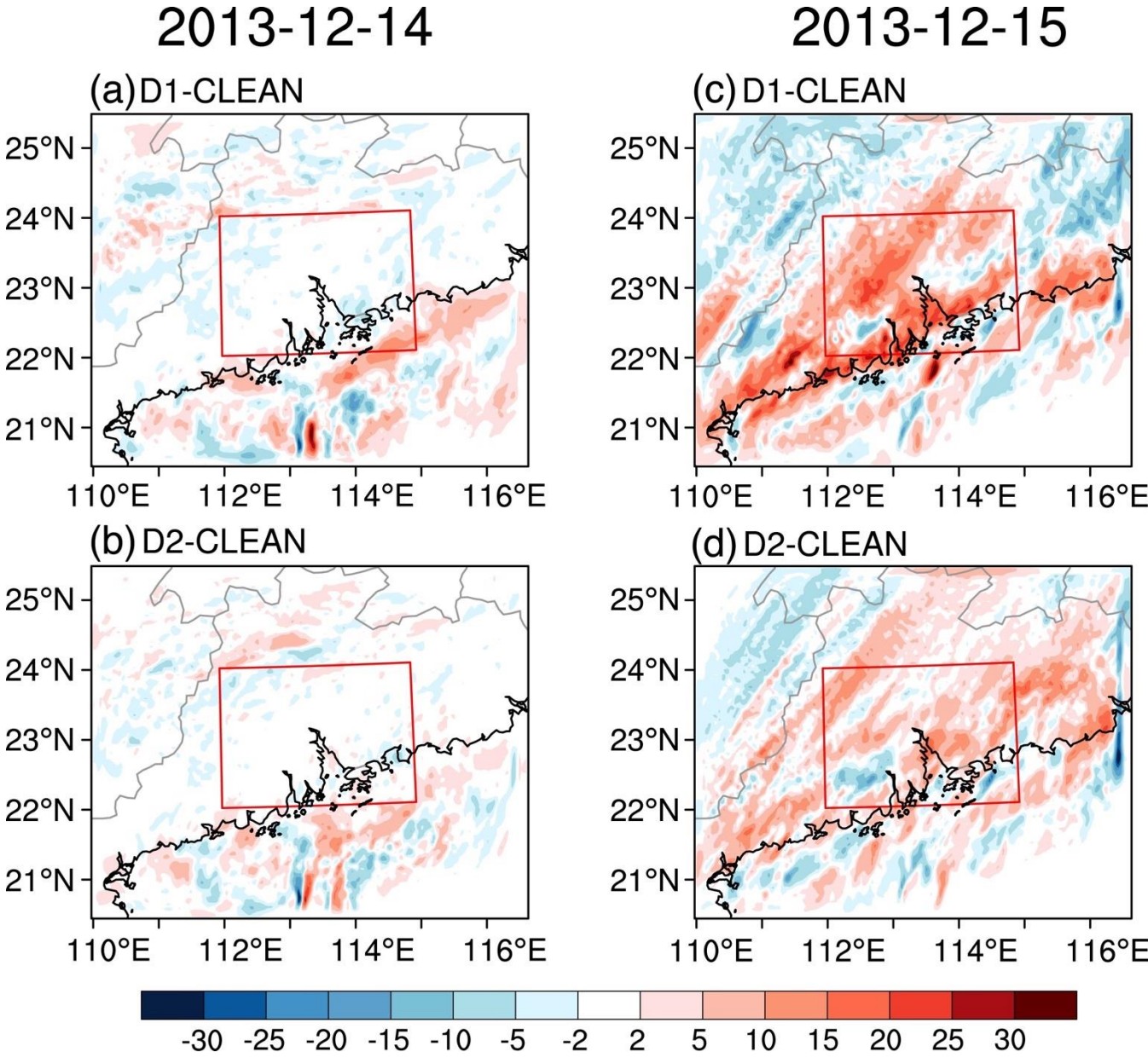

**Figure 14.** Differences in precipitation (unit: mm) between (a) D1 and CLEAN (i.e., D1 minus CLEAN) and (b) D2 and CLEAN (i.e., D2 minus CLEAN) on December 14. (c, d) Same as (a, b) but for December 15. Red boxes (22°–24° N, 112°–115° E) denote the analysis region.

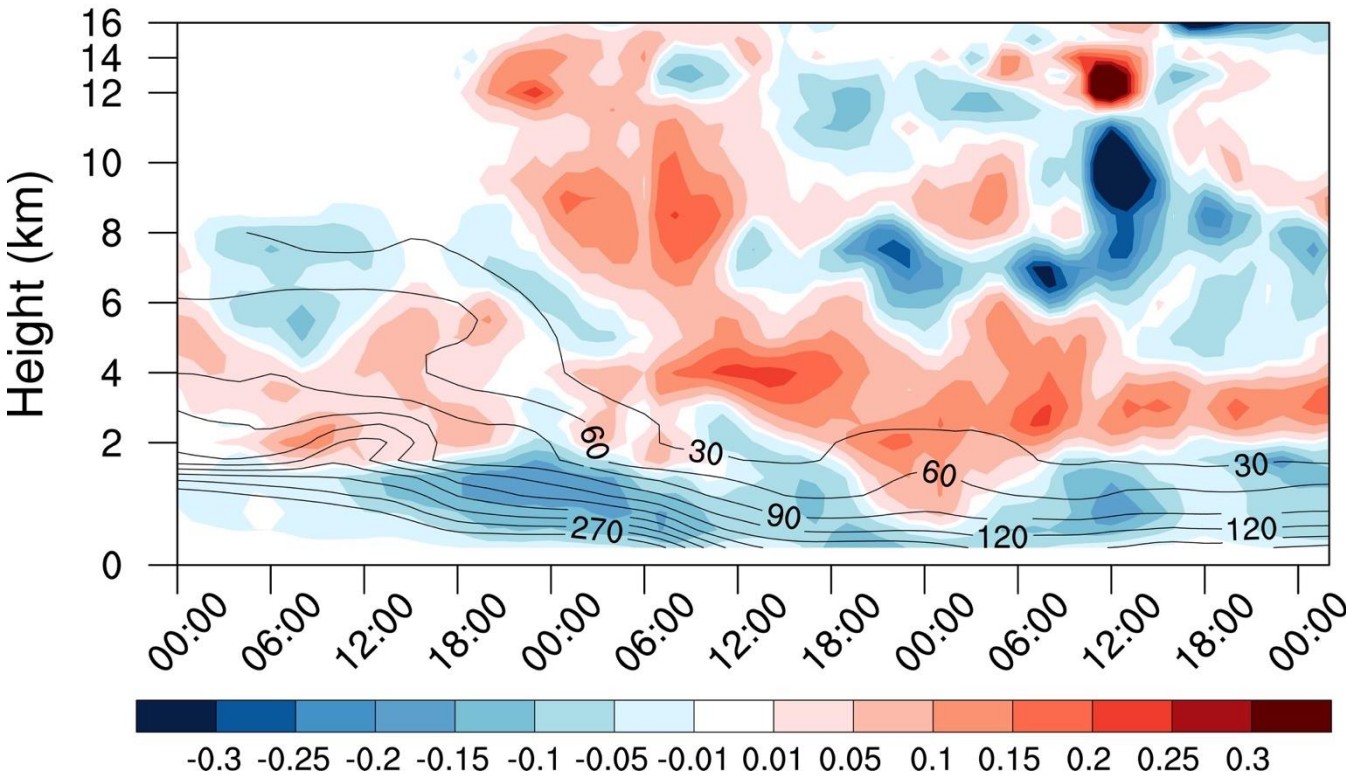

**Figure 15. Differences in the time-height cross section of cloud factor CF (shading; unit: unitless) and PM$_{2.5}$ concentrations (contour; unit: μg m$^{-3}$) averaged over the red box shown in Figure 3 between 10× and CLEAN (i.e., 10× minus CLEAN).**

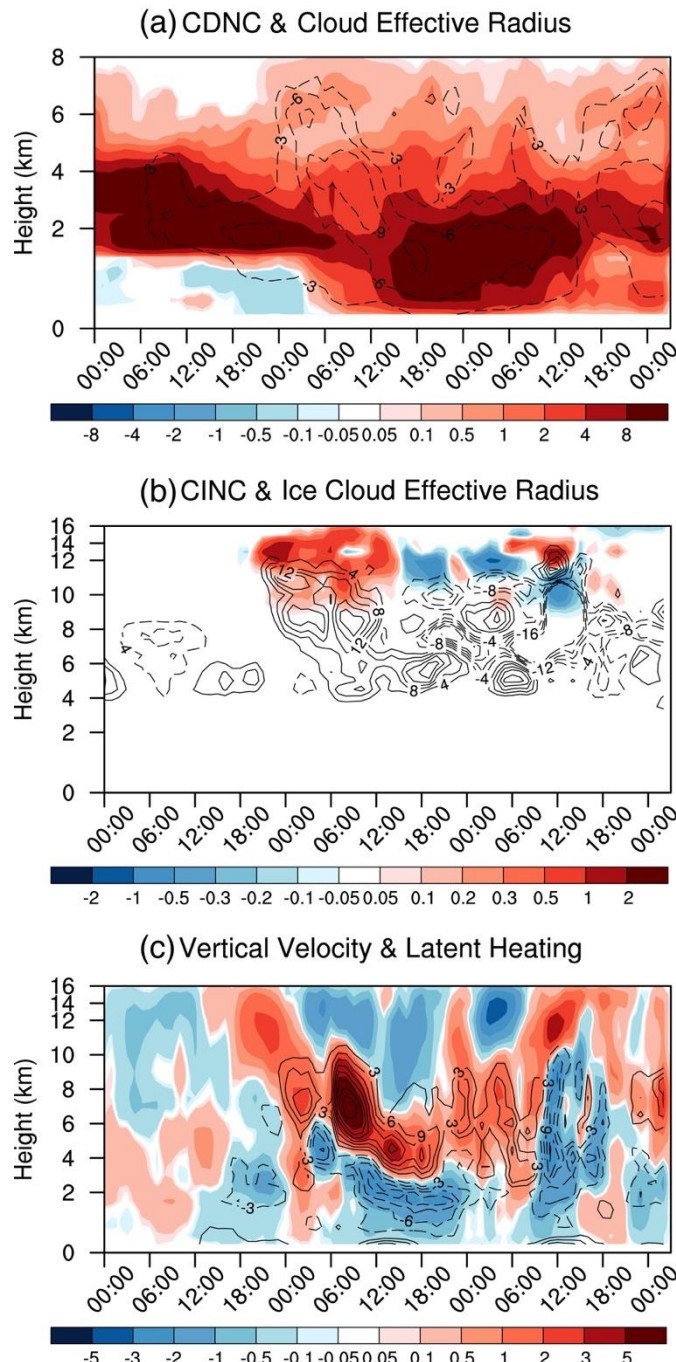

**Figure 16. Differences with time (abscissa; from 00Z on December 14 to 02Z on December 17) and height (ordinate) in (a) CDNC (shading; unit: $10^7$ kg$^{-1}$) and cloud effective radius (unit: μm), (b) CINC (shading; unit: $10^5$ kg$^{-1}$) and ice cloud effective radius (contour; unit: μm), and (c) vertical velocity (shading; unit: cm s$^{-1}$) and latent heating (contour; unit: K d$^{-1}$) averaged over the red box shown in Figure 3 between 10× and CLEAN (i.e., 10× minus CLEAN). Zero-value contour lines are omitted, and negative values are dashed.**

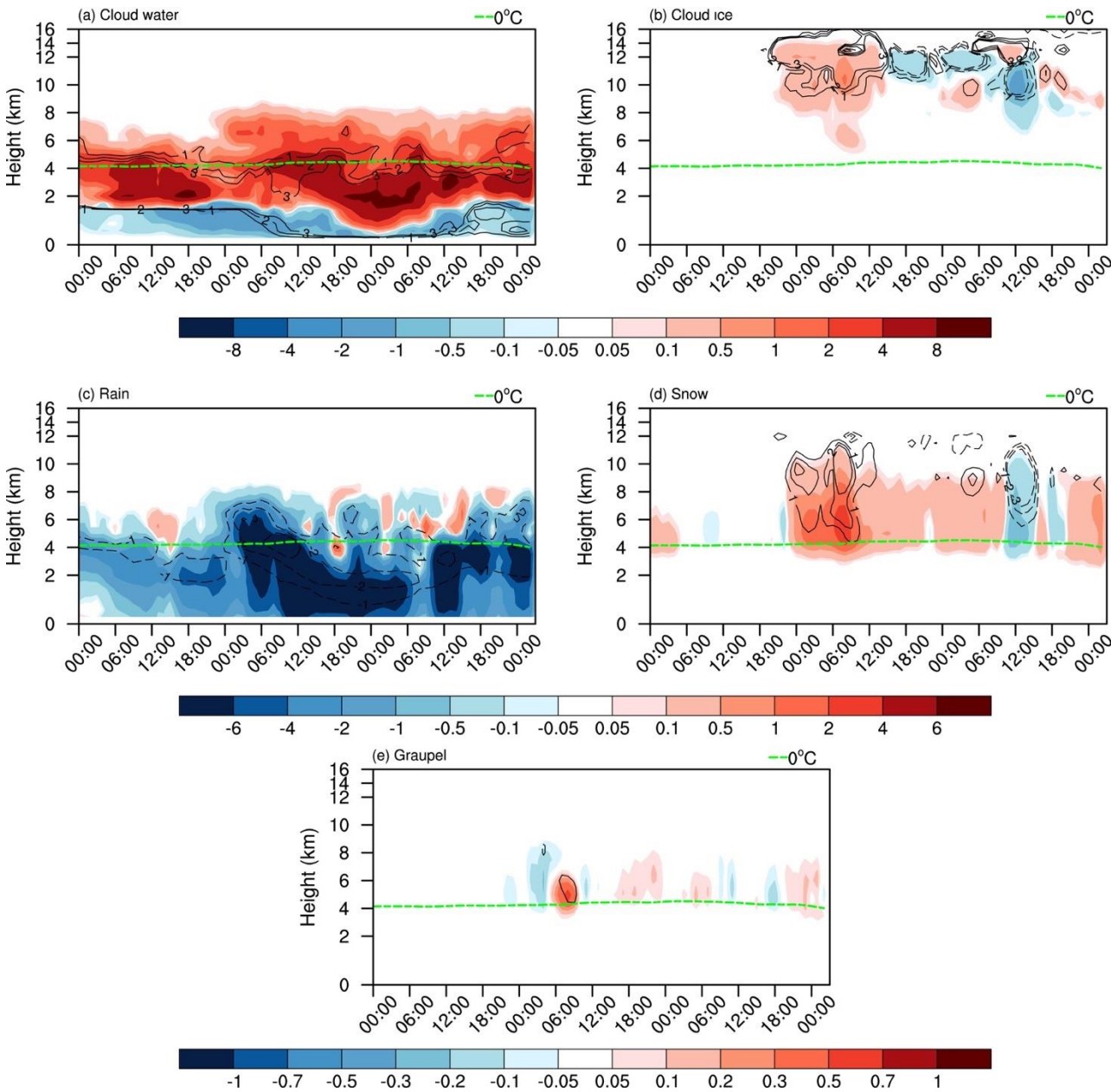

**Figure 17. Differences with time (abscissa) and height (ordinate) in (a) cloud water (shading; unit: $10^{-5}$ kg kg$^{-1}$) and CDNC (contour; unit: $10^7$ kg$^{-1}$), (b) cloud ice (shading; unit: $10^{-5}$ kg kg$^{-1}$) and CINC (contour; unit: $10^4$ kg$^{-1}$), (c) rain (shading; unit: $10^{-5}$ kg kg$^{-1}$) and rain number concentration (contour; unit: $10^5$ kg$^{-1}$), (d) snow (shading; unit: $10^{-4}$ kg kg$^{-1}$) and snow number concentrations (contour; unit: $10^3$ kg$^{-1}$), and (e) graupel (shading; unit: $10^{-4}$ kg kg$^{-1}$) and graupel number concentration (contour; unit: $10^3$ kg$^{-1}$) between 10× and CLEAN (i.e. 10× minus CLEAN) averaged over the red box.**

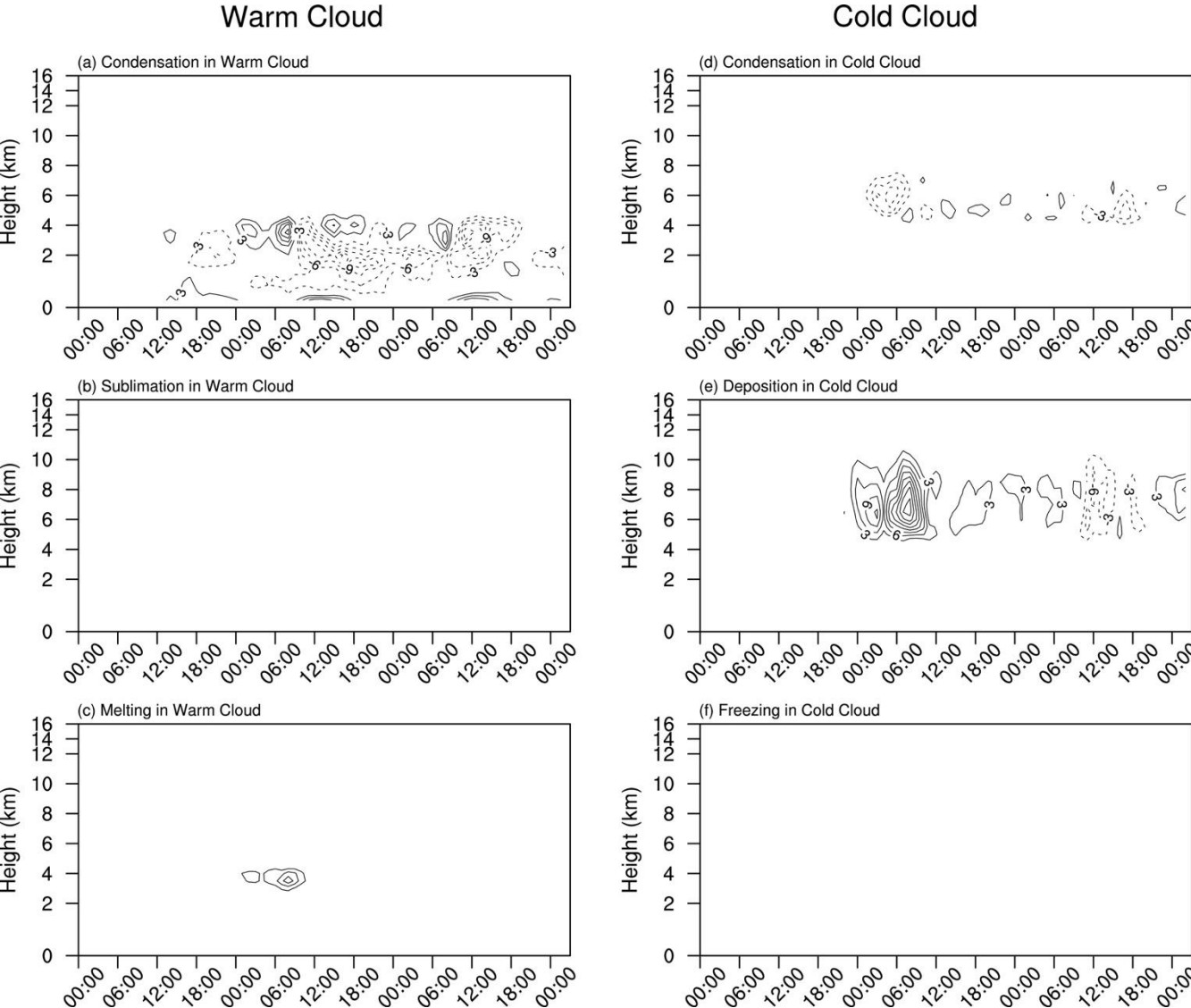

**Figure 18. Differences with time (abscissa) and height (ordinate) in latent heat release (unit: K d⁻¹) from (a) condensation, (b) deposition, and (c) freezing processes between 10× and CLEAN (i.e. 10× minus CLEAN) averaged over the red box for the warm cloud. (d–f) Same as (a–c) but from cold cloud. Zero-value contour lines are omitted, and negative values are dashed. The contour interval is 3 K d⁻¹.**

# Precipitation

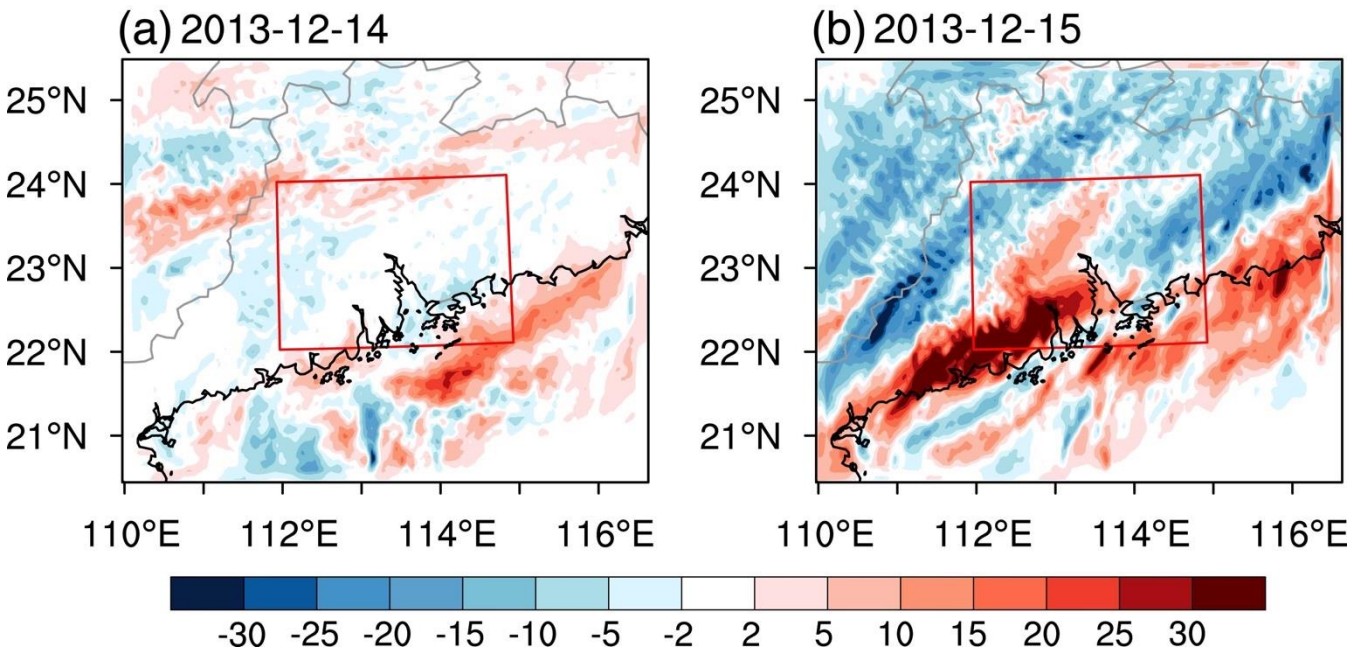

**Figure 19. Differences in precipitation (unit: mm) between 10× and CLEAN (i.e., 10× minus CLEAN) on (a) December 14 and (b) December 15. Red boxes (22°–24° N, 112°–115° E) denote the analysis region.**

# Wind Shear

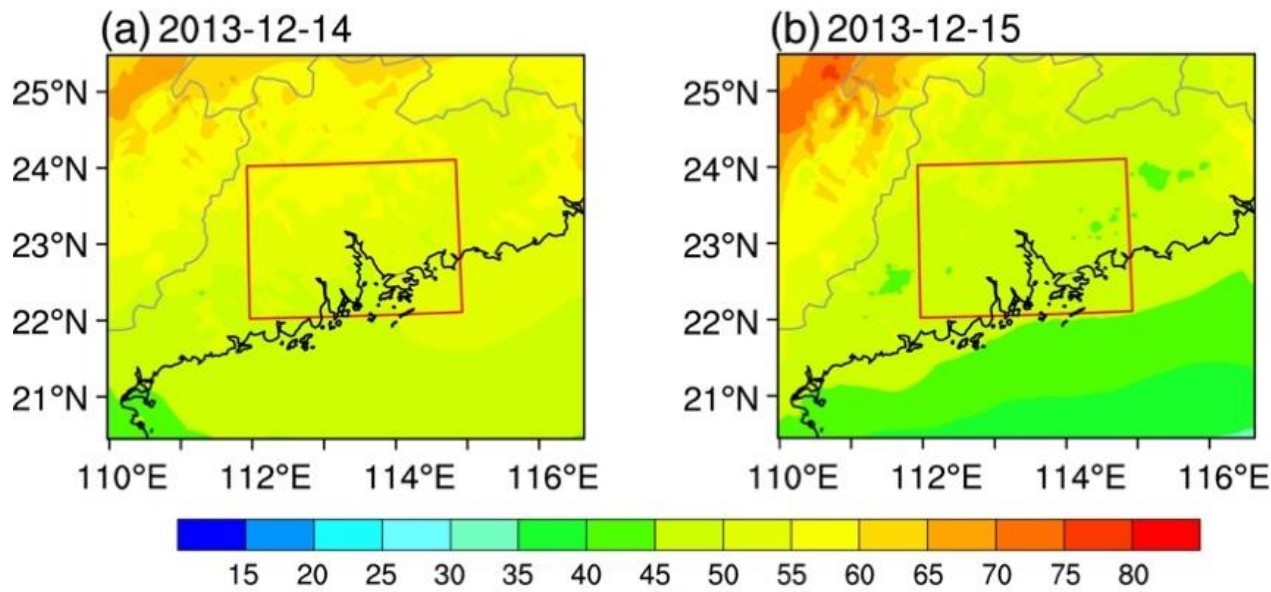

# Water Vapor & 925-hPa Wind

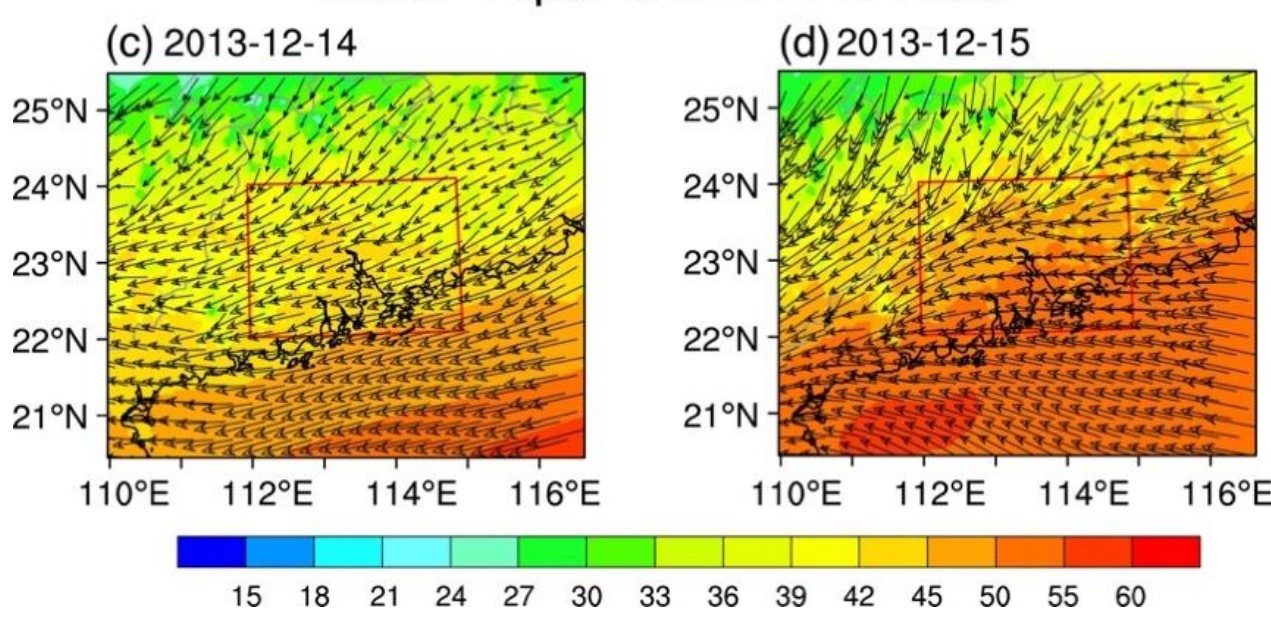

Figure 20. Spatial distribution of wind shear (unit: m s$^{-1}$) on (a) December 14 and (b) December 15 in 2013 in the CTL run (first row). Wind shear is calculated as differences between maximum wind speed and minimum wind speed at 0–10 km. Spatial distribution of column-integrated water vapor (shading; unit: mm day$^{-1}$) and 925-hPa wind (vector; unit: m s$^{-1}$) on (c) December 14 and (d) December 15 in 2013 in CTL (second row). Red boxes (22°–24° N, 112°–115° E) denote the analysis region.

Table 1. Model simulations. Abbreviations: CTL, control run; ARIoff, turn off aerosol-radiation interactions; D1, keep emissions in domain 1 as control run while make those except for chemical boundary conditions in domain 2 as CLEAN run; D2, keep emissions and chemical initial conditions in domain 2 as control run, make those and chemical boundary conditions in domain 1 as CLEAN run; 10×, tenfold of anthropogenic emissions and chemical initial and boundary conditions. * indicates that emissions, initial conditions (ICs), or boundary conditions (BCs), are scaled from the control run. Note the offline chemical BCs here were extracted from global chemical transport models and only used for domain 1.

| Simulation | Anthropogenic and fire emissions, chemical ICs and BCs* | | Aerosol-radiation interactions | Aerosol-cloud interactions |
|---|---|---|---|---|
| | Domain 1 | Domain 2 | | |
| CTL | 1 | 1 | Yes | Yes |
| ARIoff | 1 | 1 | No | Yes |
| CLEAN | 0.1 | 0.1 | Yes | Yes |
| D1 | 1 | 0.1 | Yes | Yes |
| D2 | 0.1 | 1 | Yes | Yes |
| 10× | 10 | 10 | Yes | Yes |

Table 2. Description of warm-cloud processes contributing to latent heat release. Red, green, and blue indicate condensation, deposition, and freezing related processes, respectively. If the term is negative, it refers to the opposite transfer process from sink to source. For example, negative of PRE represent the evaporation of $Q_r$.

| Abbreviation | Warm-cloud processes | Source | Sink |
|---|---|---|---|
| PRE | Condensation of $Q_v$ | $Q_v$ | $Q_r$ |
| PCC | Condensation of $Q_v$ | $Q_v$ | $Q_c$ |
| EVPMS | Sublimation of $Q_s$ | $Q_s$ | $Q_v$ |
| EVPMG | Sublimation of $Q_g$ | $Q_g$ | $Q_v$ |
| PSMLT | Melting of $Q_s$ | $Q_s$ | $Q_r$ |
| PGMLT | Melting of $Q_g$ | $Q_g$ | $Q_r$ |
| PRACS | Collection of $Q_r$ by $Q_s$ | $Q_r$ | $Q_s$ |
| PRACG | Collection of $Q_r$ by $Q_g$ | $Q_r$ | $Q_g$ |

Table 3. Same as Table 1 but for cold cloud.

| Abbreviation | Cold-cloud processes | Source | Sink |
|---|---|---|---|
| PRE | Condensation of $Q_v$ | $Q_v$ | $Q_r$ |
| PCC | Condensation of $Q_v$ | $Q_v$ | $Q_c$ |
| PRD | Deposition of $Q_v$ | $Q_v$ | $Q_s$ |
| PRDS | Deposition of $Q_v$ | $Q_v$ | $Q_g$ |
| MNUCCD | Ice nucleation | $Q_v$ | $Q_i$ |
| EPRD | Sublimation of $Q_i$ | $Q_i$ | $Q_v$ |
| EPRDS | Sublimation of $Q_s$ | $Q_s$ | $Q_i$ |
| PRDG | Deposition of $Q_v$ | $Q_v$ | $Q_g$ |
| EPRDG | Sublimation of $Q_g$ | $Q_g$ | $Q_v$ |
| PSACWS | Accretion of $Q_c$ by $Q_s$ | $Q_c$ | $Q_s$ |
| PSACWI | Accretion of $Q_c$ by $Q_i$ | $Q_c$ | $Q_i$ |
| MNUCCC | Contacting freezing of $Q_c$ | $Q_c$ | $Q_i$ |
| MNUCCR | Contacting freezing of $Q_r$ | $Q_r$ | $Q_g$ |
| QMULTS | Multiplication due to collision $Q_c$ by $Q_s$ | $Q_c$ | $Q_i$ |
| QMULTG | Multiplication due to collision $Q_c$ by $Q_g$ | $Q_c$ | $Q_i$ |
| QMULTR | Multiplication due to collision $Q_r$ by $Q_s$ | $Q_r$ | $Q_i$ |
| QMULTRG | Multiplication due to collision $Q_r$ by $Q_g$ | $Q_r$ | $Q_i$ |
| PRACS | Collection of $Q_r$ by $Q_s$ | $Q_r$ | $Q_s$ |
| PSACWG | Collection of $Q_c$ by $Q_g$ | $Q_c$ | $Q_g$ |
| PRACG | Collection of $Q_r$ by $Q_g$ | $Q_r$ | $Q_g$ |

| | | | |
|---|---|---|---|
| PGSACW | Collection of $Q_c$ by $Q_s$, conversion to $Q_g$ | $Q_c$ | $Q_g$ |
| PGRACS | Collection of $Q_r$ by $Q_s$, conversion to $Q_g$ | $Q_r$ | $Q_g$ |
| PIACR | Collection of $Q_r$ by $Q_i$, conversion to $Q_g$ | $Q_r$ | $Q_g$ |
| PIACRS | Collection of $Q_r$ by $Q_i$ conversion to $Q_s$ | $Q_r$ | $Q_s$ |