# Peer review of "Contribution of local and remote anthropogenic aerosols to a recordbreaking torrential rainfall event in Guangdong province, China"

_Atmospheric Chemistry and Physics, 2018_

## Referee Comment (RC1) · Anonymous Referee #1 · 26 Nov 2018

In this study, the authors employed WRF-Chem to study the influence of anthropogenic aerosols on a relatively-heavy rainfall event. They showed that aerosol enhanced precipitation in southern part of the domain and aerosol– cloud interactions (ACI) is the main reason for the response. They further did sensitivity studies and found that remote aerosols contributed more than twice the precipitation increase compared with local aerosols. By further increasing emission by 10 times, their figures showed that more significant decrease and increase of precipitation in the respective cloud regimes (I did not use the wording from the authors because I do not agree with it).

Major comments: The study has a good scientific value. However, the current paper is

not appropriate for a publication yet. The main problems this reviewer found are, 1. The authors missed some key points when interpreting their results. The simulated cloud system seems like a cold front system meeting with warm and moist air. The cloud regimes should be very different over the code side of the frontal system compared with clouds at the convergence zone and warm side of the system. This key message should be considered when analyzing aerosol-cloud interactions since ACI strong depends on different cloud regimes. Decrease and increase in precipitation are seen over the different parts of the domain (Figure 3d) but the authors ignored the decrease part which is at the cold side of the system but focused on the increased part. When further increasing emission by 10 times, there is enhanced decrease (Figure 12b) but the authors still ignored it. Another misleading analysis is that the authors used the domain averaged vertical cross section plots and viewed then as single deep convective cell to discuss the ACI effect. The low-level clouds shown in such plots might not be vertically connected with the higher-level clouds. For example, shallow clouds could mainly occur in the northern part of the red box area used for the analysis and deep convective clouds could mainly occur in the southern part. Increasing aerosols suppresses shallow convection, which would be different from the story that the authors described in the paper. 2. The authors did not present enough data to examine the things they claimed for. In a few places as detailed in the specific comments, the authors assumed the literature work applies well to this study without presenting the key results to prove the point. See specific comments #14, 18, 20, 21, and 22. 3. There are many inaccurate or misleading statements. I noticed they are mainly related to the lack of expertise in cloud physics and weather area, such as #7, 10, 13, 18, 20, 21, and 22.

Specific comments: 1. P2, the later part of the last paragraph discusses literature study about ACI, which does not include the most recent work on this topic from a Science article (Fan et al., 2018).

2. P2, L19: "the slowing autoconversion rate induced by aerosols forms airborne cloud droplets in clouds" is confusing. First, what is "airborne cloud droplets in clouds"?

[Figure]

since it is in clouds, why call it "airborne"? second, how does antoconversion form cloud droplets?

3. P5, L6, based on Fan et al. 2015, the factor used in the study is 0.3 (not 0.1).

4. P5, L12-14, not sure how IC, BC, and emissions were treated in both domains in both D1 and D2. If Dom1 and Dom2 are run at the same time, which means dom 1 provides IC and BC for domain 2, then how to change IC and BC in Dom 2 for D2? In addition, if emission does not change in Dom2, wouldn't the local effect be underestimated?

5. P5, L18-19: Which simulation is 10X based on?

6. Section 3.1: since there are 58 stations for PM2.5 measurements in Domain 2, why not use them to evaluate the control simulations since the aerosol property is important to aerosol impacts?

7. P7, L20-22, this sentence is not justified. It could only because in the second day there were much larger in-cloud and below-cloud scavenging of aerosols due to much cloud and rain. The smaller aerosol effect in the first day can be a result of many factors particularly meteorological conditions, and the larger effect you see in the next day might not so related to the aerosols in the first day.

8. Figure 4, how is the cloud fraction calculated? Is the difference in percentage or absolute difference?

9. P7, L25-28, Better to use percentage differences or both in terms of quantifying the accumulated rain.

10. P8, L6-11, the whole description here has a problem. The way it describes currently basically says that aerosols are the reasons responsible for the more and deeper clouds at later time and less and shallower clouds at the earlier time, which should not be true. The first order is the meteorological conditions that are responsible for the cloud amount and vertical distribution. On the top of it, aerosol may influence it, and then you can describe the influence in more quantitative way.

11. Figure 5, the figure caption needs to be consistent with the figure label. If you label your panels in a, b, c,.., you need describe your figures in the same way so that readers can follow. This comment apply to many other figures. Also, the caption already has too many acronyms while another acronym CI for contour interval is used here, which only causes poor readability and confusion here. About the differences in CINC and ice effective radius, did you only consider cloud ice crystals, or all of the ice-phase particles were considered?

12. P 8, L15, when you say "dramatically", you need to give a quantitative value. Figure 5 only shows the absolute differences, which has the maximal value at the magnitude 80 cm-3 based on the legend. This change is not large unless you did a domain average and there are many cloud-free points in the analysis domain.

13. P 8, L20, what do you mean about "the interim processes"?

14. P8, L17-27, the entire description here about the ACI effect is not about the results from their study. The authors just followed what the literature describes. First, the description and the corresponding references do not reflect the symbolic literature studies on ACI on deep convective clouds. First, the idea of convective invigoration by enhanced latent heat from cold-phase processes (due to suppression of warm rain) starts from Andreas et al. Science, 2004 (obs), then Khain et al., QJR, 2005 (model), Rosenfeld et al. Science, 2008 (theoretical), Fan et al., JGR, 2009, etc, did detailed studies about it. The authors did not mention these studies at all (Rosenfeld et al., 2008 is not discussed in an appropriate way since it is the theoretical study for this theory). Second, the most recent development of ACI is the "warm-phase invigoration" in Fan et al. Science, (2018) where latent heat release from enhanced condensation is emphasized as a reason for the enhanced updraft speed. From Figure 5, the latent heat enhancement peaks below 8 km altitudes and there is a peak at 3-5 km, suggesting condensational heating might play a significant role here as well. The latent heat enhancement at low part of clouds from condensation plays a much more significant role than the correspondent at high levels as shown in Fan et al. 2018. The authors

need to examine this in detail to understand what's the real reason behind it instead of just citing some literature studies since ACI is a key point of the study.

15. P8, Eq (1), where is the horizontal advection terms for the moisture budget? In the model, this is an important term. If you considered it in the vertically integrated moisture flux (MFC) convergence in your calculation, then the MFC should be large at the convection permitting scale.

16. P9, L8, the figure number is wrong. Also, where is the moisture coming from the northerly wind since northerly wind generally brings in drier air? It would be good to show the spatial distribution of moisture field.

17. P 9, L11-12, since there is compensation effect here, a figure for ARI effect should be shown to quantify how much is the compensation effect.

18. P9, L16-20, again, key processes are not shown and the summary description might not be accurate. First, it is not correct to say "water clouds ascend to freeze into ice clouds" since it is just that more cloud droplets are lifted to the higher levels and form more ice particles. Second, as I pointed out above, the source of latent heat enhancement is not examined and the authors just assumed it is mainly resulted from more droplet freezing. Third, the much enhanced horizontal convergence could be gradually induced by other feedback such as precipitation or radiation since the simulation duration are a few days, not just a few hours. Another question is that how the changes in domain 1 impact the results over domain 2?

19. Section 3.3, the remote and local aerosol effects can strongly depend on how strong the coupling between the two domains. With the two domains running together, the coupling is very strong and the Dom 1 keeps updating Dom 2, which could lead to very strong effect from any variable in Dom 1(not just aerosol). If you run domain 2 separately with the IC and BC updated in every 3-hours or 6 hours from Dom 1, and do the same studies, the results could be changed.

20. P10, L28-29, this may indicate secondary droplet nucleation, meaning activating enormous smaller aerosols at higher-levels due to higher supersaturation. Without looking at it carefully, you can not just assume it is mainly because of ascent of cloud droplets.

21. P11, L1-6, again, you can not just guess by citing a literature work assuming it apply to your study. Key results need to be shown. The reduction of low-level cloud could just because more deep cloud form consuming moisture and energy which would limit the formation of other type of clouds. Evaporation and sublimation have to come from clouds. In addition, the lower-level cloud and the high-level clouds shown here might not be vertically connected over the domain. For example, shallow clouds could mainly occur in the northern part of the red box associated with cold front and deep clouds could mainly occur in the southern part associated with the convergence zone. Increasing aerosols suppresses shallow convection, which would be different from the story you describe here now. It would not be nothing to do with sublimation if that is the case.

22. P11, L8-17, do not agree with some of the discussion. Compared with Figure 3, I only see the corresponding increase and decrease in the Dom 2 become more significant in Figure 12. The authors did not discuss why there are two significantly different precipitation response regimes to the change of emissions. It seems that they are located in different dynamic regimes so have different cloud types. More detailed description about what types of clouds were formed in the cold side is needed in the description of the case at the beginning of the result section. It would provide basis for the related discussion after that.

23. Section 4: a. First paragraph, Summary should include description of what have been done as well.

b. Second paragraph, see my comment above about how to look at different aerosol impacts on different cloud regimes/types. The current discussion might not relevant

because the cloud types should be very different between the code and warm sides of the frontal system.

c. The third and fourth paragraphs may need to be changed accordingly after my relevant comments above are addressed.

Grammatical problems: P4, L19: grammar error. P7, L6: grammar error. P7., L32, past tense is not needed here. There are many places in results section that have the mixed past and current tenses. Better to be consistent in tense to improve readability and avoid confusion.

---

## Referee Comment (RC2) · Anonymous Referee #2 · 28 Nov 2018

This study employed the WRF-Chem model and a series of sensitivity experiments to study the aerosol microphysical and radiative impacts on a historical heavy precipitation event in southern China. The effects of local and remote aerosols are compared by altering aerosol concentrations in different domains. The finding about the aerosol invigoration effect with a moderate aerosol increase generally agrees with the existing argument that aerosols tend to induce more extreme precipitation. The topic of study is important and fits with the scope of ACP very well. However, there are still lots of unclear writing and insufficient analyses in the manuscript. Major revisions are needed before it can be accepted by ACP.

*Two major comments:*
1) It is kind of surprising to me that the simulated aerosol properties and spatial distributions are not shown in the manuscript. They are actually about the strength of the WRF-Chem model in doing aerosol-cloud research. PM2.5 is plotted, but it is not quantitative index for either CCN effect or radiative effect. The spatial distributions are critical for us to understand the potential influence of remote aerosols. The aerosol chemical component determines the aerosol radiative properties, absorbing or scattering, as well as CCN ability.

2) Process-level analyses on ACI and ARI in this case should be strengthened. For ARI, I do not see any analysis on the radiative fluxes, temperature field, and associated dynamical adjustment. Is there any atmospheric heating due to black or brown carbon in this case? For ACI, the microphysical properties of all hydrometeor and co-varying water vapor field should be studied. See more specific comments below.

*Specific comments:*
1) I feel the literature review in the introduction part is not done thoroughly. Considering both ACI and ARI have been extensively investigated for the past 10-20 years, more credits should be given to the studies with the similar topic.
  - P2L4, inaccurate statement. Actually, there are lots of existing studies on the influence of aerosols on different types of extreme weather, such as tropical cyclone (Wang et al., 2014, Nat. Clim. Change; Zhao et al., 2018, GRL), hail storm (Ilotoviz, et al., 2016, JAS), etc.
  - P3L1, the competition between ARI and ACI has been widely discussed on both cloud-resolving scale (Lin et al., 2017, JAS; Wang et al., 2018, AAS) as well as regional climate scale (Wang et al., 2016, JGR).
  - P3L3-5, different aerosol types can be a critical factor as well to determine the invigoration or suppression effect of aerosols (Jiang et al., 2018, Nat. Commu.).

2) P3L15-20, it is not clear what are hypotheses for the different effects from local and remote aerosol emissions? Different concentrations, chemical compositions, or spatial distributions? What did observations tell us about their differences? Without stating those explicitly, readers fail to follow the logic flow of the paper.

3) Fig. 2b and 2c, rather than only showing the dots over the stations, I suggest to plot the rainfall map over the whole domain for model and satellite, which is helpful to characterize the system. You can still keep open circles to compare the rainfall over each station.

4) Fig. 1b, the photo here deliver very litter information. Suggest to replace by the wind pattern analysis . Also, P6L20, a more accurate expression here is "monsoon system" or "monsoonal flow".

5) P6L29-31, why not using TRMM which is better at heavy precipitation? What is the point to show a satellite product even worse than the model?

6) The physical meaning of ARIoff – CLEAN is not obvious, as the authors use (CTRL – CLEAN) – (CTRL – ARIoff) to approximate ACI. I suggest the authors state this assumption explicitly and use ACI to replace ARIoff – CLEAN for all figure legends. Also, be careful about the usage difference between hyphen and minus sign.

7) It is unclear for me how the statistical analysis is conducted in those figures. As I understand, the authors only have one run for each model configuration. How to get the sufficient samples for the Student's t-test at each grid point?

8) P8L15-25, the authors only mentioned about the latent heat from droplet freezing. However, according to Fig. 5, clearly there is a significant portion of latent heat release below 4 km (warmer than 0 degree C). Can you plot the changes in liquid water content to confirm it? For the oceanic DCC, aerosol induced diabetic heating has two peaks, one in the warm portion and one in the mixed-phase portion (Fig. 3a of Wang et al., 2014, Nat. Commun.). Another interesting point here is that the Morrison microphysical scheme in WRF-Chem uses the simple water vapor saturation adjustment for condensation/evaporation. I speculate that this scheme cannot account for CCN effect in fostering condensation. Since this paper indirectly infers more liquid water forms, it is intriguing to see why.

9) Fig. 4,7, cloud fraction is about cloud macrophysics, which may not accurately reflect changes in cloud microphysics (water content, number concentration). The latter are more relevant with the aerosol invigoration effect. As mentioned above, I strongly suggest the authors plot and systematically analyze the changes in the mass and number concentration of the different hydrometeors.

---

## Author Comment (AC1) · 25 Apr 2019

Dear Editor,

We would like to thank for reviewers' valuable comments and suggestions. Their comments are addressed as shown below. Hope they find our revisions useful. Thank you very much.

--
Regards,
Steve
* * *
YIM, Hung-Lam Steve, Ph.D.

Assistant Professor
Department of Geography and Resource Management

The Chinese University of Hong Kong, Shatin, Hong Kong
Tel: (852) 3943 6534
Fax: (852) 2603 5006
Email: steveyim@cuhk.edu.hk
GRMD@CUHK: http://www.grm.cuhk.edu.hk/eng/

**Reviewer 1:**

We thank the reviewer for very helpful comments. Your comments and suggestions are addressed accordingly. Thank you very much for your effort.

**In this study, the authors employed WRF-Chem to study the influence of anthropogenic aerosols on a relatively-heavy rainfall event. They showed that aerosol enhanced precipitation in southern part of the domain and aerosol– cloud interactions (ACI) is the main reason for the response. They further did sensitivity studies and found that re- mote aerosols contributed more than twice the precipitation increase compared with local aerosols. By further increasing emission by 10 times, their figures showed that more significant decrease and increase of precipitation in the respective cloud regimes (I did not use the wording from the authors because I do not agree with it).**

**Major comments:**

**1. The authors missed some key points when interpreting their results. The simulated cloud system seems like a cold front system meeting with warm and moist air. The cloud regimes should be very different over the code side of the frontal system compared with clouds at the convergence zone and warm side of the system. This key message should be considered when analyzing aerosol-cloud interactions since ACI strong depends on different cloud regimes. Decrease and increase in precipitation are seen over the different parts of the domain (Figure 3d) but the authors ignored the decrease part which is at the cold side of the system but focused on the increased part. When further increasing emission by 10 times, there is enhanced decrease (Figure 12b) but the authors still ignored it. Another misleading analysis is that the authors used the domain averaged vertical cross section plots and viewed then as single deep convective cell to discuss the ACI effect. The low-level clouds shown in such plots might not be vertically connected with the**

**higher-level clouds. For example, shallow clouds could mainly occur in the northern part of the red box area used for the analysis and deep convective clouds could mainly occur in the southern part. Increasing aerosols suppresses shallow convection, which would be different from the story that the authors described in the paper.**

**Response:** We thank the reviewer for very helpful comments. The background circulation pattern at 500 hPa is characterized by ridge in north and trough in south over Asia (Figure R1). This pattern is favorable for persistent meeting between cold air from the north and warm moist air from Bay of Bengal and South China Sea, resulting in intensive convergence near the surface (Figure 1b) and torrential rainfall over Guangdong Province. The cloud top temperature average over the land in domain 2 is lower than −15 °C almost everywhere with minimum reaching about −35 °C (Figure S1b), indicating strong convection. Moreover, cloud ice, over the region with both decreased and increased parts, extends up to 16 km shown in Figure S11 and Figure S8, respectively. Further inspection of cloud evolution within the red box shows that the cloud regimes are consistent within the increased area used. We divided the red box area in 22°–24°, 112°–115° into a north box in 23°–24°, 112°–115° and a south box in 22°–23°, 112°–115°. Shallow clouds occur in both the northern and southern parts of the red box area (Figure R2). Figure R3–R6 show the differences in microphysical and dynamic variables due to aerosols. Their similar patterns in Box_N and Box_S suggest that the processes and related physical mechanism within the red box are consistent with each other.

Thanks for your comments regarding the mechanism of decreases in precipitation. We choose another region in 24°–25°N, 110.5°–112.5°E over the northwest corner of domain 2. The analysis is added in the paragraph six in the discussion section. Thank you again.

[Figure]

Figure R1. Spatial distribution of 3-day averaged 500-hPa wind (vector; unit: m s$^{-1}$) and height (shading; unit: m) during December 14–16, 2013 for (a) OBS from ERA-interim and (b) CTL from control simulation.

[Figure]

Figure R2. Time-height cross section of cloud fraction (CF; shading; unit: unitless) and PM$_{2.5}$ concentrations (contour; unit: µg m$^{-3}$) in (a) 23°–24°, 112°–115° (Box_N) and (b) 22°–23°, 112°–115° (Box_S) from control simulation. Dashed lines denote 0°C isotherm calculated as the averaged zero-layer height over the red box in Figure 3.

[Figure]

Figure R3. Differences with time (abscissa) and height (ordinate) in CF (shading; unit: unitless) and PM$_{2.5}$ concentrations (contour; unit: µg m$^{-3}$) between CTL and CLEAN (i.e. CTL minus CLEAN) for (a) Box_N and (b) Box_S. Only CF and PM$_{2.5}$ concentration anomalies that exceed 90% significance level are depicted with shading and contour. Green dashed lines denote 0°C isotherm calculated as the averaged zero-layer height over the red box in Figure 3.

[Figure]

Figure R4. Differences with time (abscissa) and height (ordinate) in cloud droplet number concentrations (CDNC; shading, unit: $10^7 \, \text{kg}^{-1}$) and cloud effective radius (contour; unit: µm) between CTL and CLEAN (i.e. CTL minus CLEAN) for (a) Box_N and (b) Box_S. Only anomalies that exceed 90% significance level are depicted with shading and contour.

[Figure]

Figure R5. Differences with time (abscissa) and height (ordinate) in cloud droplet number concentrations (CINC; shading, unit: $10^5 \, \text{kg}^{-1}$) and ice cloud effective radius (contour; unit: µm) between CTL and CLEAN (i.e. CTL minus CLEAN) for (a) Box_N and (b) Box_S. Only anomalies that exceed 90% significance level are depicted with shading and contour.

[Figure]

Figure R6. Differences with time (abscissa) and height (ordinate) in vertical velocity (shading, unit: cm s$^{-1}$) and latent heating (contour; unit: 3 K d$^{-1}$) between CTL and CLEAN (i.e. CTL minus CLEAN) for (a) Box_N and (b) Box_S. Only anomalies that exceed 90% significance level are depicted with shading and contour.

**2. The authors did not present enough data to examine the things they claimed for. In a few places as detailed in the specific comments, the authors assumed the literature work applies well to this study without presenting the key results to prove the point. See specific comments #14, 18, 20, 21, and 22. There are many inaccurate or misleading statements. I noticed they are mainly related to the lack of expertise in cloud physics and weather area, such as #7, 10, 13, 18, 20, 21, and 22.**

**Response:** We thank the reviewer for the thoughtful and thorough comments and suggestions. The comments are addressed accordingly as follows.

*3. There are many inaccurate or misleading statements. I noticed they are mainly related to the lack of expertise in cloud physics and weather area, such as #7, 10, 13, 18, 20, 21, and 22.*

**Response:** Thanks for pointing this out. The descriptions are corrected based on your comments. Please see the corresponding responses below.

*Specific comments:*

**1. P2, the later part of the last paragraph discusses literature study about ACI, which does not include the most recent work on this topic from a Science article (Fan et al., 2018).**

**Response:** Thanks for this suggestion. The paper is cited, and corresponding descriptions are added (P3 L3–5).

**2. P2, L19: "the slowing autoconversion rate induced by aerosols forms airborne cloud droplets in clouds" is confusing. First, what is "airborne cloud droplets in clouds"? since it is in clouds, why call it "airborne"? second, how does autoconversion form cloud droplets?**

**Response:** Thanks for pointing this out. The descriptions are corrected in our revised manuscript (P2 L31–32).

**3. P5, L6, based on Fan et al. 2015, the factor used in the study is 0.3 (not 0.1).**

**Response:** We adjusted the factor to 0.1 from 0.3 in Fan et al. (2015) to represent the background situation as the emissions in 2010 is much higher than that in 2006, which is revised in the manuscript (P5 L9–11).

**4. P5, L12-14, not sure how IC, BC, and emissions were treated in both domains in both D1 and D2. If Dom1 and Dom2 are run at the same time, which means dom 1 provides IC and BC for domain 2, then how to change IC and BC in Dom 2 for D2? In addition, if emission does not change in Dom2, wouldn't the local effect be underestimated?**

**Response:** Yes, domain 1 and domain 2 were run at the same time. The IC for domain 1 and domain 2 were provided from MOZART data. In D1 experiment, the IC, BC, and emissions were kept as same with the control run simulation for domain 1. Meanwhile, the IC and emissions were scaled by a factor of 0.1 for domain 2. In D2 experiment, the IC, BC, and emissions were scaled by 0.1 for domain 1. The IC and emissions were kept as same with the control run at the same time. The impact of boundary conditions provided to domain 2 was treated as effect of aerosols from outside of domain 2.

**5. P5, L18-19: Which simulation is 10X based on?**

**Response:** This simulation was based on the control run, which is revised in the main text (P5 L21–22).

**6. Section 3.1: since there are 58 stations for PM2.5 measurements in Domain 2, why not use them to evaluate the control simulations since the aerosol property is important to aerosol impacts?**

**Response:** Thanks for your question. We agreed this suggestion. Figure R7–R8 show the spatial distribution and time series of $PM_{2.5}$ concentrations during December 14–16, 2013, respectively, based on observation and control simulation. Over the delta region, higher aerosol concentrations occur in mega cities, while lower concentrations appear over their surrounding areas. The model underestimates $PM_{2.5}$ concentrations in the first two days with a more

homogeneous pattern. This could be induced by either the relative coarse resolution of model or the pseudo surface (actually above the ground) due to model vertical layers design. The failure to get some hot spots near the estuary may be attributed to the uncertainty of emissions. In the time series, both the simulation and observation show a dramatically decreasing trend of $PM_{2.5}$ concentrations once the rainfall initiated. The model could generally replicate the spatial distribution and time evolution of $PM_{2.5}$ concentrations with some underestimation during the first two days. This bias may lead to an underestimation of the aerosol impact on rainfall.

The descriptions associated with the figure are added into the manuscript.

[Figure]

Figure R7. $PM_{2.5}$ concentration (unit: µg m$^{-3}$) average during December 14–16, 2013 for (a) observation and (b) control simulation. Colored circles denote in situ station locations.

[Figure]

Figure R8. Time series of $PM_{2.5}$ averaged over all the stations during December 14–16, 2013 for CTL (black) and OBS (red).

**7. P7, L20-22, this sentence is not justified. It could only because in the second day there were much larger in-cloud and below-cloud scavenging of aerosols due to much cloud and rain. The smaller aerosol effect in the first day can be a result of many factors particularly meteorological conditions, and the larger effect you see in the next day might not so related to the aerosols in the first day.**

**Response:** Thanks for this comment. The aerosols influence the cloud droplet number concentration and cloud effective radius during all the three days. However, the rainfall changes induced by aerosols start from the second day when rainfall peak happens. This suggests that the aerosol impact on rainfall is modulated by the meteorological conditions. Related description is revised in the revised manuscript (P8 L4–6).

**8. Figure 4, how is the cloud fraction calculated? Is the difference in percentage or absolute difference?**

**Response:** The cloud fraction parameterization in the model follows Randall (Hong et al., 1998). The cloud fraction was calculated as the sum of cloud water, cloud ice and snow. The differences in Figure 4 are the absolute differences.

**9. P7, L25-28, Better to use percentage differences or both in terms of quantifying the accumulated rain.**

**Response:** Agreed. The rainfall differences in percentage are added in the revised manuscript (P8 L10–12).

**10. P8, L6-11, the whole description here has a problem. The way it describes currently basically says that aerosols are the reasons responsible for the more and deeper clouds at later time and less and shallower clouds at the earlier time, which should not be true. The first order is the meteorological conditions that are responsible for the cloud amount and vertical distribution. On the top of it, aerosol may influence it, and then you can describe the influence in more quantitative way.**

**Response:** Thanks for pointing this out. The description is revised in the main text (P8 L23–27).

**11. Figure 5, the figure caption needs to be consistent with the figure label. If you label your panels in a, b, c,.., you need describe your figures in the same way so that readers can follow. This comment apply to many other figures. Also, the caption already has too many acronyms while another acronym CI for contour interval is used here, which only causes poor readability and confusion here. About the differences in CINC and ice effective radius, did you only consider cloud ice crystals, or all of the ice-phase particles were considered?**

**Response:** Sorry for the inconsistency. The captions are revised correspondingly. Only cloud ice is considered here.

**12. P 8, L15, when you say "dramatically", you need to give a quantitative value. Figure 5 only shows the absolute differences, which has the maximal value at the magnitude 80 cm-3 based on the legend. This change is not large unless you did a domain average and there are many cloud-free points in the analysis domain.**

**Response:** We carefully checked the calculation and updated the results accordingly. As shown in the legend, the maximal value is $8 \times 10^7$ kg$^{-1}$ ($8 \times 10^7$ [$10^3$ g]$^{-1}$=$8 \times 10^7$ [$10^3$ cm$^3$]$^{-1}$), which is equal to $8 \times 10^4$ cm$^{-3}$. The magnitude of this value is comparable to that in Zhong et al. (2015). In percentage, the cloud droplet number concentration has increased by 5.5 times. Descriptions are revised in the main text.

**13. P 8, L20, what do you mean about "the interim processes"?**

**Response:** The interim processes refer to that more cloud droplets are lifted to freeze into ice clouds. Our further analysis on source of latent heat is not attributed to freezing. The corresponding descriptions are deleted in the main text.

**14. P8, L17-27, the entire description here about the ACI effect is not about the results from their study. The authors just followed what the literature describes. First, the description and the corresponding references do not reflect the symbolic literature studies on ACI on deep convective clouds. First, the idea of convective invigoration by enhanced latent heat from cold-phase processes (due to suppression of warm rain) starts from Andreas et al. Science, 2004 (obs), then Khain et al., QJR, 2005 (model), Rosenfeld et al. Science, 2008 (theoretical), Fan et al., JGR, 2009, etc, did detailed studies about it. The authors did not mention these studies at all (Rosenfeld et al., 2008 is not discussed in an appropriate way since it is the theoretical study for this theory). Second, the most recent development of ACI is the "warm-phase invigoration" in Fan et al. Science, (2018) where latent heat release from enhanced condensation is emphasized as a reason for the enhanced updraft speed. From Figure 5, the latent heat enhancement peaks below 8 km altitudes and there is a peak at 3-5 km, suggesting condensational heating might play a significant role here as well. The latent heat enhancement at low part of clouds from condensation plays a much more significant role than the correspondent at high levels as shown in Fan et al. 2018. The authors need to examine this in detail to understand what's the real reason behind it instead of just citing some literature studies since ACI is a key point of the study.**

**Response:** Thanks for your constructive and insight comments. Following Fan et al. (2018), the latent heat released from condensation, deposition, and freezing during cold and warm cloud processes are diagnosed by rerunning the model (Figure R10). The rimming processes are included into the freezing. It is nothing to do with the freezing which means the precipitation enhancement with aerosols cannot be simply attributed to cold cloud invigoration effect due to freezing.

Figure R9 shows the changes in the mass and number concentration of the different hydrometers. The aerosols are activated to form more cloud droplets on which water condenses and produces more cloud water (Figure R9a). This process releases additional latent heat at 3–5 km due to condensation (Figure R10a) and lower supersaturation, which is also discussed in Fan et. al (2018). The smaller radius of cloud droplet shown in Figure 5a is not favorable to fast droplet coalescence and suppress warm rain. The precipitation decreases from 15Z to 20Z on 14 December (Figure S4). With aerosols, the precipitation is increased between 03Z on December 15 to 10Z on December 16. However, the changes in the hydrometers, particular for rain water, and sources of latent heat release are quite different between before and after 15Z on December 15. These differences indicate that the processes and their related mechanisms may differ from each other. In the first stage, before 15Z on December 15, there are abundant ice crystals (i.e. snow and graupel) above the 0°C isotherm around 5 km (Figure S8). With the presence of ice crystals, water vapor deposition is prior to happen on ice surface as the saturation with respect to water is supersaturation with respect to ice. As this occurs, the environment becomes unsaturated to water, resulting in the evaporation of liquid water. This is known as the Bergeron-Findeisen-Wegener theory. Correspondingly, the ice crystals (i.e. cloud ice, snow, and graupel) increase at the expense of rain water. Note the magnitude of snow and graupel mass is ten times of that of rain water. The latent heat release due to deposition in cold cloud is stronger than that due to condensation in warm cloud even though the latter is also important. After 15Z on December 15, most of the ice crystals fall as precipitation. Compared with depositional heating, the condensational heating plays a dominant role in intensifying convective strength. The rain water increases through accretion of added cloud droplets, leading to precipitation increases.

The corresponding figures and discussion are revised in the main text.

[Figure]

Figure R9. Differences with time (abscissa) and height (ordinate) in (a) cloud water (shading; unit: $10^{-5}$ kg kg$^{-1}$) and CDNC (contour; unit: $10^7$ kg$^{-1}$), (b) cloud ice (shading; unit: $10^{-5}$ kg kg$^{-1}$) and CINC (contour; unit: $10^4$ kg$^{-1}$), (c) rain (shading; unit: $10^{-5}$ kg kg$^{-1}$) and rain number concentration (contour; unit: $10^5$ kg$^{-1}$), (d) snow (shading; unit: $10^{-4}$ kg kg$^{-1}$) and snow number concentrations (contour; unit: $10^3$ kg$^{-1}$), and (e) graupel (shading; unit: $10^{-4}$ kg kg$^{-1}$) and graupel number concentration (contour; unit: $10^3$ kg$^{-1}$) between CTL and CLEAN (i.e. CTL minus CLEAN) averaged over the red box. Only anomalies that exceed 90% significance level are depicted with shading and contour.

[Figure]

Figure R10. Differences with time (abscissa) and height (ordinate) in latent heat release (unit: K d$^{-1}$) from (a) condensation, (b) deposition, and (c) freezing processes between CTL and CLEAN (i.e. CTL minus CLEAN) averaged over the red box for the warm cloud. (d–f) Same as (a–c) but from cold cloud. Only anomalies that exceed 90% significance level are depicted with and contour. Zero-value contour lines are omitted, and negative values are dashed. The contour interval is 3 K d$^{-1}$. Note the blank represent the values are within 3 K d$^{-1}$.

**15. P8, Eq (1), where is the horizontal advection terms for the moisture budget? In the model, this is an important term. If you considered it in the vertically integrated moisture flux (MFC) convergence in your calculation, then the MFC should be large at the convection permitting scale.**

**Response:** Agreed. We integrated the moisture flux convergence (MFC) in vertical direction. As discussed in the manuscript, this term dominates the rainfall changes. The MFC term is further divided into two terms as

$$-\frac{1}{g}\int_0^{P_s} \nabla \cdot \left(q\overrightarrow{V_h}\right) dp = -\frac{1}{g}\int_0^{P_s} q\nabla \cdot \overrightarrow{V_h}\, dp - \frac{1}{g}\int_0^{P_s} \overrightarrow{V_h} \cdot \nabla q\, dp$$

where the first term on the right side is the horizontal moisture convergence (hereafter CON); the second term is the horizontal advection of water vapor (hereafter ADV).

As shown in Figure R11, the CON term dominates the contribution to total MFC. The resemblance of pattern between MFC and CON suggests that the increase in rainfall is mainly driven by CON changes. The descriptions associated with the figures are added in the main text.

[Figure]

Figure R11. Differences in column-integrated flux convergence (MFC; shading; unit: mm) and moisture flux (vector; unit: kg m$^{-1}$ s$^{-1}$), between (a) CTL and CLEAN (i.e. CTL minus CLEAN) and (b) ARIoff and CLEAN (i.e., ARIoff minus CLEAN) on December 15. (c, d) Same as (a, b) but for column-integrated moisture convergence (CON; unit: mm). (e, f) Same as (a, b) but for column-integrated advection of water vapor (ADV; unit: mm). The numbers at the top-left corner of each panel represent the values averaged over the red boxes. The red boxes (22°–24° N, 112°–115° E) denote the analysis region.

**16. P9, L8, the figure number is wrong. Also, where is the moisture coming from the northerly wind since northerly wind generally brings in drier air? It would be good to show the spatial distribution of moisture field.**

**Response:** Thanks for pointing this out. Yes, the air from the northerlies is drier which is shown in the spatial distribution of water vapor (Figure 20c and 20d). It may thus not correct to claim the moisture from the northerly wind. As shown in Figure R11, the MFC term is dominated by CON term which depends on the convergence field rather than the moisture. The convergence is attributed to the microphysics-dynamics feedback discussed in the manuscript. The statement is revised in the main text (P10 L25–26).

**17. P 9, L11-12, since there is compensation effect here, a figure for ARI effect should be shown to quantify how much is the compensation effect.**

**Response:** The ARI effect is included in the revised figure (Figure 8).

**18. P9, L16-20, again, key processes are not shown and the summary description might not be accurate. First, it is not correct to say "water clouds ascend to freeze into ice clouds" since it is just that more cloud droplets are lifted to the higher levels and form more ice**

**particles. Second, as I pointed out above, the source of latent heat enhancement is not examined and the authors just assumed it is mainly resulted from more droplet freezing. Third, the much enhanced horizontal convergence could be gradually induced by other feedback such as precipitation or radiation since the simulation duration are a few days, not just a few hours. Another question is that how the changes in domain 1 impact the results over domain 2?**

**Response:** Thanks for the comments. Based on Figure R10, the source of latent heat is mainly induced by deposition before 15Z on December and condensation after 15Z when the precipitation is increased with aerosols. Although the simulation duration is a couple of days, the precipitation increases with aerosols only occur between 06Z on December 15 to 10Z on December 16. Moreover, the persistent convective system makes the impact last for longer time. Strong latent heat is released during this period, and ARI has a little impact on the increased precipitation. These results drive to conclude that latent heat release is the main reason for enhanced horizontal convergence.

With aerosol emissions in domain 1, the aerosols are emitted or formed. The aerosol concentration is transported to domain 2 through lateral boundary conditions.

**19. Section 3.3, the remote and local aerosol effects can strongly depend on how strong the coupling between the two domains. With the two domains running together, the coupling is very strong and the Dom 1 keeps updating Dom 2, which could lead to very strong effect from any variable in Dom 1(not just aerosol). If you run domain 2 separately with the IC and BC updated in every 3-hours or 6 hours from Dom 1, and do the same studies, the results could be changed.**

**Response:** Thanks for the comments. It is correct that running domain 2 separately would change the results, but it does not reflect the real situation. In reality, atmosphere does not have any domain, and should be highly connected. To reflect the real situation, the domain 1 and domain 2 should be online coupled by running them together. In addition, following the commonly used approach, the results of outermost ten grid points of each boundary of domain 2 are excluded to minimize the influence from the lateral boundary conditions.

**20. P10, L28-29, this may indicate secondary droplet nucleation, meaning activating enormous smaller aerosols at higher-levels due to higher supersaturation. Without looking at it carefully, you can not just assume it is mainly because of ascent of cloud droplets.**

**Response:** Thanks for your constructive comments. We agree your opinion that the increase of cloud droplets at 1.5–4 km cannot be attributed to ascent motion as the vertical velocity is reduced in the 10× simulation. With ten-time changes in aerosol emissions, more aerosols are activated to form cloud droplets at higher level due to higher supersaturation. The consumption of moisture and energy limits the formation of low cloud. The content is revised accordingly. Thank you.

**21. P11, L1-6, again, you can not just guess by citing a literature work assuming it apply to your study. Key results need to be shown. The reduction of low-level cloud could just because more deep cloud form consuming moisture and energy which would limit the formation of other type of clouds. Evaporation and sublimation have to come from clouds. In addition, the lower-level cloud and the high-level clouds shown here might not be**

**vertically connected over the domain. For example, shallow clouds could mainly occur in the northern part of the red box associated with cold front and deep clouds could mainly occur in the southern part associated with the convergence zone. Increasing aerosols suppresses shallow convection, which would be different from the story you describe here now. It would not be nothing to do with sublimation if that is the case.**

**Response:** Thanks for your comments. We agree that the low cloud reduction is because of the consumption of moisture and energy due to formation of high-level cloud. The cloud regimes are quite consistent in the northern and southern parts of the red box as shown in Figure R2–R6. Figure R13 shows the latent heat release due to condensation, freezing, and deposition for both warm and cold cloud. Deposition is the most important factor while freezing play a negligible role in this case. The strong latent heat released from deposition is consistent with the snow increase from 00Z to 12Z on 15 December. The underline mechanism is related to the Bergeron-Findeisen-Wegener theory as discussed in the responses to comment 14 but with a much stronger magnitude. However, after 15Z on December 15, the changes in rain water mass and latent heat in 10× are quite different from that in control simulation. We agree that our previous description may not fully reflect the mechanism. The reason is thus discussed in the revised main text. Thank you for your comment.

[Figure]

Figure R12. Differences with time (abscissa) and height (ordinate) in (a) cloud water (shading; unit: $10^{-5}$ kg kg$^{-1}$) and CDNC (contour; unit: $10^{7}$ kg$^{-1}$), (b) cloud ice (shading; unit: $10^{-5}$ kg kg$^{-1}$) and CINC (contour; unit: $10^{4}$ kg$^{-1}$), (c) rain (shading; unit: $10^{-5}$ kg kg$^{-1}$) and rain number concentration (contour; unit: $10^{5}$ kg$^{-1}$), (d) snow (shading; unit: $10^{-4}$ kg kg$^{-1}$) and snow number concentrations (contour; unit: $10^{3}$ kg$^{-1}$), and (e) graupel (shading; unit: $10^{-4}$ kg kg$^{-1}$) and graupel number concentration (contour; unit: $10^{3}$ kg$^{-1}$) between 10× and CLEAN (i.e. 10× minus CLEAN) averaged over the red box. Only anomalies that exceed 90% significance level are depicted with shading and contour.

[Figure]

Figure R13. Differences with time (abscissa) and height (ordinate) in latent heat release (unit: K d$^{-1}$) from (a) condensation, (b) deposition, and (c) freezing processes between 10× and CLEAN (i.e. 10× minus CLEAN) averaged over the red box for the warm cloud. (d–f) Same as (a–c) but from cold cloud. Only anomalies that exceed 90% significance level are depicted with and contour. Zero-value contour lines are omitted, and negative values are dashed. The contour interval is 3 K d$^{-1}$. Note the blank represent the values are within 3 K d$^{-1}$.

**22. P11, L8-17, do not agree with some of the discussion. Compared with Figure 3, I only see the corresponding increase and decrease in the Dom 2 become more significant in Figure 12. The authors did not discuss why there are two significantly different precipitation response regimes to the change of emissions. It seems that they are located in different dynamic regimes so have different cloud types. More detailed description about what types of clouds were formed in the cold side is needed in the description of the case at the beginning of the result section. It would provide basis for the related discussion after that.**

**Response:** Thanks for pointing this out. We choose a box located over the area (24–25°N, 110–112°E) where precipitation is decreased. The corresponding analysis are put in the paragraph six of discussion. The cloud types in domain 2 are also discussed in the beginning part of the result section.

**23. Section 4: a. First paragraph, Summary should include description of what have been done as well.**

**b. Second paragraph, see my comment above about how to look at different aerosol impacts on different cloud regimes/types. The current discussion might not relevant because the cloud types should be very different between the code and warm sides of the frontal system.**

**c. The third and fourth paragraphs may need to be changed accordingly after my relevant comments above are addressed.**

**Response:** Yes, these paragraphs are thus revised accordingly in the revised main text.

**Grammatical problems: P4, L19: grammar error. P7, L6: grammar error. P7., L32, past tense is not needed here. There are many places in results section that have the mixed past and current tenses. Better to be consistent in tense to improve readability and avoid confusion.**

**Response:** Thanks for pointing this out. The grammar errors are corrected. The result section is revised with current tense to keep consistency.

**Reviewer 2:**

This study employed the WRF-Chem model and a series of sensitivity experiments to study the aerosol microphysical and radiative impacts on a historical heavy precipitation event in southern China. The effects of local and remote aerosols are compared by altering aerosol concentrations in different domains. The finding about the aerosol invigoration effect with a moderate aerosol increase generally agrees with the existing argument that aerosols tend to induce more extreme precipitation. The topic of study is important and fits with the scope of ACP very well. However, there are still lots of unclear writing and insufficient analyses in the manuscript. Major revisions are needed before it can be accepted by ACP.

**Two major comments:**

**1) It is kind of surprising to me that the simulated aerosol properties and spatial distributions are not shown in the manuscript. They are actually about the strength of the WRF-Chem model in doing aerosol-cloud research. PM2.5 is plotted, but it is not quantitative index for either CCN effect or radiative effect. The spatial distributions are critical for us to understand the potential influence of remote aerosols. The aerosol chemical component determines the aerosol radiative properties, absorbing or scattering, as well as CCN ability.**

**2) Process-level analyses on ACI and ARI in this case should be strengthened. For ARI, I do not see any analysis on the radiative fluxes, temperature field, and associated dynamical adjustment. Is there any atmospheric heating due to black or brown carbon in this case? For ACI, the microphysical properties of all hydrometeor and co-varying water vapor field should be studied. See more specific comments below.**

**Response:** We thank the reviewer for the through and thoughtful comments. We tried our best to address all concerns and have revised the manuscript accordingly. Hope you find our revisions useful. Thank you very much.

For your questions, we only considered black carbon for atmospheric heating in our simulations because there is lack of reliable parameterization for brown carbon in our study region. The analysis was focused on the ACI impact because of its dominant role in this case.

**Specific comments:**

**1) I feel the literature review in the introduction part is not done thoroughly. Considering both ACI and ARI have been extensively investigated for the past 10-20 years, more credits should be given to the studies with the similar topic.**

- **P2L4, inaccurate statement. Actually, there are lots of existing studies on the influence of aerosols on different types of extreme weather, such as tropical cyclone (Wang et al., 2014, Nat. Clim. Change; Zhao et al., 2018, GRL), hail storm (Ilotoviz, et al., 2016, JAS), etc.**
- **P3L1, the competition between ARI and ACI has been widely discussed on both cloud- resolving scale (Lin et al., 2017, JAS; Wang et al., 2018, AAS) as well as regional climate scale (Wang et al., 2016, JGR).**

- **P3L3-5, different aerosol types can be a critical factor as well to determine the invigoration or suppression effect of aerosols (Jiang et al., 2018, Nat. Commu.).**

**Response:** Thanks for pointing this out. The studies are cited and a more through literature review are added into the introduction part.

**2) P3L15-20, it is not clear what are hypotheses for the different effects from local and remote aerosol emissions? Different concentrations, chemical compositions, or spatial distributions? What did observations tell us about their differences? Without stating those explicitly, readers fail to follow the logic flow of the paper.**

**Response:** We thank the reviewer for pointing this out. The different effects from local and remote aerosol emissions refer to different aerosol concentrations. Figure R14 shows the spatial distribution of aerosol optical depth during December 13–16, 2013. The values are missing over the southern China because of the mask effect of cloud. The aerosol optical depth is higher than 1 over north-eastern China, indicating strong air pollution. Given the wind pattern in Figure S1b, the aerosol concentrations over local region could be from either local emission or transport by monsoonal flow. As shown in Figure 10, the aerosol concentrations from local aerosol emissions accumulate near the surface decrease dramatically once the peak rainfall initiated. By contrast, the aerosols from transport extend a higher altitude in the atmosphere and last for much longer time. These statements are added in the main text.

[Figure]

Figure R14. Spatial distribution of averaged aerosol optical depth at 483 nm from OMI during December 14–16, 2013 in 10°–40°N, 90°–130°E.

**3) Fig. 2b and 2c, rather than only showing the dots over the stations, I suggest to plot the rainfall map over the whole domain for model and satellite, which is helpful to characterize the system. You can still keep open circles to compare the rainfall over each station.**

**Response:** Thanks for your suggestions. The rainfall map over the whole domain is aded in the revised manuscript (Figure 2).

**4) Fig. 1b, the photo here deliver very litter information. Suggest to replace by the wind pattern analysis . Also, P6L20, a more accurate expression here is "monsoon system" or "monsoonal flow".**

**Response:** The photo is replaced by the suggested wind pattern figure, while the descriptions are also corrected.

**5) P6L29-31, why not using TRMM which is better at heavy precipitation? What is the point to show a satellite product even worse than the model?**

**Response:** The CMORPH data was used in this work because of its higher spatial and temporal resolutions (i.e. 8 km and 30 mins) than those of TRMM. The resolution is comparable to that of model output (i.e. 4 km and 1 hour, respectively) and rain gauge data. The finer temporal resolution allows us to check the aerosols' effect at the peak time of the study event, which was previously discussed in the literature. In addition, the CMOPH data was also used in recent studies (e.g. Zhong et al., 2015) to evaluate the model performance on extreme rainfall cases.

To have a better understanding about the two data sets, we conducted a comparison between them. Figure R15d shows the TRMM data for this case. The TRMM data shows a better performance over south-western Guangdong Province and western Guangxi Province, in which CMORPH may underestimate precipitation. For precipitation along the coast and over the Pearl River Delta region, even though TRMM's performance is better than CMPRPH's, TRMM also shows an underestimation.

Overall, as explained above, CMORPH was used due to its higher spatiotemporal resolutions and would like to use the similar dataset used in previous studies to provide a fair comparison with literature. The discussion is added in the revised manuscript. Thank you.

[Figure]

Figure R15. Spatial distribution of accumulated precipitation (unit: mm) from 00Z on December 14, 2013, to 00Z on December 17, 2013 from (a) station observations (OBS), (b)

CMORPH, (c) control simulation (CTL), and (d) TRMM. Circles denote locations of in situ observations.

**6) The physical meaning of ARIoff - CLEAN is not obvious, as the authors use (CTRL - CLEAN) - (CTRL - ARIoff) to approximate ACI. I suggest the authors state this assumption explicitly and use ACI to replace ARIoff - CLEAN for all figure legends. Also, be careful about the usage difference between hyphen and minus sign.**

**Response:** Thanks for pointing this out. As suggested, ARIoff − CLEAN is replaced by the term ACI in all figures' legends and captions. The hyphen and minus sign are distinguished in all figure legends.

**7) It is unclear for me how the statistical analysis is conducted in those figures. As I understand, the authors only have one run for each model configuration. How to get the sufficient samples for the Student's t-test at each grid point?**

**Response:** As mentioned in P5 L4–6, to isolate robust signals from model natural variations, five ensemble members with a perturbed initial time at 3-h intervals were conducted for each experiment. The significance level was calculated based on the five ensemble members.

**8) P8L15-25, the authors only mentioned about the latent heat from droplet freezing. However, according to Fig. 5, clearly there is a significant portion of latent heat release below 4 km (warmer than 0 degree C). Can you plot the changes in liquid water content to confirm it? For the oceanic DCC, aerosol induced diabetic heating has two peaks, one in the warm portion and one in the mixed-phase portion (Fig. 3a of Wang et al., 2014, Nat. Commun.). Another interesting point here is that the Morrison microphysical scheme in WRF-Chem uses the simple water vapor saturation adjustment for condensation/evaporation. I speculate that this scheme cannot account for CCN effect in fostering condensation. Since this paper indirectly infers more liquid water forms, it is intriguing to see why.**

**Response:** Thanks for pointing this out. We agree that the condensational heat below freezing level also plays an important role. The source of latent heat is found to be not related to droplet freezing. Figure R16 shows the differences in cloud water and cloud ice induced by aerosols. The liquid water content increases below $0°C$ during almost all the period. The cloud ice also increases when the rainfall peak happens. The latent heat from microphysical processes are further divided into three parts from condensation, deposition, and freezing for warm cloud and cold cloud (Figure R17). Note the rimming processes are included into the freezing. Aerosol induced diabatic heating also has two net heating peaks in this case. However, the peaks are much higher than that in Wang et al. (2014) for oceanic deep convection, and just slight cooling occurs due to melting in warm cloud (Figure R17c). The net heating peaks are attributed to condensation in warm cloud and deposition in cold cloud at the height of 3 km and 7 km, respectively. In CLEAN experiment, fast coalescence forms warm rain and reduces the integrated droplet surface area, leading to supersaturated clouds. With aerosols, additional number of cloud droplets are nucleated (Figure 5a) on which water vapor condenses. This is consistent with Fan et al. (2018). Contents in the main text are revised accordingly (P9 L4–20). Thank you very much for your comment.

[Figure]

Figure R16. Differences with time (abscissa) and height (ordinate) in (a) cloud water (shading; unit: $10^{-5}$ kg kg$^{-1}$) and PM$_{2.5}$ concentrations (contour; unit: μg m$^{-3}$) (b) cloud ice (shading; unit: $10^{-5}$ kg kg$^{-1}$) and PM$_{2.5}$ concentrations (contour; unit: μg m$^{-3}$) between CTL and CLEAN (i.e. CTL minus CLEAN). Only cloud water, cloud ice, and PM$_{2.5}$ concentration anomalies that exceed 90% significance level are depicted with shading and contour. Green dashed lines denote 0°C isotherm calculated as the averaged zero-layer height over the red box in Figure 3.

[Figure]

Figure R17. Differences with time (abscissa) and height (ordinate) in latent heat release (unit: K d$^{-1}$) from (a) condensation, (b) deposition, and (c) freezing processes between CTL and CLEAN (i.e. CTL minus CLEAN) for the warm cloud. (d–f) Same as (a–c) but from cold cloud.

Only anomalies that exceed 90% significance level are depicted with and contour. Zero-value contour lines are omitted, and negative values are dashed. The contour interval is 3 K d$^{-1}$. Note the blank represent the values are within 3 K d$^{-1}$.

**9) Fig. 4,7, cloud fraction is about cloud macrophysics, which may not accurately reflect changes in cloud microphysics (water content, number concentration). The latter are more relevant with the aerosol invigoration effect. As mentioned above, I strongly suggest the authors plot and systematically analyze the changes in the mass and number concentration of the different hydrometeors.**

**Response:** The cloud fraction calculation in our model follows Randall (Hong et al., 1998) with value range from zero to one. Their values were calculated as the sum of cloud water, cloud ice and snow, which actually was based on mass. We chose cloud fraction in Figure 4 and Figure 7 because this variable is an indicator of mass for both liquid and ice clouds. The changes in the mass and number concentration of different hydrometeors are also analysed in Figure 6 and Figure 7 in the revised manuscript.

---

## Author Response (AR2)

**Response to Reviewer #1:**

We appreciate the reviewer who reviewed the manuscript carefully and provided insightful follow-up comments. We have tried our best to address all concerns and revised the manuscript accordingly. The comments are in normal font. A point-by-point response is listed as below in bold italics.

1. The authors missed the point of my first major comment. My comment is about the regions with opposite precipitation response, which could be corresponding to the cold and warm sectors of the convective system, respectively. In their response, the authors chose two narrow areas (Box_N and Box_S), both of which have increased precipitation, to show the consistent features between those two boxes. Both areas are in the convergence zone based on Figure R1 and their cloud properties are of course similar. The cold section is probably northwest or northeast where decreased precipitation is seen (can be identified based on temperature field). Clouds at the cold sector would not be invigorated by aerosols so decreased precipitation can be seen as a result of suppressed conversion into rain or snow. Again, the point is that the authors need to explain the opposite precipitation response for different sections of the system, particularly for the 10X run, the decrease of precipitation is in a similar magnitude with the increase and occupies half of the simulation domain (Figure 19b). Based on Figure 19b, there is really no justification of only picking up the red box region to study.

***Response: Thanks for the comments. We are sorry that we didn't make the responses clear in 1st version of the revised manuscript.***

***The mechanism of precipitation decrease over another region has been investigated in the discussion of main text along with Figure S11–S15, which is described as follows (P16L26–P17L8):***

***The mechanism of precipitation decreases over another region, in 24°–25°N, 110°–112°E, is also investigated. Figure S11 shows the distribution of time-height mass and number concentrations of different hydrometeors averaged over this region from CTL run. There are lots of ice crystals with cloud ice extending up to 16 km, indicating strong deep convection, which is consistent with low cloud top temperature in Figure S1b. However, the cloud base is higher than that over the region denoted by the red box, characterized by smaller low-level cloud water on 15 Dec when strong aerosol impact occurs. This can also***

*be indicated from the surface temperature field (Figure RR 1), characterized by a dipole with low in northwest and high in southeast. With aerosols, more cloud droplets nucleated on which water can condensate. Additional cloud water is subsequently formed near to 4 km (Figure S12a), accompanied by reduced supersaturation. The reduction of rain water and ice crystals (particularly in graupel) suggest that both the warm rain and cold rain are suppressed. Less latent heat is released dominated by condensation in warm cloud and deposition in cold cloud. There could be three reasons for this. The first one is that the mass of water vapor is small over this region in the northwest corner of the domain, so that not enough water supply for convective invigoration effect with aerosols. The second one is related to the very strong wind shear over this region with maximum value up to 80 m s⁻¹. This condition is unfavored for latent heat to accumulate, which is key factor to convection strength (Fan et al., 2009). In addition, the cold cloud bases may suppress convection and precipitation due to strong evaporative cooling and less efficient ice crystals formation (Fan et al., 2016). Thus, the precipitation is suppressed over this region with aerosols. With ten times of aerosol emissions, the mass and number of rain water and ice crystals are further reduced, accompanied by weaker latent heat release (Figure S14 and S15). As a result, the precipitation is further suppressed (Error! Reference source not found.b).*

[Figure]

*Figure RR 1. Spatial distribution of surface temperature (unit: K) on (a) December 14 and (b) December 15 in 2013 in the CTL run.*

2. The authors did not do a neat job in responses. Many responses have wrong line numbers and they also did not describe what changes they made (also did not copy the revised text to the responses), which made me have a hard time to check their changes.

*Response: Sorry to bring the troubles. The line numbers of responses are corrected and corresponding changes made in the manuscript are described in the responses to your comments.*

3. There are quite a bit misunderstandings of cloud microphysical processes by reading the responses only (since I had a difficulty to find the changes in the manuscript due to incorrect line numbers). Here are examples, (1) a mistake in calculating cloud droplet number concentrations. They got unreasonably high ($8e^4$ $cm^{-3}$) cloud droplets (particularly for area mean, not a maximum value at gird-level) by using water density instead of air density to convert to number concentrations. What's surprising me is that they still argue the reasonability of it. Such a high number concentration is only possible for aerosols (not droplets) in a very polluted condition. (2) the misunderstandings of BF process, latent heat, and precipitating particles (see my comments on #14 response below). (3) the primary driver of convergence (my comments on #16 response). All these aspects that they misunderstood are the key aspects for analyzing and interpreting the model results this study.

*Response: (1) Thanks for pointing this out. We should use the air density rather than water density to convert the unit of number concentration from $kg^{-1}$ to $cm^{-3}$; (2) and (3) please see the response below.*

4. For many comments on clarifications, the authors responded but did not clarify in the manuscript, such as comment #4.

*Response: Per your suggestions, we clarify the responses to comments #4, # in the main text.*

5. The writing is a little sloppy. There are typos and many statements are confusing. Here are a few just in a short abstract:

*Response: Thanks for pointing this out. Please see the response below accordingly. We tried our best to correct other typos and misleading statements.*

Abstract:

Line 28, "cloud property changes also resembled that in the control run" does not make sense. Changes means the differences between the 10× run and control run, how can the changes resemble control run?

*Response: We revised the description as:*

*"Compared with CLEAN experiment, the precipitation and cloud property changes in 10× run also resembled that in the control run, but with much greater magnitude."*

Line 29, "The precipitation average over Guangdong province decreased by 1.0 mm but increased by 1.4 mm in the control run" does not make sense either. Looks like you are describing an increase or decrease in the control run. Then what are you comparing with? Generally, the description should be the increase or decrease by comparing with the control run.

*Response: Sorry for the confusing. The comparison made here is between CTL and CLEAN run. The control run in this study is chosen as real case. The statement has been revised as:*

*"With aerosols, the precipitation average over Guangdong province decreased by 1.0 mm but increased by 1.4 mm in the control run by comparing with CLEAN run"*

Line 30, "reinforced" should be removed. Also, downsteam of what? Urban city or aerosol source?

Last sentence in Abstract: Be specific about "the cloud invigoration effect", which is different from convective invigoration. Cloud invigoration refers to larger and/or taller clouds. Convective invigoration refers to stronger storm intensity which usually leads to more extreme rain, more lightning, etc.

*Response: Per you suggestions, reinforced is delete. Sorry for the confusing, we mean downstream of aerosol source.*

*Thanks for explaining the differences. We change the term to convective invigoration.*

6. Detailed comments on responses

(1) #4 response: the description in the manuscript is still confusing. In the manuscript, you said BC is also scaled by a factor of 0.1 for domain 2. Since BC for domain 2 should be from domain 1 simulation, how can you scale it? About "In D2 experiment, the IC, BC, and emissions were scaled by 0.1 for domain 1. The IC and emissions were kept as same with the control run at the same time", Isn't the second sentence contradicted with the first one? I am still confused about what you wanted to say in the second sentence.

*Response: Yes, the BCs is only applied for domain 1 and this is a typo error. In D2 experiment, the IC, BC, and emissions were scaled by 0.1 for domain 1. The IC and emissions were kept as same with the control run for domain 2. Sorry for the confusion. The statement has been corrected in the main text (P5 L25–30).*

(2) #8 response: Need to clarify in the manuscript (such as in the figure caption).

*Response: Per your suggestions, the response to comment #8 has been clarified in both the caption and the main text (P8 L32–33).*

(3) #10 response: Line number is not correct so it is difficult to identify the text you revised for this comment. But I found there is a mistake in P8 Line 24, how can the cloud top for deep convection only extends up to 1 km?

*Response: Sorry for the incorrect line number. The description has been revised as (P8 L17–19):*

*"Distinct effects of aerosols appear during the second day when the rainfall peaked (Error! Reference source not found.d), although aerosols lead to more cloud droplet number concentration associated with smaller radius on the first day (Figure 5a); this suggests that the effects of aerosols on precipitation are modulated by other factors (e.g. meteorological conditions)."*

*Yes, thanks for pointing this out. The mistake has been corrected. It should be 14 km.*

(4) #11 response: Need to clarify in the figure caption that only cloud ice is considered.

*Response: Per your suggestions, the response to comment #11 has been clarified in the caption in Figure 5.*

(5) #12 response: The authors made a mistake in calculating the droplet number concentrations. They used water density (1 g cm$^{-3}$) instead of air density ($\sim 1e^{-3}$ g cm$^{-3}$ at low levels) for the calculation. The area mean value should $\sim$ 80 cm$^{-3}$ as I mentioned in the previous round, not 8e4 cm-3 that is not totally reasonable.

*Response: Thanks for correcting this mistake. We should use the air density rather than water density to convert the unit of number concentration from kg$^{-1}$ to cm$^{-3}$.*

(6) #13 response: I do not understand how more cloud droplets are lifted to freeze can be named as "interim processes". Why not directly describe the process instead of using a term that is not known?

*Response: Thanks for your suggestions. As we found the source of latent heat cannot be attributed to freezing, the description has been removed in the main text.*

(7) #14 response: there are a few fundamental misunderstandings about cloud microphysical processes: (a) BF process. This process only occurs in the limited regime where Sw<0 but Si>0. In deep convection, most of updrafts are strong enough to make Sw>0. In that situation, both droplet and ice crystal will grow. In addition, this process only increases ice crystal mass, not ice crystals as authors claimed. (b) latent heat. The statements "the magnitude of snow and graupel mass is ten times of that of rain water. The latent heat release due to deposition in cold cloud is stronger than that due to condensation in warm cloud even though the latter is also important" have problems. It is conceptually wrong to discuss latent heat magnitude based on the mass for different phase of hygrometers. snow and graupel are not mainly formed from deposition. Riming is the process for graupel forming which converts a lot of liquid mass to solid phase. The latent heat release from riming may be small only because the latent heat release for converting per unit liquid to ice is only about 1/8 of that converting per unit of water vapor to liquid. I'd want to know in detail how you calculate latent heat for each process in the model. Currently it is just said "diagnosed". If it is diagnosed from the mass like described here. Then it is not correct. (c) It is also not correct to say "most of the ice crystals fall as

precipitation". Ice crystals would not fall as precipitation. Snow and graupel are the precipitating particles. (d) The figure R10 is confusing. How can warm cloud have deposition and freezing? How do you define warm clouds? Also, why not show the values below 3 Kd-1, which is significant in differences? Please clarify "anomalies that exceed 90% significance level". First, there is no observations so please define anomaly here. Second, how the significance test is done since data between two simulations cannot be compared in pairs in grid level because very different clouds could form. If the test is conducted based on mean values, are there enough data for such a test?

*Response: (a) Yes, agree. In Figure R9, the mass of cloud water and cloud ice increases, indicating the saturated situation for both water and ice. Moreover, the number of ice crystals also increases, suggesting inappropriate to attribute to BF processes. The response to comment 14 has been revised as:*

*"Figure R9 shows the changes in the mass and number concentration of the different hydrometers. The aerosols are activated to form more cloud droplets on which water condenses and produces more cloud water (Figure R9a). This process releases additional latent heat at 3–5 km due to condensation (Figure R10a) and lower supersaturation, which is also discussed in Fan et. al (2018). The smaller radius of cloud droplet shown in Figure 5a is not favorable to fast droplet coalescence and suppress warm rain. The precipitation decreases from 15Z to 20Z on 14 December (Figure S4). With aerosols, the precipitation is increased between 03Z on December 15 to 10Z on December 16. However, the changes in the hydrometers, particular for rain water, and sources of latent heat release are quite different between before and after 15Z on December 15. These differences indicate that the processes and their related mechanisms may differ from each other. In the first stage, before 15Z on December 15, there are abundant ice crystals (i.e. snow and graupel) above the 0°C isotherm around 5 km (Figure S8). With aerosols, the snow and graupel grows at the expenses of ice crystals and rain water via aggregation and rimming, respectively (Figure R9c–e). The former refers to the collision and coalescence of ice crystals to form snow while the latter represents the accretion of cloud drops and rain drops by snow and graupel to form larger graupels. These are the main processes of converting liquid mass to solid phase, contributing to additional precipitating particles. However, the latent heat due to rimming is relatively small (Figure R10f) because the latent heat release per unit for freezing (334 kJ kg⁻¹) is only 1/8 of that for deposition (2256 kJ kg⁻¹). The latent heat release due to deposition*

precipitation". Ice crystals would not fall as precipitation. Snow and graupel are the precipitating particles. (d) The figure R10 is confusing. How can warm cloud have deposition and freezing? How do you define warm clouds? Also, why not show the values below 3 Kd-1, which is significant in differences? Please clarify "anomalies that exceed 90% significance level". First, there is no observations so please define anomaly here. Second, how the significance test is done since data between two simulations cannot be compared in pairs in grid level because very different clouds could form. If the test is conducted based on mean values, are there enough data for such a test?

*Response: (a) Yes, agree. In Figure R9, the mass of cloud water and cloud ice increases, indicating the saturated situation for both water and ice. Moreover, the number of ice crystals also increases, suggesting inappropriate to attribute to BF processes. The response to comment 14 has been revised as:*

*"Figure R9 shows the changes in the mass and number concentration of the different hydrometers. The aerosols are activated to form more cloud droplets on which water condenses and produces more cloud water (Figure R9a). This process releases additional latent heat at 3–5 km due to condensation (Figure R10a) and lower supersaturation, which is also discussed in Fan et. al (2018). The smaller radius of cloud droplet shown in Figure 5a is not favorable to fast droplet coalescence and suppress warm rain. The precipitation decreases from 15Z to 20Z on 14 December (Figure S4). With aerosols, the precipitation is increased between 03Z on December 15 to 10Z on December 16. However, the changes in the hydrometers, particular for rain water, and sources of latent heat release are quite different between before and after 15Z on December 15. These differences indicate that the processes and their related mechanisms may differ from each other. In the first stage, before 15Z on December 15, there are abundant ice crystals (i.e. snow and graupel) above the 0°C isotherm around 5 km (Figure S8). With aerosols, the snow and graupel grows at the expenses of ice crystals and rain water via aggregation and rimming, respectively (Figure R9c–e). The former refers to the collision and coalescence of ice crystals to form snow while the latter represents the accretion of cloud drops and rain drops by snow and graupel to form larger graupels. These are the main processes of converting liquid mass to solid phase, contributing to additional precipitating particles. However, the latent heat due to rimming is relatively small (Figure R10f) because the latent heat release per unit for freezing (334 kJ $kg^{-1}$) is only 1/8 of that for deposition (2256 kJ $kg^{-1}$). The latent heat release due to deposition*

*in cold cloud is stronger than that due to condensation in warm cloud even though the latter is also important (Figure R9a and f). In deep convection, the strong updraft usually makes the atmospheric condition saturated for water which is supersaturated with respect to ice. With the presence of ice crystals (Figure S8), the formation of ice crystals is enhanced accompanied by additional latent heat release due to deposition (Figure R9 and 10). After 15Z on December 15, most of the snow and graupel sedimentate. Compared with depositional heating, the condensational heating plays a dominant role in intensifying convective strength. The rain water increases through accretion of added cloud droplets, leading to precipitation increases."*

*The descriptions have been integrated in the main text (P9L14–P10L27).*

*(b) Thanks for pointing this out. Yes, agree, rimming and aggregation are the main processes for the growth of snow and graupel. Please the revision accordingly in the responses to (a) above.*

*The output latent heat release due to phase change (i.e., condensation, deposition and freezing) are derived by adding additional diagnostic in the Morrison microphysical scheme. Each term is calculated based on the equation as follows:*

*For warm clouds,*

$$\text{T3D\_Wcon(K)} = (\text{PRE(K)} + \text{PCC(K)}) * \text{XXLV(K)}/\text{CPM(K)} \quad (1)$$

$$\text{T3D\_Wdep(K)} = (\text{EVPMS(K)} + \text{EVPMG(K)} * \text{XXLS(K)})/\text{CPM(K)} \quad (2)$$

$$\text{T3D\_Wfrz(K)} = (\text{PSMLT(K)} + \text{PGMLT(K)} - \text{PRACS(K)} - \text{PRACG(K)}) * \text{XLF(K)}/\text{CPM(K)} \quad (3)$$

*Where the left terms refer to latent heat release due to condensation, sublimation, and melting in Equation (1), (2), and (3), respectively. K is the layer in vertical for loop. The first term in the bracket on the right side represent different microphysical processes contributing the latent heat release. Based on Mao et al., (2018), more information on the warm-cloud transfer processes between different hydrometers for each process is described in Table 1. The terms of XXLV, XXLS, and XLF denote the latent heat release per unit of condensation, deposition, and freezing, respectively. CPM is specific heat at constant pressure for moist air.*

*Similarly, for cold clouds,*

$$T3D\_Ccon(K) = (PRE(K) + PCC(K)) * XXLV(K)/CPM(K) \quad (4)$$

$$T3D\_Cdep(K) = (PRD(K) + PRDS(K) + MNUCCD(K) + EPRD(K) + EPRDS(K) + PRDG(K) + EPRDG(K))$$
$$* XXLS(K)/CPM(K) \quad (5)$$

$$T3D\_Cfrz(K) = (PSACWS(K) + PSACWI(K) + MNUCCC(K) + MNUCCR(K) + QMULTS(K) + QMULTG(K)$$
$$+ QMULTR(K) + QMULTRG(K) + PRACS(K) + PSACWG(K) + PRACG(K) + PGSACW(K)$$
$$+ PGRACS(K) + PIACR(K) + PIACRS(K)) * XLF(K)/CPM(K) \quad (6)$$

***The information on the cold-cloud transfer processes between different hydrometers for each process is described in Table 2.***

*Table 1. Description of warm-cloud processes contributing to latent heat release. Red, green, and blue indicate condensation, deposition, and freezing related processes, respectively. If the term is negative, it refers to the opposite transfer process from sink to source. For example, negative of PRE represent the evaporation of $Q_r$.*

| Abbreviation | Warm-cloud processes | Source | Sink |
|:---:|:---:|:---:|:---:|
| PRE | Condensation of $Q_v$ | $Q_v$ | $Q_r$ |
| PCC | Condensation of $Q_v$ | $Q_v$ | $Q_c$ |
| EVPMS | Sublimation of $Q_s$ | $Q_s$ | $Q_v$ |
| EVPMG | Sublimation of $Q_g$ | $Q_g$ | $Q_v$ |
| PSMLT | Melting of $Q_s$ | $Q_s$ | $Q_r$ |
| PGMLT | Melting of $Q_g$ | $Q_g$ | $Q_r$ |
| PRACS | Collection of $Q_r$ by $Q_s$ | $Q_r$ | $Q_s$ |
| PRACG | Collection of $Q_r$ by $Q_g$ | $Q_r$ | $Q_g$ |

*Table 2. Same as Table 1 but for cold cloud.*

| Abbreviation | Cold-cloud processes | Source | Sink |
|:---:|:---:|:---:|:---:|
| PRE | Condensation of $Q_v$ | $Q_v$ | $Q_r$ |
| PCC | Condensation of $Q_v$ | $Q_v$ | $Q_c$ |
| PRD | Deposition of $Q_v$ | $Q_v$ | $Q_s$ |
| PRDS | Deposition of $Q_v$ | $Q_v$ | $Q_g$ |
| MNUCCD | Ice nucleation | $Q_v$ | $Q_i$ |
| EPRD | Sublimation of $Q_i$ | $Q_i$ | $Q_v$ |
| EPRDS | Sublimation of $Q_s$ | $Q_s$ | $Q_i$ |
| PRDG | Deposition of $Q_v$ | $Q_v$ | $Q_g$ |
| EPRDG | Sublimation of $Q_g$ | $Q_g$ | $Q_v$ |
| PSACWS | Accretion of $Q_c$ by $Q_s$ | $Q_c$ | $Q_s$ |
| PSACWI | Accretion of $Q_c$ by $Q_i$ | $Q_c$ | $Q_i$ |
| MNUCCC | Contacting freezing of $Q_c$ | $Q_c$ | $Q_i$ |
| MNUCCR | Contacting freezing of $Q_r$ | $Q_r$ | $Q_g$ |
| QMULTS | Multiplication due to collision $Q_c$ by $Q_s$ | $Q_c$ | $Q_i$ |
| QMULTG | Multiplication due to collision $Q_c$ by $Q_g$ | $Q_c$ | $Q_i$ |
| QMULTR | Multiplication due to collision $Q_r$ by $Q_s$ | $Q_r$ | $Q_i$ |
| QMULTRG | Multiplication due to collision $Q_r$ by $Q_g$ | $Q_r$ | $Q_i$ |
| PRACS | Collection of $Q_r$ by $Q_s$ | $Q_r$ | $Q_s$ |

| | | | |
|---|---|---|---|
| PSACWG | Collection of $Q_c$ by $Q_g$ | $Q_c$ | $Q_g$ |
| PRACG | Collection of $Q_r$ by $Q_g$ | $Q_r$ | $Q_g$ |
| PGSACW | Collection of $Q_c$ by $Q_s$, conversion to $Q_g$ | $Q_c$ | $Q_g$ |
| PGRACS | Collection of $Q_r$ by $Q_s$, conversion to $Q_g$ | $Q_r$ | $Q_g$ |
| PIACR | Collection of $Q_r$ by $Q_i$, conversion to $Q_g$ | $Q_r$ | $Q_g$ |
| PIACRS | Collection of $Q_r$ by $Q_i$ conversion to $Q_s$ | $Q_r$ | $Q_s$ |

*These contents has been integrated in the main text as Appendix A.*

*(c) Agree. The text has been revised as:*

*"most of the snow and graupel sedimentate".*

*(d) Yes, agree. The deposition and freezing in warm cloud refer to sublimation and melting, respectively. The figures have been revised accordingly. The warm cloud in this study is defined as the cloud at the vertical layer above 0°C. Sorry for the confusing, the values below 3 K d⁻¹ are not shown because zero-value lines are omitted, and the contour interval is 3 K d⁻¹. To avoid confusion, we remove the description "Note the blank represent the values are within 3 K d⁻¹". The anomaly is the deviation of experiment relative to CLEAN run. The significance test in this study is analogous to that conducted in climate. For example, given the climatology differences between two 30-year datasets, the sample is 30 for each experiment to conduct significance test. In this work, if we look at the significance of the differences in precipitation average on 15 December, the sample is 120 which is derived as the product of hours per day (24 hours precipitation performing average) and number of ensemble members for each experiment. This means we conduct significance test at grid level. We removed them if this is inappropriate. Thanks for pointing this out.*

(8) #16 response: The convergence should be primarily because the dry cold air meet with warm humid air as a result of large-scale dynamics. Microphysics might enhance the convergence, but it is not the cause of the convergence over the large region. In addition, moisture is increased in the red box domain, which need to discuss where the source is.

*Response: Agree. We revised the text to "The convergence is enhanced via microphysical processes". As discussed in the main text, the column-integrated water vapor changes are small compared with precipitation changes (Figure RR 2). The precipitation increase is mainly through the enhanced moisture flux convergence via microphysical-dynamical*

*feedback (Figure 8). The changes in moisture flux convergence is driven by convergence in a dynamical way. The source of moisture is mainly from the ocean transported by the southerly flow (Figure 20d). A clear gradient of moisture is seen from the ocean to the land.*

[Figure]

*Figure RR 2. Spatial distribution of column-integrated water vapor changes (unit: mm) on 15 December between CTL and CLEAN.*

(9) #18 response: (a) I do not understand "the persistent convective system makes the impact last for longer time". (b) The authors missed the point about my question "how the changes in domain 1 impact the results over domain 2". When emissions and aerosols are changed in Domain 1, the methodological field including temperature and moisture would be changed too. Those changes would impact domain 2 simulation since BC is from domain 1.

*Response: (a) Sorry for the confusion. We removed the description in the main text.*

*(b) Agree and thanks for pointing this out. We integrate the following contents into our discussion (P17 L9–L25):*

*"One may wonder whether the precipitation differences over domain 2 in D1 experiment is driven by meteorological fields changes or by transport of aerosols because the scaling of emissions in domain 1 also modify the local atmospheric conditions. The changes in meteorology in turn may affect the precipitation in domain 2. Figure RR 3 shows the aerosol effects on 2-m temperature and column water vapor in domain 1. With aerosols, the moisture change is small over the whole China. The surface temperature decreases up to about 1 K is seen over northeastern China, Sichuan, and northeastern Indo-China Peninsula through*

*absorbing and scattering solar radiation as well as serving cloud condensation nuclei. The temperature over Guangdong province show marginal changes as the aerosol concentration is concentrated to the north of Guangdong and incident solar radiation is weak in rainy days. The relatively small changes in meteorological fields over domain 2 may indicate a dominant role of transboundary aerosols. Figure RR 4 shows the precipitation differences over domain 2 on 15 December based on domain 1 output. The pattern of precipitation changes is very different from that calculated based on domain 2 output, suggesting that the atmospheric condition changes in domain 1 cannot account for the precipitation differences in Figure 3d. Moreover, the importance of aerosol-cloud interactions discussed above works for both D1 and D2 experiment which may further confirm the precipitation changes in Guangdong is driven by transboundary aerosols rather than changes in meteorology in domain 1. Note the cumulus scheme is used in domain 1 but not in domain 2 which may result in different response of precipitation to atmospheric changes in domain 1. To completely disentangle the meteorology impact from that of transboundary aerosols, the possible solution could be application of nudging to constrain the meteorology as same as CTL and scale the emissions in domain 1. This could be in future sensitivity studies."*

**T2**

[Figure]

Figure RR 3. *Differences in 2-m temperature (unit: K) between (a) CTL and CLEAN (i.e. CTL minus CLEAN) and (b) D1 and CLEAN (i.e. D1 minus CLEAN) on December 15. (c, d) Same as (a, b) but for column water vapor (unit: mm). Red boxes (22°–24°N, 112°–115°E) denote the analysis region.*

**Precipitation**

[Figure]

Figure RR 4. *Differences in precipitation (unit: mm) between (a) CTL and CLEAN (i.e. CTL minus CLEAN) and (b) D1 and CLEAN (i.e. D1 minus CLEAN) on December 15 based on domain 1 output.*

**Reference:**

Fan, J., Wang, Y., Rosenfeld, D. and Liu, X.: Review of Aerosol–Cloud Interactions: Mechanisms, Significance, and Challenges, J. Atmos. Sci., 73(11), 4221–4252, doi:10.1175/JAS-D-16-0037.1, 2016.

Mao, J., Ping, F., Yin, L. and Qiu, X.: A Study of Cloud Microphysical Processes Associated With Torrential Rainfall Event Over Beijing, J. Geophys. Res. Atmos., doi:10.1029/2018JD028490, 2018.

**Contribution of local and remote anthropogenic aerosols to  a record-breaking torrential rainfall event in Guangdong Province, China**

Z. Liu[1,2,3], Y. Ming[5], C. Zhao[6], N.C. Lau[1,2,4], J.P. Guo[7], M. Bollasina[3], Steve H.L. Yim[1,4,2]

[1]Institute of Space and Earth Information Science, The Chinese University of Hong Kong, Hong Kong, China
[2]Institute of Environment, Energy and Sustainability, The Chinese University of Hong Kong, Sha Tin, N.T., Hong Kong
[3]School of Geosciences, University of Edinburgh, UK
[4]Department of Geography and Resource Management, The Chinese University of Hong Kong, Sha Tin, N.T., Hong Kong
[5]Geophysical Fluid Dynamics Laboratory/NOAA, Princeton, New Jersey, USA
[6]School of Earth and Space Sciences, University of Science and Technology of China, Hefei, Anhui, China
[7]State Key Laboratory of Severe Weather, Chinese Academy of Meteorological Sciences, Beijing 100081, China

*Correspondence to*: Steve H.L. Yim (steveyim@cuhk.edu.hk)

[Figure]

**Figure S1. (a) WRF-Chem model two-nested domains with resolutions of 20 km and 4 km for domain 1 (D1) and domain 2 (D2), respectively. Shading represents terrain height (unit: m). (b) Spatial distribution of 3-day averaged cloud top temperature (shading; unit: °C) during December 14–16, 2013 over domain 2 in control run.**

**500-hPa Z and Wind**

[Figure]

Figure S2. Spatial distribution of 3-day averaged 500-hPa wind (vector; unit: m s⁻¹) and height (shading; unit: m) during December 14–16, 2013 for (a) OBS from ERA-interim and (b) CTL from control simulation.

[Figure]

**Figure S3. Differences in accumulated precipitation (unit: mm) on December 16 between (a) CTL and CLEAN (i.e., CTL minus CLEAN), (b) CTL and ARIoff (i.e., CTL minus ARIoff), (c) ARIoff and CLEAN (i.e., ARIoff minus CLEAN), (d) D1 and CLEAN (i.e., D1 minus CLEAN), (e) D2 and CLEAN (D2 minus CLEAN), and (f) 10X and CLEAN (10X minus CLEAN). Red boxes (22°–24° N, 112°–115° E) denote the analysis region. ARIoff run refers to simulation with aerosol-radiation interactions off.**

[Figure]

**Figure S4. Time series of station average rain rate (unit: mm h⁻¹) over 22°–24° N, 112°–115° E (a) for OBS (red), CMORPH (black), CTL (blue), ARIoff (green), and CLEAN (purple).**

[Figure]

**Figure S5. Spatial distribution of accumulated precipitation (unit: mm) from 00Z on December 14, 2013, to 00Z on December 17, 2013 from (a) station observations (OBS), (b) CMORPH, (c) control simulation (CTL), and (d) TRMM. Circles denote locations of in situ observations.**

PM$_{2.5}$

[Figure]

Figure S6. PM$_{2.5}$ concentration (unit: μg m$^{-3}$) average during December 14–16, 2013 for (a) observation and (b) control simulation. Colored circles denote in situ station locations.

PM$_{2.5}$

[Figure]

5    Figure S7. Time series of PM$_{2.5}$ averaged over all the stations during December 14–16, 2013 for CTL (black) and OBS (red).

[Figure]

**Figure S8. Distribution with time (abscissa) and height (ordinate) in (a) cloud water (shading; unit: $10^{-5}$ kg kg$^{-1}$) and CDNC (contour; unit: $10^{7}$ kg$^{-1}$), (b) cloud ice (shading; unit: $10^{-5}$ kg kg$^{-1}$) and CINC (contour; unit: $10^{4}$ kg$^{-1}$), (c) rain (shading; unit: $10^{-5}$ kg kg$^{-1}$) and rain number concentration (contour; unit: $10^{5}$ kg$^{-1}$), (d) snow (shading; unit: $10^{-4}$ kg kg$^{-1}$) and snow number concentrations (contour; unit: $10^{3}$ kg$^{-1}$), and (e) graupel (shading; unit: $10^{-4}$ kg kg$^{-1}$) and graupel number concentration (contour; unit: $10^{3}$ kg$^{-1}$) averaged over the red box in CTL run. Only anomalies that exceed 90% significance level are depicted with shading and contour.**

[Figure]

[Figure]

**Figure S9.** Differences with time (abscissa) and height (ordinate) in (a) cloud water (shading; unit: $10^{-5}$ kg kg$^{-1}$) and CDNC (contour; unit: $10^{7}$ kg kg$^{-1}$), (b) cloud ice (shading; unit: $10^{-5}$ kg kg$^{-1}$) and CINC (contour; unit: $10^{4}$ kg kg$^{-1}$), (c) rain (shading; unit: $10^{-5}$ kg kg$^{-1}$) and rain number concentration (contour; unit: $10^{5}$ kg kg$^{-1}$), (d) snow (shading; unit: $10^{-4}$ kg kg$^{-1}$) and snow number concentrations (contour; unit: $10^{3}$ kg kg$^{-1}$), and (e) graupel (shading; unit: $10^{-4}$ kg kg$^{-1}$) and graupel number concentration (contour; unit: $10^{3}$ kg kg$^{-1}$) between D2 and CLEAN (i.e. D2 minus CLEAN) averaged over the red box.

**Warm Cloud**                    **Cold Cloud**

[Figure]

[Figure]

[Figure]

Figure S10. Differences with time (abscissa) and height (ordinate) in latent heat release (unit: K d⁻¹) from (a) condensation, (b) deposition, and (c) freezing processes between D2 and CLEAN (i.e. D2 minus CLEAN) averaged over the red box for the warm cloud. (d–f) Same as (a–c) but from cold cloud.  Zero-value contour lines are omitted, and negative values are dashed. The contour interval is 3 K d⁻¹.

[Figure]

**Figure S11. Distribution with time (abscissa) and height (ordinate) in (a) cloud water (shading; unit: $10^{-5}$ kg kg$^{-1}$) and CDNC (contour; unit: $10^{7}$ kg$^{-1}$), (b) cloud ice (shading; unit: $10^{-5}$ kg kg$^{-1}$) and CINC (contour; unit: $10^{4}$ kg$^{-1}$), (c) rain (shading; unit: $10^{-5}$ kg kg$^{-1}$) and rain number concentration (contour; unit: $10^{5}$ kg$^{-1}$), (d) snow (shading; unit: $10^{-4}$ kg kg$^{-1}$) and snow number concentrations (contour; unit: $10^{3}$ kg$^{-1}$), and (e) graupel (shading; unit: $10^{-4}$ kg kg$^{-1}$) and graupel number concentration (contour; unit: $10^{3}$ kg$^{-1}$) averaged over the region in 24°–25°N, 110°–112°E from CTL run. **

[Figure]

[Figure]

[Figure]

**Figure S12. Differences with time (abscissa) and height (ordinate) in (a) cloud water (shading; unit: $10^{-5}$ kg kg$^{-1}$) and CDNC (contour; unit: $10^7$ kg$^{-1}$), (b) cloud ice (shading; unit: $10^{-5}$ kg kg$^{-1}$) and CINC (contour; unit: $10^4$ kg$^{-1}$), (c) rain (shading; unit: $10^{-5}$ kg kg$^{-1}$) and rain number concentration (contour; unit: $10^5$ kg$^{-1}$), (d) snow (shading; unit: $10^{-4}$ kg kg$^{-1}$) and snow number concentrations (contour; unit: $10^3$ kg$^{-1}$), and (e) graupel (shading; unit: $10^{-4}$ kg kg$^{-1}$) and graupel number concentration (contour; unit: $10^3$ kg$^{-1}$) between CTL and CLEAN (i.e. CTL minus CLEAN) averaged over the region in 24°–25°N, 110°–112°E. **

[Figure]

[Figure]

**Figure S13. Differences with time (abscissa) and height (ordinate) in latent heat release (unit: K d⁻¹) from (a) condensation, (b) deposition, and (c) freezing processes between CTL and CLEAN (i.e. CTL minus CLEAN) averaged over the region in 24°–25°N, 110°–112°E for the warm cloud. (d–f) Same as (a–c) but from cold cloud.**  **Zero-value contour lines are omitted, and negative values are dashed. The contour interval is 3 K d⁻¹.**

[Figure]

[Figure]

**Figure S14.** Differences with time (abscissa) and height (ordinate) in (a) cloud water (shading; unit: $10^{-5}$ kg kg$^{-1}$) and CDNC (contour; unit: $10^{7}$ kg$^{-1}$), (b) cloud ice (shading; unit: $10^{-5}$ kg kg$^{-1}$) and CINC (contour; unit: $10^{4}$ kg$^{-1}$), (c) rain (shading; unit: $10^{-5}$ kg kg$^{-1}$) and rain number concentration (contour; unit: $10^{5}$ kg$^{-1}$), (d) snow (shading; unit: $10^{-4}$ kg kg$^{-1}$) and snow number concentrations (contour; unit: $10^{3}$ kg$^{-1}$), and (e) graupel (shading; unit: $10^{-4}$ kg kg$^{-1}$) and graupel number concentration (contour; unit: $10^{3}$ kg$^{-1}$) between 10× and CLEAN (i.e. 10× minus CLEAN) averaged over the region in 24°–25°N, 110°–112°E.

[Figure]

[Figure]

**Figure S151515. Differences with time (abscissa) and height (ordinate) in latent heat release (unit: K d⁻¹) from (a) condensation, (b) deposition, and (c) freezing processes between 10× and CLEAN (i.e. 10× minus CLEAN) averaged over the region in 24°–25°N, 110°–112°E for the warm cloud. (d–f) Same as (a–c) but from cold cloud. Only anomalies that exceed 90% significance level are depicted with and contour. Zero-value contour lines are omitted, and negative values are dashed. The contour interval is 3 K d⁻¹. **

[Figure]

Figure S16. Spatial distribution of surface temperature (unit: K) on (a) December 14 and (b) December 15 in 2013 in the CTL run.

Field Code Changed

[Figure]

Figure S17. Differences in 2-m temperature (unit: K) between (a) CTL and CLEAN (i.e. CTL minus CLEAN) and (b) D1 and CLEAN (i.e. D2 minus CLEAN) on December 15. (c, d) Same as (a, b) but for column water vapor (unit: mm). Red boxes (22°–24°N, 112°–115°E) denote the analysis region.

[Figure]

**Figure S18. Differences in precipitation (unit: mm) between (a) CTL and CLEAN (i.e. CTL minus CLEAN) and (b) D1 and CLEAN (i.e. D1 minus CLEAN) on December 15 based on domain 1 output.**

---

## Author Response (AR3)

**Response to Reviewer #1:**

We appreciate the reviewer who reviewed the manuscript and revision carefully and provided insightful follow-up comments. We have tried our best to address all concerns and revised the manuscript accordingly. The comments are in normal font. A point-by-point response is listed as below in bold italics.

This the third around review. The authors did not take good use of the chances to address the concerns I raised. Besides they still missed some points (examples below), they even did not try to make effort to organize the paper and present the most important points. The paper gets so lengthy and appears lack of organization (see the comment #1 below for an example below). They now got 20 formal figures and 15 supplemental figures. It appears they added lengthy text and figures to address comments but did not think of how to better organize and only present the most important points. Another evidence showing the lack of effort in presenting the study is that the figures are out of orders, for example, it is jumped from Figure S2 to Figure S6 in referencing figures. The first appearance of Figure S3 and S4 is after Figure S7. The first appearance of Figure S5 is after Figure S7 but before the first appearance of Figures S3 and S4. The first reference to Figure 1 is also after Figure 2.

***Response: Per your suggestions, the main text has been restructured and the order of the figures in both the main text and supplement has been corrected accordingly.***

I did not have time to read the whole paper but only read their response and changes and the following corrections and clarifications are needed:

1. To address my first comment of the second round review, the authors conveniently only added a few figures to the end of the supplemental materials and discussed it in the Summary and Discussion section, which does not address the point. I emphasized before that the point is to explain the opposite precipitation response at the different sectors of the system. So, to address this point well, it is equally important to describe both the increase of the precipitation at the convergence zone and warm sector and the decrease of the precipitation in the cold sector.

Then explain the reasons causing the increase and the decrease, respectively. Therefore, the changes should be started from the first paragraph in P8 where Figure 3 is discussed.

*Response: Thanks for your suggestions. In the latest version, we mention the cold front system firstly. The responses of rainfall amount to increased aerosols are described at the beginning of the result part, with increase in the warm sector and decrease in the cold sector. Given the details discussed with enough text length, it would be tedious if we equally explain the mechanism of both precipitation increase and decrease. It may be more appropriate to focus on the aggravated side with precipitation enhancement where also covers most developed cities over southern China. However, the discussion about the mechanism of the precipitation decrease in the cold sector is revised per your suggestions in comment #4.*

2. Also, the authors argued "There are lots of ice crystals with cloud ice extending up to 16 km, indicating strong deep convection". At the cold sector of a frontal system, generally there is no mechanism to form deep convective clouds. It should be deep stratiform clouds, not strong convective clouds. If this case is different from the general understanding, then needs to present evidence such as large CAPE or large low-level upward motion to support the argument of deep convective clouds.

*Response: Figure R 1 shows the spatial distribution of CAPE on December 15 in 2013. There is a salient gradient between the northwest and southeast of the domain 2 which is consistent with the surface temperature gradient (Figure S3). We agree that the relatively low CAPE over the northwest of the domain 2 suggests the stable situation there. It is more likely to form stratiform clouds. The corresponding description has been revised in the main text.*

[Figure]

*Figure R 1. Spatial distribution of CAPE (unit: J kg⁻¹) on December 15 in 2013 in the CTL run.*

3. *"The reduction of rain water and ice crystals (particularly in graupel) suggest that both the warm rain and cold rain are suppressed" - wrong statement. As I emphasized previously, ice crystals are non-precipitating particles which does not suggest precipitation information, but graupel is precipitating particle but it is not ice crystal, it is just one type of ice particles. They made such mistakes in terminology in many places throughout the paper. When talking about specific hydrometeors types, you may use ice crystal (or cloud ice), snow, graupel. When you want to take about ice particle in general covering different types, use "ice particle" not "ice crystal". This need to be changed throughout the paper.*

***Response: Thanks for your explanation and clarification. In this sentence, the ice crystals referred to snow and graupel. The misuse of hydrometeor terminology has also been corrected in the main text.***

4. The suppressed precipitation for cold-based clouds should be mainly because the reduced warm rain formation at the early times and reduced graupel formation at the later time period based on Figure S12. This is a typical response of deep stratiform clouds to CCN since the most dominant changes by aerosols for this type of clouds are collision processes including autoconversion and riming, which are less efficient due to smaller droplet size. This should be

the major argument for the suppressed precipitation. The three reasons listed by authors are mainly for deep convective clouds.

*Response: Per your suggestions. There is a strong surface temperature gradient between the southeast and northwest of the domain 2, indicating a front system. In the cold sector, the clouds tends to be stratiform clouds, which is most winter precipitation falls from. We remove the mechanism for convective clouds and revised it for stratiform clouds as you suggested.*

5. The sentence in the abstract that I pointed out previously still did not make sense. As I asked previously, how can the changes of precipitation between a polluted and clean condition resemble that from control run since the changes mean the differences?

*Response: Per you Suggestions. This sentence has been revised as "In response to 10× aerosol emissions, the pattern of precipitation and cloud property changes resembled the differences between CTL and CLEAN, but with a much greater magnitude."*

6. In response of #7 comments in the last round, I read their changes and still have the following problems,

(1) "The warm rain is still suppressed before 15Z on December 15 (Figure 6c) even though with strong latent heat release through cloud water formation": Warm rain will be always suppressed with the two-moment bulk scheme with the parameterization of autoconversion. This is very different from the treatment of in bin microphysics used in Fan et al. 2018. This needs to be clarified. Also, warm rain means the rain formed from autoconversion. Rain mass below 0 C level can be contributed by the melted particles. You only can discuss the warm rain at the times when there are no ice particles at all above 0 C level in Figure 6c.

*Response: Thanks for pointing this out. We agree that the warm rain is always suppressed as the number of converted droplets into rain drops is inversely proportional to cloud droplet numbers (Khairoutdinov and Kogan, 2000). This is also clarified in the main text. Per your suggestions, the description of warm rain has been removed.*

(2) "To further analyze the source of this latent heat release, following Fan et al. (2018), the

latent heat released from condensation, deposition, and freezing during cold and warm cloud processes are diagnosed", I do not think you can say "following Fan et al. (2018)". The author's response did not give a clear description about how the latent heat is calculated. They should not have sent a bunch of codes, instead, it should be easily described with words about how the latent heat is calculated for each process. I think the latent heat calculation should be the part of Morrison scheme since this is the feedback to temperature that a full coupled model should consider. The authors should not need to add additional code for such calculation (probably only need to find the right variable name to output it). Because the authors had wrong statements about latent heat and also said they diagnosed it in the code before, I asked these details to check if this important part was done and interpreted correctly. I did not get the answer from their response.

*Response: Yes, the latent heat is calculated in the Morrison scheme. However, in the calculation, the latent heat is only derived for warm cloud and cold cloud rather than attributed to different microphysical processes. The latent heat of each process is not calculated based on the mass. To avoid the confusion, we revise the description as follows:*

*The latent heat for each process is calculated as the product of mass conversion between different phases and its associate latent heat release rate in the model.*

*The Appendix A part is removed per your suggestion.*

(3) P10 Line 5-28, the lengthy statements they added need to revised due to misunderstandings. First, it is the basic cloud microphysics that latent heat from freezing is not a major component deep clouds as I explained last round. Condensation and deposition are always the very important condensate forming process in deep convective clouds. Second, in those past studies that the author mentioned, when they discussed the effect from freezing changed by aerosols, it is not just about the latent heat from freezing only, instead about the latent heat changes from all the processes due to the change of freezing induced by aerosols. For example, when there are more freezing, more ice crystals form, then riming and deposition will change as well.

*Response: Thanks for pointing this out. Yes, latent heat release is dominated by condensation and deposition. Sorry for the misunderstanding, we revise the description of the marginal role of freezing by attribution to amount of latent heat. In the mentioned*

*literature, the effect of freezing is indeed not only due to its latent heat release. The statement has been shortened and modified as follows:*

*"The latent heat released for each process, which is calculated as the product of mass conversion between different phases and its associate latent heat release rate in the model, is further analyzed for both cold and warm clouds (Figure 7). The salient latent heat changes mentioned above in Figure 5g is caused by deposition in cold cloud (Figure 7e). Figure S9 shows the time-height distribution of mass and number concentrations for different hydrometers in control run. Note the magnitude of snow and graupel mass is ten times of that of rain water. There are affluent snow and graupel before 15Z on 15 December located where the distinct changes in depositional heat appears. With aerosols, the snow and graupel grows at the expenses of ice crystals and rain water via aggregation and riming, respectively (Figure 6c–e). The former refers to the collision and coalescence of ice crystals to form snow while the latter represents the accretion of cloud drops and rain drops by snow and graupel to form larger graupels. These are the main processes of converting liquid mass to solid phase, contributing to additional precipitating particles. However, the latent heat due to riming is relatively small (Figure 7f) because the latent heat release per unit for freezing (334 kJ kg–1) is only 1/8 of that for deposition (2256 kJ kg–1). The latent heat release due to deposition in cold cloud is stronger than that due to condensation in warm cloud even though the latter is also important (Figure 7a and 7e). In deep convection, the strong updraft usually makes the atmospheric condition saturated for water which is supersaturated with respect to ice. With the presence of snow and graupel (Figure S9), the formation of ice particles is enhanced accompanied by additional latent heat release due to deposition (Figure 6 and Figure 7). After 15Z on December 15, most of the snow and graupel sedimentate. Compared with depositional heating, the condensational heating plays a dominant role in intensifying convective strength. The rain water increases through accretion of added cloud droplets, leading to precipitation increases. These findings highlight two different processes and mechanisms in the precipitation increase before and after 15Z on December 15. The dominant source for latent heat release is depositional heating in the former case (cold rain enhancement) while condensational heating in the latter (warm rain enhancement). Due to latent heat release with aerosols, the vertical motion is boosted (Figure 5g) which further enhance the supersaturation and associated with latent heat release. Via microphysics–dynamics feedback, the convection is intensified, and precipitation increased. This feedback has been widely discussed in ACI effects on deep convection (Fan et al., 2018; Koren et al., 2015; Tao et al., 2012)."*

With through rounds of review, I have tried hard to correct many basic knowledges and results about cloud microphysics and aerosol-cloud interaction processes to improve the quality of this paper. I urge the authors to take the opportunity to do a careful job in writing and organizing the results so that the paper can reach a certain level of qualify for publication.

*Response: We appreciate your great effort in improving this study significantly. We have tried our best to write precisely and organise the structure smoothly.*

**Reference:**

[revised manuscript text omitted]
 focusesed on December 15. The rainfall differences differences between scenarios on December 16 are put in the supplementary materials for reference (Figure S732). Distinct effects of aerosols appeared onduring the second day when the rainfall peaksed (Figure 3Figure 3Figure 3d), although aerosol concentration peaks occur s lead to more cloud droplet number concentration associated with smaller radius on the first day (Figure 4Figure 5a)but the aerosol concentration differences occurred on the first day, as shown in Figure 4b.; Tthis suggests that the a time lag effects of aerosol impact s on precipitation isare modulated by other factors (e.g. meteorological conditions). On December 15, the domain-averaged precipitation increasesed by 1.4 mm. Interestingly, a dipole pattern is manifested by aA reduction of up to 19.4 mm over appearsred in northern Guangdong Pprovince and, whereas an increase of up to 33.7 mm over occursred in southern Guangdong Pprovince, (
[revised manuscript text omitted]
 startsed at 07Z on December 15, because aerosols awere transported continuously from the north remote area. The cloud fraction reduction iwas coherent with aerosol concentration peaks, indicating that increased aerosols lead small cloud droplets to evaporate. Moreover, more deep cloud formation consuming moisture and energy. The similar Comparing patterns of cloud fraction changes between Figure 8Figure 10 Figures 7a and 107b and Figure 4Figure 4Figure 4b indicates the dominant effects of aerosols from remote areas. The CDNC (shading) increasesd in both D1 and D2 runs compared with the CLEAN run before the rainfall peak (Figure S12Figure 11 Figures 8a and S1218b). However, the discernible cloud effective radius (contours) decrease appearsed only in the D1 run and iwas attributed to a stronger CDNC increase. Correspondingly, the CINC and ice cloud effective radius showed more remarkable increases in the D1 run during the rainfall peak time (Figure S12Figure 11Figures 8c and S1218d). The associated latent heat and vertical velocity awere much stronger in the D1 run compared with that in the D2 run (Figure S12Figure 11Figure 11Figures 8e and S1218f). Interestingly, most of latent heat release with local emissions are occurshappened below the 0°C isotherm line. Figure 9Figure 9Figure 12 shows the changes in mass and number of different hydrometeors with remote aerosols emissions. There are plenty of snow and graupel formations at the expense of rain water when precipitation increases before 15Z on 15 December, indicating an intensified cold rain process. The corresponding latent heat release is dominated by deposition in cold cloud (Figure S13). By contrast, after 15Z on 15 December 15, rain water increases significantly during precipitation enhancement, representing stronger warm rain process. The associated latent heat release is due to condensational heating in warm cloud concentrated below the 0°C isotherm line. The patterns of changes in hydrometeors and latent heat in D1 assembles that in CTL run, which further confirming the driving factor dominant role of remote aerosols emissions. The distribution of time-height changes in hydrometeors and latent heat between D2 and CLEAN runs are shown in Figure S149 and Figure S150, respectively. As aerosols from local emissions are concentrated near the surface and are washed out dramatically once the rain initiated, much less cloud water formed than that in D1 run. Thus, the supersaturation is lowered as

strongly as that in D1 simulation. More rain water is formed by accretion of cloud droplets which indicates that intensified warm rain is the only reason for the precipitation increase with local aerosol emissions. As a result, the average precipitation increase over R1 the analysis region on December 15 iwas 7.3 mm with remote aerosol emissions, much greater than that with local aerosol emissions (3.1 mm, Figure 10Figure 10Figure 14c a Figures 9c and 1049d). These findings suggest that both the

5   effects of local and, to a much greater extent, remote aerosol emissions contribute to precipitation increases over the analysis region.

**3.4 Tenfold anthropogenic emissions and chemical ICs and BCs**

An optimal aerosol loading should exist theoretically in which the convection is the most vigorous (Rosenfeld et al., 2008). For aerosol concentrations below the optimum, the convection is invigorated by smaller droplets; thus, stronger updraft releases

10   larger latent release (Dagan et al., 2015b). By contrast, suppression effects dominate above the optimum (Small et al., 2009). The optimum value is determined by environmental conditions (e.g., relative humidity, see Dagan et al., 2015a). In this section, a tenfold aerosol emission simulation (10×) iwas conducted to examine the sensitivity of precipitation and associated cloud properties to aerosol concentrations.

The PM$_{2.5}$ concentrations (contours) in the tenfold aerosol emission simulation (10×)10× increasesd significantly to

15   approximately ten times that in CTL, indicating a linear relationship from emissions to aerosol concentration (Figure S16Figure 15Figure 10). The associated boundary layer cloud formation (shading) iwas further suppressed below 2 km, which is consistent with the result in Figure 4Figure 4b. The change patterns changes in cloud fraction and aerosol concentration in Figure S16 Figure 15. DFigure 10 are similar to that in Figure 4Figure 4Figure 4b, but with Figure 15 Figure 10 shows a much greater magnitude. The CDNC (shading) increase and cloud effective radius (contours) reduction in Figure S17Figure

20   16Figure 11a are also more pronounced than those in Figure 5Figure 5Figure 5a. CDNC noticeably decreasesd below 1.5 km but increasesd substantially from 1.5 km to 4 km before 04Z4:00 a.m. on December 14, associating with smaller radius. Smaller cloud droplet tends to evaporate. In addition, more cloud droplets are produced due to higher supersaturation upward. The consumption of water and energy leads to a further reduction ofin low cloud (Figure S18Figure 17a). This finding suggests the ascent of cloud droplets, which is attributed to the smaller effective radius induced by excessive aerosols in 10× compared

25   with that in CTL. The smaller cloud droplets favored the formation of deeper convection manifested by more CINC and larger ice cloud effective radii (Figure 11b). The involved latent heat and vertical velocity during the rainfall peak time (from 08Z on December 15 to 10Z on December 16) in Figure S17Figure 16c Figure 11c exhibit a stronger increase associated with a higher altitude above the freezing level than thoseat in Figure 5Figure 5Figure 5c. Besides, a distinct weaker latent heat release associated with negative vertical velocity anomaly appears below freezing level between 10Z and 22Z on 15 December. This

30   indicate a more important role of cold related processes in latent heat release. The ice crystals also increase drastically with bigger radius. Figure 17 shows the changes in mass and number concentrations of different hydrometeors in 10× simulation. Compared with the CTL run, the snow and graupel are also increased with a strongerlarger magnitude, particularly before 15Z

on 15 December, indicating enhanced cold rain. However, rain water shows decrease during all the time instead of an increase after 15Z when precipitation increases in the CTL run. This means the warm rain is suppressed much stronger in 10× simulation. As wWith ten times of aerosols emissions, the aerosolss lower the supersaturation much stronger by activation to form much smaller cloud droplets. The rain water evaporates rather than increases by accretion of additional cloud droplet, associating with strong condensational cooling in warm cloud (Figure S19Figure 18a). This means that water ascended higher and froze before precipitating, which led to additional latent heat release. A more salient negative anomaly of latent heat and vertical velocity arose below 4 km from 06Z to 22Z on December 15 and below 10 km from 06Z to 18Z on December 16. This should relate to stronger cloud evaporation and ice melting, as discussed by Rosenfeld et al. (2008). The greater cooling below and greater heating above suggest the intensified upward energy transport. This configuration should enhance updraft above and downdraft below induced by additional warming and cooling respectively, which could further invigorate convection and produce more precipitation (Rosenfeld, 2006). Precipitation on December 15 iwas suppressed up to 39.6 mm over the upstream region of aerosol sources up to 39.6 mm in the northwest of Guangdong province but substantially enhanced up to 59.7 mm over the downstream region near the coastal region (Figure 11Figure 19bFigure 12b). A similar finding iwas reported by Zhong et al. (2015). TThe delay of early rain in the upstream area resultesd in more rainfall with a and stronger rain intensity within the downstream area and a more narrowed region in the downstream areacompared with the red box in Figure 3Figure 3b. The average precipitation overin Guangdong Pprovince on December 15 decreasesd by 1.0 mm in 10×, whereas it increasesd 
[revised manuscript text omitted]
 cloud top temperature (shading; unit: °C) during December 14–16, 2013 over domain 2 in control run.**

**500-hPa Z and Wind**

[Figure]

Figure S2. Spatial distribution of 3-day averaged 500-hPa wind (vector;  m s⁻¹) and height (shading;  m) during December 14–16, 2013 for (a) OBS from ERA-interim and (b) CTL from control simulation.

**Surface Temperature (K)**

[Figure]

5   Figure S3. Spatial distribution of surface temperature (K) on (a) December 14 and (b) December 15 in 2013 in the CTL run.

[Figure]

**Figure S446.** PM$_{2.5}$ concentration (unit: μg m$^{-3}$) average during December 14–16, 2013 for (a) observation and (b) control simulation. Colored circles denote in situ station locations.

[Figure]

5    **Figure S557.** Time series of PM$_{2.5}$ concentration averaged over all the air quality stations during December 14–16, 2013 for CTL (black) and OBS (red).

[Figure]

Precipitation

**Figure S3. Differences in accumulated precipitation (unit: mm) on December 16 between (a) CTL and CLEAN (i.e., CTL minus CLEAN), (b) CTL and ARIoff (i.e., CTL minus ARIoff), (c) ARIoff and CLEAN (i.e., ARIoff minus CLEAN), (d) D1 and CLEAN (i.e., D1 minus CLEAN), (e) D2 and CLEAN (D2 minus CLEAN), and (f) 10X and CLEAN (10X minus CLEAN). Red boxes (22°–24° N, 112°–115° E) denote the analysis region. ARIoff run refers to simulation with aerosol-radiation interactions off.**

[Figure]

Figure S4. Time series of station average rain rate (unit: mm h⁻¹) over 22°–24° N, 112°–115° E (a) for OBS (red), CMORPH (black), CTL (blue), ARIoff (green), and CLEAN (purple).

[Figure]

**Figure S6. Spatial distribution of accumulated precipitation ( mm) from 00Z on December 14, 2013 to 00Z on December 17, 2013 from (a) station observations (OBS), (b) CMORPH, (c) control simulation (CTL), and (d) TRMM. Circles denote locations of in situ observations.**

[Figure]

**Figure S7. Differences in accumulated precipitation (mm) on December 16 between (a) CTL and CLEAN, (b) CTL and ARIoff, (c) ARIoff and CLEAN, (d) D1 and CLEAN, (e) D2 and CLEAN, and (f) 10× and CLEAN. ARIoff run refers to simulation with aerosol-radiation interactions off.**

[Figure]

**Figure S8.** Time series of rain rate (mm h$^{-1}$) averaged over R1 (a) for 10× (red), CTL (black), ARIoff (blue), and CLEAN (green).

[Figure]

**Figure S9.** Differences with time (abscissa) and height (ordinate) in latent heat release (K d$^{-1}$) from (a) condensation, (b) deposition, and (c) freezing processes between CTL and CLEAN averaged over R1 for warm cloud. (d–f) Same as (a–c) but for cold cloud. Zero-value contour lines are omitted, and negative values are dashed.  The contour interval is 3 K d$^{-1}$.

PM$_{2.5}$

[Figure]

Figure S6. PM$_{2.5}$ concentration (unit: μg m$^{-3}$) average during December 14–16, 2013 for (a) observation and (b) control simulation. Colored circles denote in situ station locations.

PM$_{2.5}$

[Figure]

5    Figure S7. Time series of PM$_{2.5}$ averaged over all the stations during December 14–16, 2013 for CTL (black) and OBS (red).

[Figure]

Figure S10108. Distribution with time (abscissa) and height (ordinate) in (a) cloud water (shading; unit: 10⁻⁵ kg kg⁻¹) and CDNC (contour; unit: 10⁷ kg⁻¹), (b) cloud ice (shading; unit: 10⁻⁵ kg kg⁻¹) and CINC (contour; unit: 10⁴ kg⁻¹), (c) rain (shading; unit: 10⁻⁵ kg kg⁻¹) and rain number concentration (contour; unit: 10⁵ kg⁻¹), (d) snow (shading; unit: 10⁻⁴ kg kg⁻¹) and snow number concentrations (contour; unit: 10³ kg⁻¹), and (e) graupel (shading; unit: 10⁻⁴ kg kg⁻¹) and graupel number concentration (contour; unit: 10³ kg⁻¹) averaged over R1 the red box in CTL run. Only anomalies that exceed 90% significance level are depicted with shading and contour.

[Figure]

**Figure S11. Differences in column-integrated moisture convergence (CON; mm) between (a) CTL and CLEAN and (b) ARIoff and CLEAN on December 15. (c, d) Same as (a, b) but for column-integrated advection of water vapor (ADV; mm). The numbers at the top-left corner of each panel represent the values averaged over R1.**

[Figure]

**Figure S12. Differences with time (abscissa) and height (ordinate) in (a) CDNC (shading; $10^7$ kg$^{-1}$) and cloud effective radius (contour; μm), (c) CINC (shading; $10^5$ kg$^{-1}$) and ice cloud effective radius (contour; μm), and (e) vertical velocity (shading; cm s$^{-1}$) and latent heating (contour; K d$^{-1}$) averaged over R1 between D1 and CLEAN. (b, d, f) same as (a, c, e) but for differences between D2 and CLEAN. Zero-value contour lines are omitted, and negative values are dashed.**

[Figure]

**Figure S13.** Differences with time (abscissa) and height (ordinate) in latent heat release (K d$^{-1}$) from (a) condensation, (b) deposition, and (c) freezing processes between D1 and CLEAN averaged over R1 for warm cloud. (d–f) Same as (a–c) but for cold cloud.

[Figure]

[Figure]

**Figure S14149.** Differences with time (abscissa) and height (ordinate) in (a) cloud water (shading;  10⁻⁵ kg kg⁻¹) and CDNC (contour;  10⁷ kg⁻¹), (b) cloud ice (shading;  10⁻⁵ kg kg⁻¹) and CINC (contour;  10⁴ kg⁻¹), (c) rain (shading; unit: 10⁻⁵ kg kg⁻¹) and rain number concentration (contour;  10⁵ kg⁻¹), (d) snow (shading;  10⁻⁴ kg kg⁻¹) and snow number concentrations (contour;  10³ kg⁻¹), and (e) graupel (shading;  10⁻⁴ kg kg⁻¹) and graupel number concentration (contour;  10³ kg⁻¹) between D2 and CLEAN  averaged over the red box.

[Figure]

**Warm Cloud**          **Cold Cloud**

[Figure]

**Figure S1510.** Differences with time (abscissa) and height (ordinate) in latent heat release ( K d⁻¹) from (a) condensation, (b) deposition, and (c) freezing processes between D2 and CLEAN  averaged over R1  for  warm cloud. (d–f) Same as (a–c) but from cold cloud.  Zero-value contour lines are omitted, and negative values are dashed. The contour interval is 3 K d⁻¹.

[Figure]

**Figure S16. Differences in the time-height cross section of cloud factor CF (shading; unitless) and PM$_{2.5}$ concentration (contour; µg m$^{-3}$) averaged over R1 between 10× and CLEAN.**

[Figure]

Figure S17. Differences with time (abscissa; from 00Z on December 14 to 02Z on December 17) and height (ordinate) in (a) CDNC (shading; $10^7$ kg$^{-1}$) and cloud effective radius (μm), (b) CINC (shading; $10^5$ kg$^{-1}$) and ice cloud effective radius (contour; μm), and (c) vertical velocity (shading; cm s$^{-1}$) and latent heating (contour; K d$^{-1}$) averaged over R1 between 10× and CLEAN.

[Figure]

**Figure S18. Differences with time (abscissa) and height (ordinate) in (a) cloud water (shading; $10^{-5}$ kg kg$^{-1}$) and CDNC (contour; $10^7$ kg kg$^{-1}$), (b) cloud ice (shading; $10^{-5}$ kg kg$^{-1}$) and CINC (contour; $10^4$ kg$^{-1}$), (c) rain (shading; $10^{-5}$ kg kg$^{-1}$) and rain number concentration (contour; $10^5$ kg$^{-1}$), (d) snow (shading; $10^{-4}$ kg kg$^{-1}$) and snow number concentration (contour; $10^3$ kg$^{-1}$), and (e) graupel (shading; $10^{-4}$ kg kg$^{-1}$) and graupel number concentration (contour; $10^3$ kg$^{-1}$) between 10× and CLEAN averaged over R1.**

[Figure]

**Figure S19.** Differences with time (abscissa) and height (ordinate) in latent heat release (K d⁻¹) from (a) condensation, (b) deposition, and (c) freezing processes between 10× and CLEAN averaged over R1 for warm cloud. (d–f) Same as (a–c) but for cold cloud.

[Figure]

Figure S20. Spatial distribution of wind shear (m s⁻¹) on (a) December 14 and (b) December 15 in 2013 in the CTL run. Wind shear is calculated as differences between maximum wind speed and minimum wind speed at 0–10 km. Spatial distribution of column-integrated water vapor (shading; mm day⁻¹) and 925-hPa wind (vector; m s⁻¹) on (c) December 14 and (d) December 15 in 2013 in CTL.

[Figure]

**Figure S21. Distribution with time (abscissa) and height (ordinate) in (a) cloud water (shading;  $10^{-5}$ kg kg$^{-1}$) and CDNC (contour; unit: $10^7$ kg$^{-1}$), (b) cloud ice (shading;  $10^{-5}$ kg kg$^{-1}$) and CINC (contour;  $10^4$ kg$^{-1}$), (c) rain (shading;  $10^{-5}$ kg kg$^{-1}$) and rain number concentration (contour;  $10^5$ kg$^{-1}$), (d) snow (shading;  $10^{-4}$ kg kg$^{-1}$) and snow number concentrations (contour;  $10^3$ kg$^{-1}$), and (e) graupel (shading;  $10^{-4}$ kg kg$^{-1}$) and graupel number concentration (contour;  $10^3$ kg$^{-1}$) averaged over R2  from CTL run. **

[Figure]

[Figure]

**Figure S2.** Differences with time (abscissa) and height (ordinate) in (a) cloud water (shading;  $10^{-5}$ kg kg$^{-1}$) and CDNC (contour;  $10^{7}$ kg$^{-1}$), (b) cloud ice (shading;  $10^{-5}$ kg kg$^{-1}$) and CINC (contour;  $10^{4}$ kg$^{-1}$), (c) rain (shading;  $10^{-5}$ kg kg$^{-1}$) and rain number concentration (contour;  $10^{5}$ kg$^{-1}$), (d) snow (shading;  $10^{-4}$ kg kg$^{-1}$) and snow number concentrations (contour;  $10^{3}$ kg$^{-1}$), and (e) graupel (shading;  $10^{-4}$ kg kg$^{-1}$) and graupel number concentration (contour;  $10^{3}$ kg$^{-1}$) between CTL and CLEAN  averaged over the region in 24°–25°N, 110°–112°E.

[Figure]

[Figure]

**Figure S23.** Differences with time (abscissa) and height (ordinate) in latent heat release ( K d⁻¹) from (a) condensation, (b) deposition, and (c) freezing processes between CTL and CLEAN  averaged over R2  for  warm cloud. (d–f) Same as (a–c) but from cold cloud.

[Figure]

[Figure]

**Figure S242414.** Differences with time (abscissa) and height (ordinate) in (a) cloud water (shading;  $10^{-5}$ kg kg$^{-1}$) and CDNC (contour;  $10^{7}$ kg$^{-1}$), (b) cloud ice (shading;  $10^{-5}$ kg kg$^{-1}$) and CINC (contour;  $10^{4}$ kg$^{-1}$), (c) rain (shading;  $10^{-5}$ kg kg$^{-1}$) and rain number concentration (contour;  $10^{5}$ kg$^{-1}$), (d) snow (shading;  $10^{-4}$ kg kg$^{-1}$) and snow number concentrations (contour;  $10^{3}$ kg$^{-1}$), and (e) graupel (shading;  $10^{-4}$ kg kg$^{-1}$) and graupel number concentration (contour;  $10^{3}$ kg$^{-1}$) between 10× and CLEAN  averaged over R2.

Warm Cloud                    Cold Cloud

[Figure]

[Figure]

Figure S25251515. **Differences** with time (abscissa) and height (ordinate) in latent heat release (unit: K d⁻¹) from (a) condensation, (b) deposition, and (c) freezing processes between 10× and CLEAN (i.e. 10× minus CLEAN) averaged over R2 the region in 24°–25°N, 110°–112°E for the warm cloud. (d–f) Same as (a–c) but for rom cold cloud. Only anomalies that exceed 90% significance level are depicted with and contour. Zero-value contour lines are omitted, and negative values are dashed. The contour interval is 3 K d⁻¹. Note the blank represent the values are within 3 K d⁻¹.

[Figure]

**Figure S262617. Differences in 2-m temperature (K) between (a) CTL and CLEAN and (b) D1 and CLEAN on December 15. (c, d) Same as (a, b) but for column water vapor (mm).**

[Figure]

**Figure S272718. Differences in precipitation (mm) between (a) CTL and CLEAN and (b) D1 and CLEAN on December 15 based on domain 1 output.**

**References**

[revised manuscript text omitted]

---

## Author Response (AR4)

Dear editor,

  Thanks for your suggestions. We have incorporated your suggestions into the manuscript.
Thank you again.

--

Regards,

Steve
* * *
YIM, Hung-Lam Steve, Ph.D.

Assistant Professor
Department of Geography and Resource Management

The Chinese University of Hong Kong, Shatin, Hong Kong
Tel: (852) 3943 6534
Fax: (852) 2603 5006
Email: steveyim@cuhk.edu.hk
GRMD@CUHK: http://www.grm.cuhk.edu.hk/eng/

**p2 l27 - flawed sentence**
Original:
Khain (2009) and Fan et al. (2007) have reported that increases in humidity generate more
condensate than lose, resulting in more precipitation from deep convective clouds.
Revised:
Khain (2009) and Fan et al. (2007) have reported that increases in humidity generate more
condensation with aerosols, resulting in more precipitation from deep convective clouds.

**p11 l19 - please clarify with respect to what CTL increases**
Original:
However, rain water shows decrease during all the time instead of an increase after 15Z when
precipitation increases in the CTL run.
Revised:
However, rain water shows a decrease during all the time instead of an increase after 15Z in the
CTL run when comparing with that in CLEAN run.

**p11 l27**
Original:
The average precipitation over Guangdong Province on December 15 decreases by 1.0 mm in
10×, whereas it increases by 1.4 mm in CTL.
Revised:

The average precipitation over Guangdong Province on December 15 decreases by 1.0 mm in 10× while increases by 1.4 mm in CTL by comparing with that in CLEAN.

**p13 l14**
Original:
The average precipitation over Guangdong Province decreases by 1.0 mm in 10× but increases by 1.4 mm in CTL. These results indicate that aerosol concentration in 10× exceeds the optimal aerosol loading for convective invigoration and suppresses the rainfall amount instead.
Revised:
As discussed above, the average precipitation over Guangdong Province shows a decrease in CTL but an increase in CTL when comparing with that in CLEAN. These opposite changes indicate that aerosol concentration in 10× exceeds the optimal aerosol loading for convective invigoration and thus suppresses the rainfall amount instead.

**p12 l18 - revise wording, unclear**
Original:
On average, ACI enhances precipitation over R1. Conversely, ARI partially compensates for the precipitation increase by 14%.
Revised:
On average, ACI enhances precipitation over R1, while ARI reduces precipitation, offsetting the precipitation increase through ACI by 14%.